# Uncertainty in Federated Granger Causality: From Origins to Systemic Consequences

## Abstract

Granger Causality (GC) provides a rigorous framework for learning causal structures from time-series data. Recent federated variants of GC have targeted distributed infrastructure applications (e.g., smart grids) with distributed clients that generate high-dimensional data bound by data-sovereignty constraints. However, Federated GC algorithms only yield deterministic point estimates of causality and neglect uncertainty. This paper establishes the first methodology for rigorously quantifying uncertainty and its propagation within federated GC frameworks. We systematically classify sources of uncertainty, explicitly differentiating aleatoric (data noise) from epistemic (model variability) effects. We derive closed-form recursion expressions modeling the evolution of uncertainty through client-server interactions, and identify four novel cross-covariance components that couple data uncertainties with model parameter uncertainties across the federated architecture. Moreover, we define rigorous convergence conditions for these uncertainty recursions and obtain explicit steady-state variances for both server and client model parameters. More importantly, our convergence analysis demonstrates that steady-state variances depend exclusively on client data statistics, thus eliminating the dependence on initial epistemic priors and enhancing robustness. Empirical evaluations on synthetic benchmarks and real-world industrial datasets demonstrate that explicitly characterizing uncertainty significantly improves the reliability and interpretability of federated causal inference. These results enable robust root-cause analysis in safety-critical privacy-constrained and distributed infrastructures.

## 1 Introduction

Complex industrial systems such as smart grids, distributed manufacturing, and supply chain networks are comprised of geographically distributed, tightly coupled system components (clients) that interact dynamically and require seamless coordination for safe and reliable operations. The resulting interactions also mean that any disruptions, such as equipment failures or cyber intrusions, can propagate quickly, triggering cascading effects across the entire system. Mitigating these challenges requires risk-aware causal models that can identify causal influences between clients and quantify their associated uncertainties. This ensures that limited resources for mitigation are efficiently allocated to prevent or minimize cascading disruptions, especially in safety-critical applications.

Granger Causality (GC) is well-suited for learning causal structures from time-series data  Shojaie & Fox (2022); Wismüller et al. (2021); Tank et al. (2021). Conventional GC methods rely on data centralization. Clients in our problem setting cannot readily share their data because they are equipped with modern sensors that generate high-dimensional data, and any downsampling or aggregation of this data destroys critical diagnostic information. Additionally, many clients are bound by data-privacy stipulations.

Recent advances in **federated Granger causality** (FedGC) partly address these challenges by exchanging only sufficient statistics rather than raw data  Mohanty et al. (2025). Nonetheless, existing FedGC methods provide only deterministic point estimates and neglect the uncertainty introduced by data noise and model variability. To the best of our knowledge, this is the first methodology that rigorously quantifies uncertainty and its propagation within FedGC frameworks.

Without quantifying uncertainties in causal models, decision-makers may underestimate risks, leading to catastrophic failures. For instance, during the 2003 Northeast blackout Muir & Lopatto (2004), control room alarms failed to trigger, and operators believed the system was operating normally. This binary fault detection approach ignored uncertainty and missed early warning signs. Had the variances on inter-utility causal interactions been monitored rather than relying on single-point alarm estimates, operators might have seen a widening of predictive confidence intervals, providing sufficient warning to mitigate the cascade. Unfortunately, the cascade resulted in a massive outage.

This paper establishes the first theoretical foundation for analyzing how uncertainties propagate in the FedGC framework. We address three fundamental questions:
1. **Sources:** What are the primary contributors to uncertainty in federated causal learning?
2. **Propagation:** How do uncertainties compound as clients and servers iteratively optimize loss?
3. **Impact:** What is the cumulative effect of these uncertainties on the fidelity of causal estimates?

**Contributions:** The key technical contributions of our paper are:

- A systematic classification of uncertainty sources in federated causal learning, distinguishing between *data noise* (aleatoric), and *model variability* (epistemic) effects.

- Closed-form propagation expressions that track how these uncertainty move through *client–to–server*, and *server–to–client* communication channels, and *within the client*, and *within the server* paths, revealing four previously unrecognized *cross-covariance terms* that couple data and parameters across the FedGC framework.

- Spectral-radius based conditions that guarantee convergence of all covariance recursions leading to explicit solutions for the steady-state variances of the server and client models.

- Theoretical results on convergence, showing that these steady-state variances depend only on raw data statistics of the clients (aleatoric), thereby eliminating any influence of the initial epistemic priors in the FedGC framework.

## 2 LITERATURE REVIEW

In centralized settings, uncertainty propagation under gradient-based optimization has been studied extensively. Recent works Wang et al. (2023); Durasov et al. (2024); Gawlikowski et al. (2023) analyze how noise from stochastic gradients or perturbations propagates through updates, while others Chan et al. (2024); Huseljic et al. (2021); Meinert et al. (2023); Hofman et al. (2024) formalize the coupling between aleatoric and epistemic uncertainties Hüllermeier & Waegeman (2021). These analyses assume centralized training with full data access and do not generalize to federated settings.

In federated learning, uncertainty quantification has advanced mainly through Bayesian methods, including federated Bayesian neural networks Yurochkin et al. (2019), personalized inference Kotelevskii et al. (2022); Zhang et al. (2022), ensemble strategies Chen & Chao (2020), and Monte Carlo dropout Park et al. (2022). While effective for horizontal IID data Yang et al. (2019), these approaches rely on approximate posteriors (variational inference, sampling, dropout) and typically treat aleatoric and epistemic uncertainties independently. VertiBayes Van Daalen et al. (2024) extends Bayesian inference to vertical partitions but remains static, lacking temporal dynamics and client–server uncertainty interactions. Beyond Bayesian approaches, federated Kalman filtering Xing & Xia (2016); Baucas & Spachos (2023) and distributed state estimation Korres (2010); Primadianto & Lu (2016) propagate uncertainty recursively but ignore cross-client causality. Causal federated works such as FedGC Mohanty et al. (2025) focus on deterministic recovery, while variational inference methods Guo et al.; Vedadi et al. (2024) approximate posteriors without modeling cross-covariances. Recent horizontally federated causal-discovery methods, including Guo et al. (2024b;a), recover causal graphs under IID partitions but do not extend to vertically partitioned dynamical systems with temporal coupling. Robust aggregation Pillutla et al. (2022); Li et al. (2021) mitigates heterogeneity but remains agnostic to compounding uncertainty in causal graphs.

Our work departs from these directions by unifying causal structure learning with federated uncertainty propagation. Unlike Bayesian FL methods that assume horizontal partitioning or decouple uncertainties, we explicitly trace how aleatoric noise and epistemic variability interact via cross-covariance terms. This captures uncertainty propagation in federated time-series data while providing a causal interpretation of client–server interactions.

## 3 PRELIMINARIES

**State Space Model.** The underlying system is modeled as a linear time-invariant (LTI) state–space:

$$h^t \; = \; A \, h^{t-1} + w^t, \quad y^t \; = \; C \, h^t + v^t \tag{1}$$

where $h^t \in \mathbb{R}^p$ are the **latent low-dimensional states** and $y^t \in \mathbb{R}^d$ are the **measured high-dimensional data** at time $t$ with $d >> p$. $A$ and $C$ are the constant state-transition and observation matrices; $w^t$ and $v^t$ are zero-mean i.i.d. Gaussian system and measurement noise with covariances $Q$ and $R$, respectively. We make the following assumptions about the LTI state space model:

1. There are $M$ subsystems such that $h^t = [h_1^t, \cdots, h_M^t], y^t = [y_1^t, \cdots, y_M^t]$. Each of these subsystems is a client for our problem setting. The states are such that, $h_m^t \in \mathbb{R}^{p_m}$, and $y_m^t \in \mathbb{R}^{d_m}$ with $d_m >> p_m$, and $\sum_{m=1}^M p_m = p$, and $\sum_{m=1}^M d_m = d$.

2. The observation matrix $C$ is block-diagonal i.e., $C = \mathrm{diag}(C_{11}, \cdots, C_{MM})$ where each block $C_{mm} \in \mathbb{R}^{d_m \times p_m}$. The block $C_{mm}$ is known at client $m$.

3. The state-transition matrix $A$ is not block-diagonal i.e., $\exists\, n \neq m$ s.t., $A_{mn} \neq 0$ with $A_{mn} \in \mathbb{R}^{p_m \times p_n}$. Each client $m$ knows (or can locally compute) its diagonal block $A_{mm}$, while the off-diagonal blocks $A_{mn} \forall n \neq m$ are **unknown**.

**Granger Causality.** A time series $h_n$ is said to *Granger-cause* another series $h_m$ if the inclusion of past values of $h_n$ improves the prediction of $h_m$. In the state-space setting, this notion is captured by the off-diagonal entries of the state-transition matrix $A$. For instance, in a system given by,

$$\begin{pmatrix} h_1^t \\ h_2^t \end{pmatrix} = \begin{pmatrix} A_{11} & A_{12} \\ A_{21} & A_{22} \end{pmatrix} \begin{pmatrix} h_1^{t-1} \\ h_2^{t-1} \end{pmatrix} + \begin{pmatrix} w_1^t \\ w_2^t \end{pmatrix} \tag{2}$$

the series $h_2$ Granger-causes $h_1$ precisely when $A_{12} \neq 0$. More generally, $A_{mn} \neq 0$ indicates that past values of $h_n$ influence the future of $h_m$, revealing a directed causal link from client $n$ to $m$. Estimating $A_{mn}$ in a decentralized system is the goal of the federated GC framework.

### 3.1 FEDERATED GRANGER CAUSALITY

FedGC is a server-client framework with $M$ different clients having **unknown interdependencies** across them. We discuss the details of the client and server models in the FedGC framework.

**Client Model.** Each client $m$ models its subsystem as an LTI state–space using only the local (diagonal) blocks $A_{mm} \in \mathbb{R}^{p_m \times p_m}$ and $C_{mm} \in \mathbb{R}^{d_m \times p_m}$. A Kalman filter based "***local client model***" is used to compress the high-dimensional data (measurement) $y_t^{(m)} \in \mathbb{R}^{d_m}$ into a low-dimensional state estimate $\hat{h}_t^{(m)} \in \mathbb{R}^{p_m}$ with $d_m >> p_m \forall m \in \{1, \cdots, M\}$ such that:

$$h_{m,c}^t \; = \; A_{mm} \, \hat{h}_{m,c}^{t-1}, \quad \hat{h}_{m,c}^t \; = \; h_{m,c}^t \; + \; K_m \big(y_m^t - C_{mm}(h_{m,c}^t)_c\big) \tag{3}$$

where $K_m$ is the Kalman gain computed from $(A_{mm}, C_{mm}, Q_{mm}, R_{mm})$. The local client model (Kalman Filter) ignores cross-client dynamics $A_{mn}$ ($n \neq m$). In order to compensate for these cross-client dynamics, the client states are *augmented* with machine learning (ML) models to obtain an "***augmented client model***" as follows:

$$\hat{h}_{m,a}^t = \hat{h}_{m,c}^t + \theta_m \, y_m^t, \quad h_{m,a}^t = A_{mm} \hat{h}_{m,a}^t \tag{4}$$

where $\theta_m \in \mathbb{R}^{p_m \times d_m}$ is a learnable matrix that captures missing cross-client effects. The loss function $(L_m)_a$ is optimized at client $m$ with $\theta_m$ being learned using Gradient-descent based update,

$$(L_m)_a = \|y_m^t - C_{mm} h_{m,a}^t\| \text{ s.t., } \quad \theta_m^{t+1} = \theta_m^t \; - \; \eta_1 \, \nabla_{\theta_m^t} (L_m)_a \; - \; \eta_2 \, \nabla_{\theta_m^t} L_s \tag{5}$$

In the above equation, $\eta_1, \eta_2$ are the learning rates corresponding to client $m$'s loss function $(L_m)_a$ and server model's loss function $L_s$ respectively (the server model is discussed in next paragraph).

**Remark 3.1.** In the gradient update equation above, while, $\nabla_{\theta_m^t} (L_m)_a$ can be computed locally at the client $m$, the gradient $\nabla_{\theta_m^t} L_s$ needs information from the server. However, the server cannot directly compute $\nabla_{\theta_m^t} L_s$ as the gradient is w.r.t. client model parameter $\theta_m$. Therefore, the FedGC

framework adopts chain rule to decompose it such that: $\nabla_{\theta_m^t} L_s = \left( \nabla_{\hat{h}_{m,a}^t} L_s \right) \left( \nabla_{\theta_m^t} \hat{h}_{m,a}^t \right)$. A key advantage of this decomposition is that the first factor i.e., $\nabla_{\hat{h}_{m,a}^t} L_s$ can be communicated from the server to client $m$, while $\nabla_{\theta_m^t} \hat{h}_{m,a}^t$ can be computed locally at client $m$.

**Server Model.** The server collects from each client $m$ the pair $\left( \hat{h}_{m,c}^{t-1}, \hat{h}_{m,a}^{t-1} \right)$ and stacks them into

$$H_c^{t-1} = \left[ \hat{h}_{1,c}^{t-1}, \cdots, \hat{h}_{M,c}^{t-1} \right], \quad H_a^t = \left[ A_{11} \hat{h}_{1,a}^{t-1}, \cdots, A_{MM} \hat{h}_{M,a}^{t-1} \right] \tag{6}$$

The diagonal matrix $diag(A_{11}, \cdots, A_{MM})$ is assumed to be known at the server. However, the off-diagonal blocks (representing the GC) $A_{mn} \forall n \neq m$ are **unknown**. The server model's goal is to predict the next-step state $H_s^t$ as follows:

$$H_s^t = \left[ h_{1,s}^t, \cdots, h_{M,s}^t \right] \text{ s.t., } \quad h_{m,s}^t = A_{mm} \hat{h}_{m,c}^{t-1} + \sum_{n \neq m} \hat{A}_{mn} \hat{h}_{n,c}^{t-1}. \tag{7}$$

where the estimated causality $\hat{A}_{mn} \forall n \neq m$ are learned by minimizing the server loss function $L_s$ with gradient-descent based updating used to learn $\hat{A}_{mn} \forall n \neq m$ with server learning rate $\gamma$ s.t.

$$L_s = \left\| H_a^t - H_s^t \right\|_2^2, \quad \hat{A}_{mn}^{t+1} = \hat{A}_{mn}^t - \gamma \nabla_{\hat{A}_{mn}^t} L_s \tag{8}$$

The server then sends $\nabla_{\hat{h}_{m,a}^t} L_s$ to client $m$ for subsequent client parameter updating.

## 4 PROBLEM FORMULATION

The reliance of the FedGC framework on deterministic point estimates of the Granger causal parameters $\hat{A}_{mn}$ $(\forall n \neq m)$ and the client parameters $\theta_m$ overlooks the influence of data noise and model variability. In this work, we rigorously characterize and quantify how uncertainty propagates through the federated pipeline. Our analysis is anchored around three core objectives:

1. Characterize the **sources of uncertainty**—namely, aleatoric noise from client data and epistemic variability in model parameters.

2. Derive exact **propagation recursions** that track how these uncertainties evolve across client–server communication and local/global updates within the client/server.

3. Analyze the steady-state **impact of uncertainty**, proving that under mild assumptions, the resulting variances depend only on data noise and fixed gains, not on initial model priors.

## 5 SOURCES OF UNCERTAINTY

In this section, we partition the stochastic elements of FedGC into two disjoint categories: **(ii)** *aleatoric* noise originating in the data (measurements) collected by each client, and **(ii)** *epistemic* variance arising from incomplete knowledge of both client-side and server-side model parameters.

**(1) Aleatoric.** Aleatoric uncertainty captures the irreducible measurement noise $\epsilon_m^t$ in client $m$'s data i.e., $y_m^t$. When a client updates its parameter $\theta_m^t$ via gradient descent, this data noise $\epsilon_m^t$ propagates directly into the gradient $\nabla_{\theta_m} (L_m)_a$ and hence into the variance of $\theta_m^t$. Likewise, it also enters the augmented state $\hat{h}_{m,a}^t = \hat{h}_{m,c}^t + \theta_m^t y_m^t$. Since $\hat{h}_{m,a}^t$ is communicated to the server from clients, this data noise also influences the estimation of Granger causality $\hat{A}_{mn}^t \forall n \neq m$ (also called the *server model parameter*). Furthermore, this data noise affects the server loss $L_s$ and shows up in the gradients $\nabla_{\hat{h}_{m,a}^t} L_s$ $\forall m$ communicated from the server to the clients.

**(2) Epistemic.** Epistemic uncertainty reflects our lack of knowledge about the model parameters —both the client parameters $\theta_m$ and the server parameters $A_{mn}$. We assume both of these parameters to be **random variables** in our problem setting. Sampling from a prior $\theta_m^0 \sim \mathcal{D}_1(\mu_{\theta_m}^0, \Sigma_{\theta_m}^0)$ and $A_{mn}^0 \sim \mathcal{D}_2(\mu_{A_{mn}}^0, \Sigma_{A_{mn}}^0)$ where, $\mathcal{D}_1$ and $\mathcal{D}_2$ are any location-scale distributions, we refine $\theta_m^0$, and $\hat{A}_{mn}^0$ using gradient-descent updates. By accumulating sufficient gradient-descent iterations, we reduce this epistemic uncertainty and thereby increase our confidence in the estimated causality.

Table 1: Summary of notation (client index $m$, time index $t$)

| Symbol | Meaning | Shape / Statistics |
|---|---|---|
| $y_m^t$ | Raw data for client $m$ at time $t$ | $\in \mathbb{R}^{d_m}$; $\mu_{y_m}^t = \mathrm{E}[y_m^t]$, $\Sigma_{y_m}^t = \mathrm{Var}(y_m^t)$ |
| $\theta_m^t$ | Model parameter at client $m$ | $\in \mathbb{R}^{p_m \times d_m}$ |
| $v_m^t$ | Vectorised $\theta_m^t$ i.e., $v_m^t = \mathrm{Vec}(\theta_m^t)$ | $\mu_{\theta_m}^t = \mathrm{E}[v_m^t]$, $\Sigma_{\theta_m}^t = \mathrm{Var}(v_m^t)$ |
| $\Omega_m^t$ | Parameter-data covariance at client $m$ | $\mathrm{Cov}(v_m^t, y_m^t)$ |
| $\Lambda_m^t$ | Client parameter-state covariance | $\mathrm{Cov}(v_m^t, \hat{h}_{m,a}^t)$ |
| $\Psi_{mn}^t$ | Server-client parameter covariance | $\mathrm{Cov}(a_{mn}^t, v_m^t)$ |
| $\Gamma_{mn}^t$ | Cross-covariance between $a_{mn}^t$ and $\hat{h}_{m,a}^t$ | $\mathrm{Cov}(a_{mn}^t, \hat{h}_{m,a}^t)$ |
| $\hat{h}_{m,a}^t$ | Augmented state estimate at client $m$ | $\Sigma_{h_m}^t = \mathrm{Var}(\hat{h}_{m,a}^t)$ |
| $\hat{A}_{mn}^t$ | Server parameter estimate ($n \to m$) | $\in \mathbb{R}^{p_m \times p_n}, \forall n \neq m$ |
| $a_{mn}^t$ | Vectorised $\hat{A}_{mn}^t$ i.e., $a_{mn}^t = \mathrm{Vec}(\hat{A}_{mn}^t)$ | $\Sigma_{A_{mn}}^t = \mathrm{Var}(a_{mn}^t)$ |
| $L_s$ | Loss function at server | $\in \mathbb{R}^1$ (Scalar) |
| $(L_m)_a$ | Loss function at client $m$ | $\in \mathbb{R}^1$ (Scalar) |
| $g_{m,s}^t$ | Gradient of server loss $L_s$ w.r.t $\hat{h}_{m,a}^t$ | $\in \mathbb{R}^{p_m}$; $\mathrm{Var}(g_{m,s}^t)$ |
| $\gamma$ | Learning rate of the server model | $\in \mathbb{R}^1$ (Scalar) |
| $\eta_1, \eta_2$ | Learning rates of the client model | $\in \mathbb{R}^1$ (Scalars) |

**Assumptions.** Based on the sources of uncertainty, we make the following assumptions:

• **(A1) *Stochasticity.*** For every client $m$, the *client model parameter* $\theta_m^t$, and *client data* $y_m^t$ are random variables. **The local client state $\hat{h}_{m,c}^t$ is deterministic**. Consequently, the only randomness entering the augmented states $\hat{h}_{m,a}^t = \hat{h}_{m,c}^t + \theta_m^t y_m^t$ comes from $\theta_m^t$ and $y_m^t$. In the server model, the Granger causal estimation $\hat{A}_{mn}^t \, \forall n \neq m$ (also called the *server model parameter*) is random.

• **(A2) *Model Parameters.*** The server parameters $A_{mn}^t$, $n \neq m$, are mutually independent across block-rows and times. As a consequence, for any distinct clients $m \neq n$, the induced parameters $\theta_m^t$ and $\theta_n^t$ have zero cross-covariance at every time $t$, i.e. $\mathrm{Cov}(\theta_m^t, \theta_n^t) = 0$, $\forall m \neq n$, $\forall t$. We formally establish this result in Lemma D.1, and Proposition D.2.

• **(A3) *Prior.*** The initial server and client parameters are independent, i.e., $\hat{A}_{mn}^0 \perp\!\!\!\perp \theta_m^0$, and the initial client parameters are uncorrelated, i.e., $\mathrm{Cov}(\theta_m^0, \theta_n^0) = 0 \;\; \forall n \neq m$.

• **(A4) *Stationarity.*** Client data are weakly stationary with time-invariant first and second moments: $\mathrm{E}[y_m^t] = \mu_{y_m}$, $\mathrm{Var}(y_m^t) = \Sigma_{y_m}, \forall t$

• **(A5) *Noise.*** The underlying states $h^t$ in the state–space model (see Eq 1) are noiseless. All stochasticity enters through additive data (measurement) noise: $y_m^t = \mu_{y_m} + \varepsilon_m^t$, with $\mathrm{E}[\varepsilon_m^t] = 0$, and $\mathrm{Var}(\varepsilon_m^t) = \Sigma_{y_m}$. We formally show this using the Proposition D.3.

## 6 UNCERTAINTY PROPAGATION

*Notation.* Table 1 gives an overview of the symbols discussed in Sections 3, 4, and 5.

### 6.1 CROSS-COVARIANCES

The FedGC framework intertwines data $y_m^t$, client state $\hat{h}_{m,a}^t$, client model parameter $v_m^t$, and server model parameter $a_{mn}^t$, creating four essential cross-covariances: $\Omega_m^t$, $\Lambda_m^t$, $\Gamma_{mn}^t$ and $\Psi_{mn}^t$. This section provides explicit recursions so they can be propagated together with the individual variances. From Table 1, we know that $v_m^t = \mathrm{Vec}(\theta_m^t)$, $\mu_{\theta_m}^t = \mathrm{E}[v_m^t]$, $\Sigma_{\theta_m}^t = \mathrm{Var}(v_m^t)$, $\mu_{y_m}^t = \mathrm{E}[y_m^t]$ and $\Omega_m^t = \mathrm{Cov}(v_m^t, y_m^t)$. Using the above notations, Proposition 6.1 shows that $v_m^t$, and $y_m^t$ are dependent with a non-zero cross-covariance $\Omega_m^t$.

**Proposition 6.1** (**Client Model-Client Data Dependence**). *Assume* $\mathrm{Var}(y_m^{t-1}) > 0$. *Then under the federated Granger-causality updates,* $\Omega_m^t := \mathrm{Cov}(v_m^t, y_m^t) \neq 0$.

Using Eq 4 we know that client states $\hat{h}^t_{m,a}$ are a function of the client data $y^t_m$. Since $\Omega^t_m \neq 0$, there must exists a dependence between the client model and client states. Proposition 6.2 analyzes the evolution of the cross-covariance between the client's model parameter $v^t_m$, and the states $\hat{h}^t_{m,a}$.

**Proposition 6.2** (**Client Model-Client State Dependence**). *Let* $\Lambda^t_m = \text{Cov}(v^t_m, \hat{h}^t_{m,a})$. *Then we have the following recursion within the client,* $\Lambda^t_m = \Sigma^t_{\theta_m}(I_{d_m} \otimes \mu^t_{y_m}) + \Omega^t_m(\mu^t_{v_m} \otimes I_{d_m})$,

Due to the iterative communication between client and server, the client model dynamics are coupled with that of the server model in a feedback loop. Essentially, the client's noisy state estimates $\hat{h}^t_{m,a}$ affect the server estimate $a^t_{mn}$, and the server's uncertain $a^t_{mn}$ in turn influences subsequent client state estimates. This effect is captured as the cross-covariance term $\Gamma^t_{mn}$ given in Lemma 6.3.

**Lemma 6.3** (**Client State-Sever Model Dependence**). *The cross-covariance term* $\Gamma^t_{mn}$ := $\text{Cov}(a^t_{mn}, \hat{h}^t_{m,a})$ *follows the recursive equation:* $\Gamma^{t+1}_{mn} = D^t_n \Gamma^t_{mn} + 2\gamma B^t_{mn} \Sigma^t_{h_m}$ *where,* $D^t_n = (I - 2\gamma \hat{h}^t_{n,c} \hat{h}^{t\top}_{n,c}) \otimes I, B^t_{mn} = \hat{h}^t_{n,c} \otimes A_{mm}$, *and* $\Sigma^t_{h_m} = \text{Var}(\hat{h}^t_{m,a})$.

Because the client parameter and its augmented state are already linked through the cross-covariance in Proposition 6.2, the client state–to–server model coupling of Lemma 6.3 propagates that link one step further, yielding a direct client model–to-server model dependence captured in Lemma 6.4.

**Lemma 6.4** (**Client Model-Server Model Dependence**). *The term* $\Psi^t_{mn}$ := $\text{Cov}(a^t_{mn}, v^t_m)$ *evolves as,* $\Psi^{t+1}_{mn} = D^t_n \Psi^t_{mn} H^{t\top}_m + D^t_n \Gamma^t_{mn} G^{t\top}_m - D^t_n \Sigma^t_{A_{mn}} P^{t\top}_m + 2\gamma B^t_{mn} \Lambda^t_m H^{t\top}_m + 2\gamma B^t_{mn} \Sigma^t_{h_m} G^{t\top}_m - 2\gamma B^t_{mn} \Gamma^{t\top}_{mn} P^{t\top}_m$, *with the following gain matrices,* $B^t_{mn} = \hat{h}^t_{n,c} \otimes A_{mm}$,

$$D^t_n = (I - 2\gamma \hat{h}^t_{n,c} \hat{h}^{t\top}_{n,c}) \otimes I, G^t_m = 2\eta_1(y^t_m \otimes (C_{mm} A_{mm})^\top), P^t_m = -2\eta_2(y^t_m \otimes A^\top_{mm})$$

*and* $H^t_m = I_{p_m d_m} - 2\eta_1(y^t_m y^{t\top}_m) \otimes ((C_{mm} A_{mm})^\top C_{mm} A_{mm}) - 2\eta_2(y^t_m y^{t\top}_m) \otimes (A^\top_{mm} A_{mm})$

## 6.2 During Communication

This section characterizes the communication channel as the conduit through which every existing uncertainty—client-side data noise, client-parameter variance, and server-parameter variance—is redistributed at each iteration. Specifically we analyze the uncertainty propagation in both **(I)** *client-to-server*, and **(II)** *server-to-client* communication channel.

**(I) Client to Server.** At iteration $t$, client $m$ sends its augmented states $\hat{h}^t_{m,a}$ to the server. While $\hat{h}^t_{m,a}$ naturally captures $\Sigma^t_{\theta_m}$ and $\Sigma^t_{y_m}$, it may also include cross-covariance $\Omega^t_m := \text{Cov}(v^t_m, y^t_m)$. Lemma 6.5 provides a closed-form expression for the uncertainty in $\hat{h}^t_{m,a}$ using $\Sigma^t_{\theta_m}, \Sigma^t_{y_m}$, and $\Omega^t_m$.

**Lemma 6.5** (**Uncertainty in Client to Server Communication**). *Let* $\kappa_m = \text{tr}(\Sigma^t_{y_m}) + \|\mu^t_{y_m}\|^2$. *Then the variance in the* $\hat{h}^t_{m,a}$ *is,* $\Sigma^t_{h_m} = \kappa_m \Sigma^t_{\theta_m} + \Omega^t_m(\mu^t_{y_m} \otimes I_{p_m})^\top + (\mu^t_{y_m} \otimes I_{p_m})\Omega^{t\top}_m$.

**(II) Server to Client.** At iteration $t$, the server's uncertainty is encoded in the random matrix $\hat{A}^t_{mn}$. Instead of sending $\hat{A}^t_{mn}$, the server computes and transmits the gradient: $g^{t+1}_{m,s}$ := $\nabla_{\hat{h}^t_{m,a}} L^t_s$ where $L^t_s = \|A_{mm}[\hat{h}^t_{m,a} - \hat{h}^t_{m,c}] - \sum_{n \neq m} \hat{A}^t_{mn} \hat{h}^t_{n,c}\|^2$. This gradient inherits uncertainty from both $\hat{A}^t_{mn}$ and $\hat{h}^t_{m,a}$, propagating the server's model uncertainty to client $m$. Lemma 6.6 shows that $g^t_{m,s}$ captures the uncertainty in the server parameters, client states, and their cross-covariance.

**Lemma 6.6** (**Uncertainty in Server to Client Communication**). *With notation as above, the uncertainty in the gradient communicated by the server is given by,* $\text{Var}(g^{t+1}_{m,s}) = A^\top_{mm} U^t A_{mm}$, *where* $U^t = A_{mm} \Sigma^t_{h_m} A^\top_{mm} + \sum_{n \neq m}(h^t_{n,c} h^{t\top}_{n,c})\Sigma^t_{A_{mn}} - 2\sum_{n \neq m} A_{mm} \Gamma^t_{mn} h^{t\top}_{n,c}$.

## 6.3 Within Server

In Sections 6.1 and 6.2 , we quantified how **(i)** the client–server cross-covariance $\Gamma^t_{mn}$ , and **(ii)** client $m$'s state variance $\Sigma^t_{h_m}$ propagate during the iterative optimization of the FedGC framework. We now analyze their contribution to the propagation of the server's parameter uncertainty $\Sigma^t_{A_{mn}}$. Theorem 6.7 combines these components into a closed-form recursion for $\Sigma^t_{A_{mn}}$ within the server.

**Theorem 6.7 (Uncertainty Propagation within the Server).** *The server model parameter $a_{mn}^t$'s covariance evolves as,* $\Sigma_{A_{mn}}^{t+1} = D_n^t \Sigma_{A_{mn}}^t D_n^{t\top} + 4\gamma^2 \left(\hat{h}_{n,c}^t \otimes A_{mm}\right) \Sigma_{h_m}^t \left(\hat{h}_{n,c}^t \otimes A_{mm}\right)^\top + 2\gamma\left(D_n^t \Gamma_{mn}^t B_{mn}^{t\top} + B_{mn}^t \Gamma_{mn}^{t\top} D_n^{t\top}\right)$ *with* $D_n^t = \left(I - 2\gamma \hat{h}_{n,c}^t \hat{h}_{n,c}^{t\top}\right) \otimes I$, *and* $B_{mn}^t = \hat{h}_{n,c}^t \otimes A_{mm}$

## 6.4 WITHIN CLIENT

Each of the four cross-covariances discussed in Section 6.1 contribute to the propagation of uncertainty of the client model parameter $\theta_m$ (or, $v_m$ in vectorized form). Theorem 6.8 expresses the evolution of client model's variance $\Sigma_{\theta_m}^t$ in terms of the uncertainty in its states $\Sigma_{h_m}^{t-1}$, the server model $\Sigma_{A_{mn}}^{t-1}$, and those cross-covariances $\Omega_m^{t-1}, \Gamma_{mn}^{t-1}, \Psi_{mn}^{t-1}$ and $\Lambda_m^{t-1}$.

**Theorem 6.8 (Uncertainty Propagation within the Client).** *The client-parameter covariance $\Sigma_{\theta_m}^t$ obeys the following recursion:* $\Sigma_{\theta_m}^t = H_m^{t-1} \Sigma_{\theta_m}^{t-1} H_m^{t-1\top} + G_m^{t-1} \Sigma_{h_m}^{t-1} G_m^{t-1\top} + (X_m + X_m^\top) - \sum_{n \neq m}(Y_{mn} + Y_{mn}^\top) - \sum_{n \neq m}(Z_{mn} + Z_{mn}^\top) + \sum_{n \neq m} P_m^{t-1} \Sigma_{A_{mn}}^{t-1} P_m^{t-1\top},$

*where,* $X_m = H_m^{t-1} \Lambda_m^{t-1} G_m^{t-1\top}$, $Y_{mn} = H_m^{t-1} \Psi_{mn}^{t-1} P_m^{t-1\top}, Z_{mn} = G_m^{t-1} \Gamma_{mn}^{t-1} P_m^{t-1\top}$, *and,* $G_m^{t-1} := 2\eta_1 \left(y_m^{t-1} \otimes (C_{mm} A_{mm})^\top\right), P_m^{t-1} := -2\eta_2 \left(y_m^{t-1} \otimes A_{mm}^\top\right), H_m^{t-1} := I_{p_m d_m} - 2\eta_1 \left(y_m^{t-1} y_m^{t-1\top}\right) \otimes \left((C_{mm} A_{mm})^\top C_{mm} A_{mm}\right) - 2\eta_2 \left(y_m^{t-1} y_m^{t-1\top}\right) \otimes \left(A_{mm}^\top A_{mm}\right)$

## 7 STEADY-STATE IMPACT OF UNCERTAINTY

For tractability, we assume, **(I)** $\lim_{t\to\infty} y_m^t = \mu_{y_m}$, and **(II)** $\lim_{t\to\infty} \hat{h}_{m,c}^t = \hat{h}_{m,c} \forall m$

**Proposition 7.1 (Gain Matrices Convergence).** *Under the above assumptions, the gain matrices used in Section 6 converges as,* $\lim_{t\to\infty}\left(D_n^t, H_m^t, G_m^t, P_m^t\right) = \left(D_n, H_m, G_m, P_m\right)$ *where,* $D_n = (I - 2\gamma \hat{h}_{n,c} \hat{h}_{n,c}^\top) \otimes I$, $G_m = 2\eta_1\left(\mu_{y_m} \otimes (C_{mm} A_{mm})^\top\right)$, $P_m = -2\eta_2\left(\mu_{y_m} \otimes A_{mm}^\top\right)$ *and* $H_m = I_{p_m d_m} - 2\eta_1(\mu_{y_m} \mu_{y_m}^\top) \otimes ((C_{mm} A_{mm})^\top C_{mm} A_{mm}) - 2\eta_2(\mu_{y_m} \mu_{y_m}^\top) \otimes (A_{mm}^\top A_{mm})$.

Proposition 7.1 proves that the gains $(D_n^t, H_m^t, G_m^t, P_m^t)$ converge in the limit as $t \to \infty$.

**Proposition 7.2.** *If $\rho(D_n), \rho(H_m) < 1$ then we have,* $\lim_{t\to\infty}\left(\Gamma_{mn}^t, \Psi_{mn}^t\right) = \left(\Gamma_{mn}^\infty, \Psi_{mn}^\infty\right)$ *with* $\Gamma_{mn}^\infty = (I - D_n)^{-1} 2\gamma B_{mn} \Sigma_{h_m}^\infty$, & $\Psi_{mn}^\infty = (I - H_m \otimes D_n)^{-1} \text{Vec}\left(D_n \Gamma_{mn}^\infty G_m^\top - D_n \Sigma_{A_{mn}}^\infty P_m^\top\right)$.

With stable gains, Proposition 7.2 shows that the cross-covariance terms $\Gamma_{mn}^t$ and $\Psi_{mn}^t$ also converge, each given in closed form. Corollary 7.3 then expresses the client-state variance $\Sigma_{h_m}^\infty$ in terms of the client-parameter variance $\Sigma_{\theta_m}^\infty$ and the data moments; no other stochastic quantity survives.

**Corollary 7.3.** *The above assumptions lead to convergence of the uncertainty of the client states $\Sigma_{h_m}^t$ as follows:* $\lim_{t\to\infty} \Sigma_{h_m}^t := \Sigma_{h_m}^\infty = \kappa_m \Sigma_{\theta_m}^\infty + \Omega_m^\infty(\mu_{y_m} \otimes I)^\top + (\mu_{y_m} \otimes I)\Omega_m^{\infty\top}$, *where* $\kappa_m = \text{tr}(\Sigma_{y_m}) + \|\mu_{y_m}\|^2$ *and* $\Omega_m^\infty = \Sigma_{\theta_m}^\infty \mu_{y_m}$.

The key theoretical result on the impact of uncertainty quantification is mentioned next in Theorems 7.4, and 7.5, where we show that the steady-state uncertainties of server and client models are dependent only on the client data distribution (aleatoric uncertainty), and independent of the prior distribution of the parameters (epistemic uncertainty).

We know that the steady distribution of the client $m$'s raw data is given by $E[y_m] = \mu_{y_m}$, and $Var(y_m) = \Sigma_{y_m}$. Using $(\mu_{y_m}, \Sigma_{y_m})$ we define the following two terms that will be used in the Theorems 7.4, and 7.5: **(I)** $M_m = \mu_{y_m} \otimes I_{p_m}$, and **(II)** $\kappa_m = \text{tr}(\Sigma_{y_m}) + \|\mu_{y_m}\|^2$

**Theorem 7.4 (Convergence of Server Model's Uncertainty).** *Let $\rho(D_n) < 1$. Define $\mathcal{L}_n(X) := D_n X D_n^\top$ and $Q_{mn}(\Sigma) := 4\gamma^2 B_{mn}\left(\kappa_m \Sigma + \Sigma M_m M_m^\top + M_m M_m^\top \Sigma\right) B_{mn}^\top$. Then, $\Sigma_{A_{mn}}^\infty := \lim_{t\to\infty} \Sigma_{A_{mn}}^t$ exists, is unique, and is given by, $\Sigma_{A_{mn}}^\infty = \sum_{k=0}^\infty \mathcal{L}_n^k\left(Q_{mn}(\Sigma_{\theta_m}^\infty)\right)$*

**Theorem 7.5 (Convergence of Client Model's Uncertainty).** *Let $\rho(H_m) < 1$. Write $\mathcal{M}_m(\Sigma) = H_m \Sigma H_m^\top$ and $R_m(\Sigma) = G_m\left(\kappa_m \Sigma + \Sigma M_m M_m^\top + M_m M_m^\top \Sigma\right)G_m^\top$ Then the steady-state $\Sigma_{\theta_m}^\infty := \lim_{t\to\infty} \Sigma_{\theta_m}^t$ is the unique solution to $\Sigma_{\theta_m}^\infty = \mathcal{M}_m\left(\Sigma_{\theta_m}^\infty\right) + R_m\left(\Sigma_{\theta_m}^\infty\right) + P_m \Sigma_{A_{mn}}^\infty P_m^\top$.*

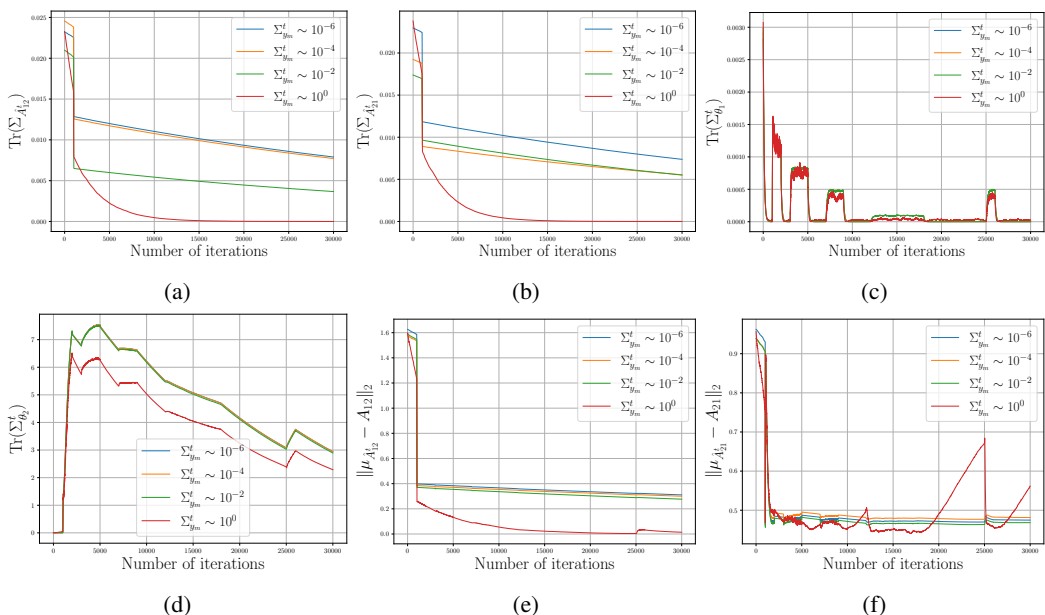

(a)          (b)          (c)

(d)          (e)          (f)

Figure 1: Uncertainty propagation during training for different levels of $\Sigma_{y_m}^t$ highting (a) $\mathrm{Tr}(\Sigma_{\hat{A}_{12}^t})$, (b) $\mathrm{Tr}(\Sigma_{\hat{A}_{21}^t})$, (c) $\mathrm{Tr}(\Sigma_{\theta_1}^t)$, (d) $\mathrm{Tr}(\Sigma_{\theta_2}^t)$, (e) $\|\mu_{\hat{A}_{12}^t} - A_{12}\|_2$, (f) $\|\mu_{\hat{A}_{21}^t} - A_{21}\|_2$ vs iterations.

Theorem 7.4 shows that the steady state uncertainty of the server parameter represented by $\Sigma_{\hat{A}_{mn}}^\infty$ depends only on the distribution of client data $(\mu_{y_m}, \Sigma_{y_m})$, and the steady state client model's variance $\Sigma_{\theta_m}^\infty$. It is **independent of the prior variance** $\Sigma_{\hat{A}_{mn}}^0$. Similarly, Theorem 7.5 establishes that the steady-state variance of the client model parameter $\Sigma_{\theta_m}^\infty$ is uniquely determined by the client data distribution $(\mu_{y_m}, \Sigma_{y_m})$, and the converged server uncertainty $\Sigma_{\hat{A}_{mn}}^\infty$. Crucially, this result confirms that the client's epistemic uncertainty is governed entirely by aleatoric quantities and training dynamics, and is **independent of the initial variance** $\Sigma_{\theta_m}^0$.

# 8 EXPERIMENTS

## 8.1 SYNTHETIC DATASET

**Experimental Settings.** We simulate a multi-client linear time-invariant (LTI) state space model described in 3. To enable interpretability, the experiments (except scalability) rely on a two-client setup with $p_m = 2$ and $d_m = 8$ for $m = \{1, 2\}$ such that the off-diagonal blocks of the state matrix $(A)$ are $A_{12} = 0$ and $A_{21} \neq 0$. Both client and server models are regularized to ensure feasible solutions. Further experimental details are provided in Section A of the Appendix.

**Aleatoric.** To analyze the effects of aleatoric noise, we change the data variance $\Sigma_{y_m}$ and observe the uncertainty of the server and client models at each iteration. The trace of covariance of the server parameter $(\hat{A}_{12}^t, \hat{A}_{21}^t)$ and the corresponding parameters $(\theta_1^t, \theta_2^t)$ at clients 1 and 2, respectively are plotted for different $\Sigma_{y_m}$ in Figures 1(a)-(f). The evolution plots for the covariance of both server and client models validate the claims (in Section 6, and Theorems 7.4 & 7.5).

The jumps in the trace covariance plots of $(\theta_1^t, \theta_2^t)$ refer to the points where the mean-shifts of data (i.e., $\mu_{y_m}$) occur. A preliminary inspection of Figures 1(a)-(e) shows that higher $\Sigma_{y_m}$ accelerates variance decay and hence faster learning. However, we can see from the norm error plots in Figure1(f) that very high $\Sigma_{y_m}$ causes larger estimation error in learning, and this effect is more profound during mean shifts. These errors stem from violating Assumption (A4) (Section 5) in mean-shifts.

**Epistemic.** The effect of epistemic uncertainty is demonstrated by sampling $\hat{A}_{mn}^0$ and $\theta_m^0$ from normal distributions while progressively increasing their variance. This allows us to observe how uncertainty propagates through the server and client models at each iteration. Figure 2 and Figure 3

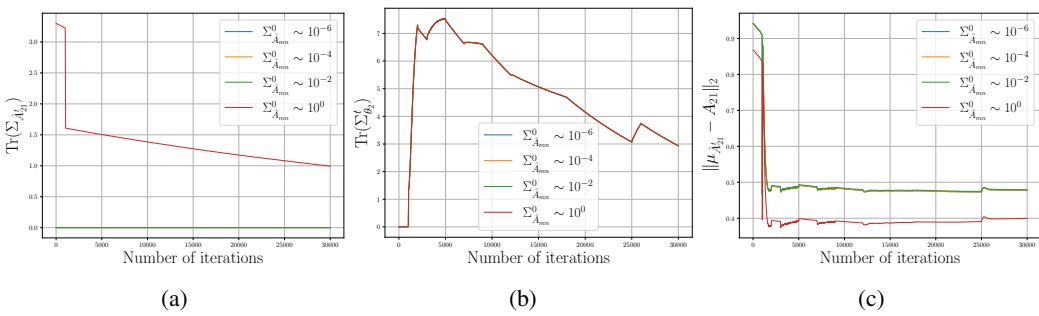

(a)    (b)    (c)

Figure 2: Uncertainty propagation during training for different levels of $\Sigma_{\hat{A}^0_{mn}}$ highlighting (a) $\text{Tr}(\Sigma_{\hat{A}^t_{21}})$, (b) $\text{Tr}(\Sigma^t_{\theta_2})$, (c) $\|\mu_{\hat{A}^t_{21}} - A_{21}\|_2$ vs iterations.

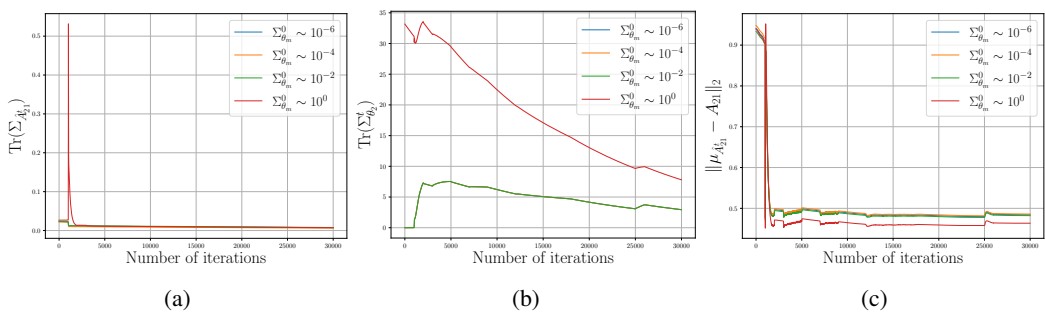

(a)    (b)    (c)

Figure 3: Uncertainty propagation during training for different levels of $\Sigma^0_{\theta_m}$ highlighting (a) $\text{Tr}(\Sigma_{\hat{A}^t_{21}})$, (b) $\text{Tr}(\Sigma^t_{\theta_2})$, (c) $\|\mu_{\hat{A}^t_{21}} - A_{21}\|_2$ vs iterations.

shows that the evolution of the covariances for server and client models is agnostic to the variance of the prior normal distribution, further validating the claim of Theorems 7.4 and 7.5.

Please refer to Appendix B for additional results highlighting the evolution of the **(a)** cross-covariance terms, **(b)** communicated terms (mentioned in Section 6), and **(c)** Scalability studies.

## 8.2 REAL-WORLD DATASET

We perform real-world experiments on two industrial cybersecurity datasets: **(1)** HAI Shin et al. (2021) and **(2)** SWaT Mathur & Tippenhauer (2016), each containing multiple interacting processes. We fit a linear state space model separately to each process. Specifically, we use the subspace identification method Overschee & Moor (1994) to identify the process-specific state-transition and observation matrices, which are further used for analyzing theoretical results of Section 6 & 7. Preprocessing details are provided in Appendix C.

**Results.** Cross-process GC in both datasets is learned using the FedGC framework. Owing to space constraints, we provide the uncertainty evolution plots and the convergence results in Appendix C.

## 9 LIMITATIONS

The analysis is based on the FedGC framework, which can be restrictive due to its linear models. We provide extensions to non-linear models in Appendix E. The uncertainty propagation relies on data stationarity, and we discuss possible relaxations in Appendix E.2, though a full treatment remains open. Theoretical guarantees scale unfavorably with dimensionality, which limits applicability to very large systems; we complement this by analyzing both communication and computation complexity in Appendix D.5. Finally, client–server cross-covariance terms may expose sensitive data; we present preliminary differential privacy modeling in Appendix D.4.

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

# Appendices

## CONTENTS

**Note.** All the plots corresponding to Section B, and C are present at the end of this document.

## A    THE USE OF LARGE LANGUAGE MODELS (LLMS)

In preparing this work, large language models (LLMs), specifically ChatGPT, were used only as a writing aid to polish the clarity, flow, and readability of the manuscript. LLMs were not involved in generating or shaping the underlying theoretical ideas, arguments, or results. All core contributions, analyses, and conclusions presented here are entirely original and solely attributable to the authors.

## B    ADDITIONAL RESULTS ON ON SYNTHETIC DATASET

### B.1    EXPERIMENTAL DETAILS

Experiments on synthetic dataset was conducted by simulating a two-client LTI system. Each client has hidden latent state dimension $p_m = 2$ and measurement dimension $d_m = 8$. A one way dependency in which client 1 granger causes client 2 (and not vice-versa) is considered. Therefore $A_{12} = 0$ and $A_{21} \neq 0$. The loss functions in (5) and (8) are regularized and the learning rates $\gamma$, $\eta_1$ and $\eta_2$ are adjusted to ensure convergence but not optimally tuned.

All the results reported are in terms of the trace of the covariance matrices. Trace of the covariance matrix was chosen as a scalar quantity to quantify the uncertainty from the covariance matrices for a multitude of reasons such as: 1) it is the sum of variances across all directions, 2) it is rotation invariant, 3) computationally cheap etc. Unless otherwise specified, the quantities plotted are relevant to explaining the uncertainty propagation in learning $\hat{A}_{21}$.

As discussed in Section 8, the effect of aleatoric noise in all the experiments is studied by changing the data variance $\Sigma_{y_m}^t$ and the effect of epistemic noise is studied by changing the variance of the initial values $\Sigma_{\hat{A}_{mn}}^0$ and $\Sigma_{\theta_m}^t$.

#### B.1.1    CROSS-COVARIANCE AND COMMUNICATED TERMS

**Cross-Covariances.** The evolution of cross-covariance terms $\Lambda_m^t$, $\Gamma_{mn}^t$ and $\Psi_{mn}^t$ discussed in Section 6 for different regimes of aleatoric and epistemic noise are given in Figures 5, 6 and 7. We can observe that the cross-covariance terms converge even for very high noise regimes.

**During Communication.** The augmented client state $\hat{h}_{m,a}^t$ sent from the client carries the uncertainty from the client to the server and the gradient $\nabla_{\hat{h}_{m,a}^t} L_s^t$ sent back to the client carries the uncertainty from the server to the client. The evolution of covariances of these communicated terms for different regimes of aleatoric and epistemic noises are plotted in Figures 8, 9 and 10.

### B.2    SCALABILITY STUDIES

In the scalability experiment, we consider the two-client system with similar causal relationship as in previous experiments. The hidden state dimension $p_m = 2$ is kept constant while the measurement dimension $d_m$ is increased for both clients. Trace of covariance of $\hat{A}_{21}$ at convergence for regimes of $\Sigma_{\hat{A}_{mn}}^0 \sim \{10^{-6}, 10^{-4}, 10^{-2}, 10^0\}$ is summarized in Table 2. We could observe that the performance of the framework remains more or less similar with increased dimension of the measurements $d_m$.

Table 2: Trace(Cov($\hat{A}_{21}$)) vs measurement (i.e., raw data) dim. $d_m$

| Order of Variance | Measurement (Raw Data) Dimension ($d_m$) | | | |
| --- | --- | --- | --- | --- |
| | $d_m = 16$ | $d_m = 32$ | $d_m = 64$ | $d_m = 128$ |
| $\sim 10^{-6}$ | $\approx 10^{-9}$ | $\approx 10^{-5}$ | $\approx 10^{-7}$ | $\approx 10^{-7}$ |
| $\sim 10^{-4}$ | $\approx 10^{-8}$ | $\approx 10^{-5}$ | $\approx 10^{-7}$ | $\approx 10^{-7}$ |
| $\sim 10^{-2}$ | $1.0 \times 10^{-4}$ | $4.0 \times 10^{-4}$ | $1.0 \times 10^{-4}$ | $\approx 10^{-5}$ |
| $\sim 10^0$ | 1.6722 | 1.9733 | 2.4601 | 1.5882 |

In the second experiment, we keep $p_m = 2$ and $d_m = 8$ and vary the number of clients $M$. We report the trace of covariance of $\hat{A}_{21}$ at convergence for different regimes of $\Sigma^0_{\hat{A}_{mn}}$ in Table 3.

Table 3: Trace($\text{Cov}(\hat{A}_{21})$) vs. no. of clients $M$

| Order of Variance | Number of Clients ($M$) | | | |
|---|---|---|---|---|
| | $M = 2$ | $M = 4$ | $M = 8$ | $M = 16$ |
| $\sim 10^{-6}$ | $\approx 10^{-5}$ | $\approx 10^{-5}$ | $\approx 10^{-5}$ | $\approx 10^{-6}$ |
| $\sim 10^{-4}$ | $\approx 10^{-5}$ | $\approx 10^{-5}$ | $\approx 10^{-5}$ | $\approx 10^{-6}$ |
| $\sim 10^{-2}$ | 0.0001 | 0.0002 | 0.0002 | 0 |
| $\sim 10^{0}$ | 1.7233 | 1.6456 | 2.5835 | 4.7 |

## B.3 COMPARISON TO BASELINES

To contextualize our uncertainty–propagation framework, we compare our method against several natural baselines for vertically federated Granger-causal learning. Table 4 reports two metrics: **(i)** the Frobenius error $\|\hat{A}_{21} - A_{21,\text{true}}\|_F$ measuring estimation accuracy, and **(ii)** the trace $\text{tr}(\Sigma_{A_{21}})$ of the propagated covariance, quantifying parameter uncertainty.

- **Centralized.** All client data are pooled and $A_{21}$ is estimated in a non-federated, oracle fashion. This constitutes the ideal upper bound on estimation accuracy because no privacy or communication constraints are present.

- **Our Method** Our method computes empirical uncertainty by running FedGC from *multiple independently sampled initializations* (multi-start ensemble). Each run follows the standard FedGC alternating client–server updates. The empirical mean and covariance of the resulting estimates are reported.

- **Independent Clients.** Each client estimates its own local block without modeling cross-client dependencies (equivalent to setting $\eta_2 = 0$ in Eq. (5)). This baseline tests the effect of removing inter-client coupling altogether.

Table 4 summarizes the comparison. As expected, Centralized estimation achieves the lowest error and nearly zero uncertainty since it has full access to all data. Our method achieves the best performance among federated approaches: it produces the lowest Frobenius error and the smallest covariance trace, indicating both accurate and stable estimation of $A_{21}$. In contrast, Independent Clients baseline underperform: removing iterative coupling leads to a higher estimation error and significantly larger propagated uncertainty.

Table 4: Comparison across baselines.

| Method | $\|\hat{A}_{21} - A_{21,\text{true}}\|_F$ | $\text{tr}(\Sigma_{A_{21}})$ |
|---|---|---|
| Centralized | $4.804058 \times 10^{-1}$ | $3.960954 \times 10^{-14}$ |
| **Our** | $4.96649 \times 10^{-1}$ | $7.96647 \times 10^{-9}$ |
| Independent Clients | $5.02350 \times 10^{-1}$ | $1.87971 \times 10^{-8}$ |

## B.4 EFFECT OF DP NOISE ON UNCERTAINTY AND CAUSAL ACCURACY

In this subsection, we study how differentially private (DP) Gaussian noise injected into the federated messages affects **(i)** steady-state parameter uncertainty and **(ii)** accuracy of causal link detection. We consider two communication directions: *client→server*, where noise is added to the states sent by the clients to the server, and *server→client*, where noise is added to the gradients sent back to the clients. In both cases, we use the Gaussian mechanism and sweep the noise standard deviation over a log-scale grid $\sigma \in \{10^{-6}, 10^{-5}, \ldots, 10^{-1}\}$.

Figure 4 summarizes the results on the synthetic dataset. The top row shows the steady-state uncertainty of the cross-client Granger block $A_{21}$, measured via $\text{tr}(\Sigma_{A_{21}})$, as a function of the in-

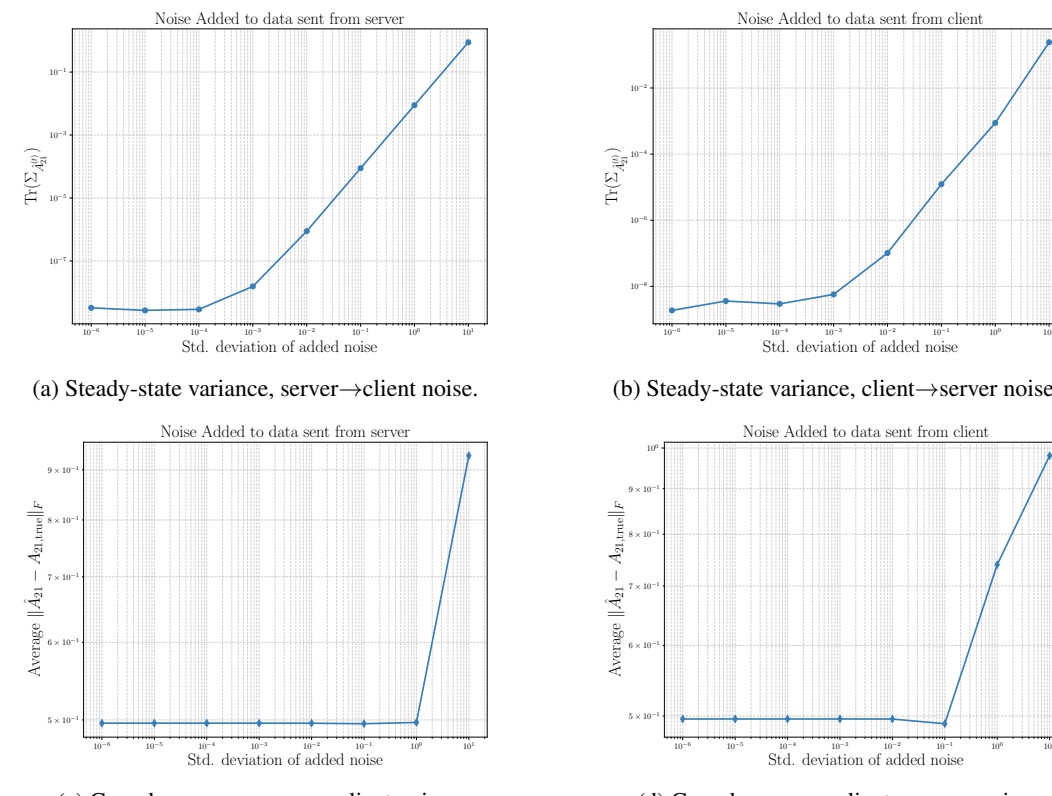

(a) Steady-state variance, server→client noise.

(b) Steady-state variance, client→server noise.

(c) Causal accuracy, server→client noise.

(d) Causal accuracy, client→server noise.

Figure 4: Effect of DP Gaussian noise on steady-state uncertainty and causal link detection in the synthetic dataset. Top row: trace of the propagated covariance $\mathrm{tr}(\Sigma_{A_{21}})$ as a function of the noise scale $\sigma$. Bottom row: Frobenius error $\|\hat{A}_{21} - A_{21,\mathrm{true}}\|_F$ (normalized) versus $\sigma$.

jected noise scale $\sigma$. The bottom row reports the corresponding causal estimation accuracy, measured by the Frobenius norm $\|\hat{A}_{21} - A_{21,\mathrm{true}}\|_F$ and normalized for visualization. Panels labelled "server→client" and "client→server" correspond to the direction in which DP noise is added. Across all configurations, the uncertainty recursions remain numerically stable for the entire range of $\sigma$, and the steady-state variance increases smoothly as the DP noise level grows. For small to moderate noise ($\sigma \lesssim 10^{-3}$), the inflation in $\mathrm{tr}(\Sigma_{A_{21}})$ is modest and the causal estimation error remains close to the non-DP baseline in both communication directions. Only for the largest noise levels ($\sigma \approx 10^{-2}$–$10^{-1}$) do we observe a pronounced increase in variance accompanied by a degradation in causal link detection, as expected from strong DP perturbations. These experiments confirm that our uncertainty propagation framework is robust to reasonably strong DP noise and that the empirical privacy–utility trade-off behaves consistently with the Gaussian mechanism in both communication channels.

## C  RESULTS ON REAL-WORLD DATASET

### C.1  EXPERIMENTAL SETTING

We utilize two industrial cybersecurity datasets – **(1)** HAI and **(2)** SWaT, to conduct real-world experiments.

**HAI.** The HAI dataset captures SCADA (Supervisory Control and Data Acquisition) time-series data from a realistic industrial control system testbed augmented with a Hardware-in-the-Loop (HIL) simulator that emulates steam-turbine power generation and pumped-storage hydropower generation. HAI testbed comprises four processes: 1. Boiler Process (P1), 2. Turbine Process (P2), 3. Water Treatment (P3), 4. HIL Simulator (P4). Processes P1–P3 each represent a physical subsys-

tem with multiple sensor measurements and are treated as individual clients in our study. On the other hand, P4 is a HIL simulation engine exposing control signals rather than sensor outputs and is therefore excluded from our analysis.

**SWaT.** SWaT is the data from an industry-compliant six-stage water treatment testbed commissioned in March 2015 by iTrust. Different stages in water treatment includes raw water storage, pH balancing, ultrafiltration, dechlorination, reverse osmosis, and membrane backwash. Each stage is treated as a client in our setting. We do not include the last stage (membrane backwash) in our study because all the measurements in this stage are constant except one that takes few more values and inclusion of this stage was causing convergence issues.

### C.2    PREPROCESSING

We utilize different preprocessing methodologies for the two real-world datasets.

**HAI.** We first normalized the measurements to make sure they are evenly scaled. A subspace identification algorithm was used to fit an LTI system to these normalized measurements. The subspace identification outputs the state-transition matrix $A$ (which also signifies the Granger causality), and the observation matrix $C$. The size of low-dimensional states $p$ for the entire system was obtained by looking at the decay of singular values of Hankel matrix involved in the subspace identification step.

The observation matrix $C$ obtained from subspace identification is not block diagonal. An $L_2$ norm based thresholding was used to assign each state variable to a process and build a block diagonal $C$ matrix. This is followed by re-estimating $A$ matrix using least squares with the block diagonal $C$ matrix. This re-estimated $A$ matrix is considered the ground truth Granger causality for our study.

**SWaT.** For each client, the measurement variables with pearson correlation greater 0.3 with other clients are selected. This is followed by the same steps as in the case of HAI data.

### C.3    RESULTS

We plot the evolution of covariance of all the off-diagonal blocks $\hat{A}_{mn}^t$, covariance of $\theta_m^t$ for all clients and average $L_2$ norm error of each $\hat{A}_{mn}^t$ for both datasets in Figures 17-28.

**HAI.** The server-side uncertainty on all off-diagonal $\hat{A}_{mn}$ terms drops rapidly within the first few hundred iterations and stabilizes at very low traces (Figure 17), indicating fast convergence even under high data (measurement) noise. Local parameter variances $\Sigma_{\theta_m}$ likewise shrink to nearly zero (Figure 6), with only minor differences between the three processes. The average estimation error $\|\mu_{A_{mn}} - A_{mn}\|$ decreases monotonically on the log–iteration scale (Figure 7), confirming that reduced uncertainty directly translates to higher causal-link accuracy.

**SWaT.** In contrast to HAI dataset, SWaT's causal-block covariances decline more gradually over $\sim 1,000$ iterations and do not show signs of complete convergence (Figure 25). This suggests that the six-stage water-treatment testbed requires more iterations or additional data to fully resolve the underlying causal links. Local model variances $\Sigma_{\theta_m}$ exhibit a wider spread (at the end of 1000 iterations), with some retaining significantly more uncertainty than others (Figure 27). Correspondingly, the $\ell_2$ estimation errors plateau earlier and remain higher (Figure 28), underscoring the slower, higher-variance convergence behavior in SWaT, possibly due to higher noise in the collected data.

## D    ADDITIONAL THEORETICAL RESULTS

### D.1    LEMMA D.1

**Lemma D.1.** *At any time $t$ and, for each client $m$, define*

$$g_m = \nabla_{\theta_m} L_s = X_m (y_m^{t-1})^\top, \qquad X_m := A_{mm}^\top \Big( A_{mm} \, \Delta h_m - \sum_{r \neq m} \hat{A}_{mr} (\hat{h}_r^{t-1})_c \Big).$$

*Under Assumptions (A1)-(A4), we have for any $m \neq n$,*

$$\mathrm{Cov}\big(\mathrm{vec}(g_m), \mathrm{vec}(g_n)\big) = 0.$$

*Proof.* By Assumption **(A2)**, the block-row server parameters for clients $m$ and $n$ are mutually independent, hence $X_m \perp X_n$. By Assumptions **(A1)**, **(A4)**, and **(A5)**, the local data satisfy

$$\mu_{y_m} = \mathbb{E}[y_m^{t-1}], \qquad \Sigma_{y_m} = \mathrm{Cov}(y_m^{t-1}, y_m^{t-1}),$$

with $\mathrm{Cov}(y_m^{t-1}, y_n^{t-1}) = 0$ for $m \neq n$. Since $y_m^{t-1}$ is also independent of $\{A_{pq}^t\}$ by Assumption **(A3)**, we may write

$$\mathrm{vec}(g_m) = (I \otimes X_m) \, y_m^{t-1}.$$

Thus, for $m \neq n$,

$$\mathrm{Cov}\big(\mathrm{vec}(g_m), \mathrm{vec}(g_n)\big) = \mathbb{E}\big[(I \otimes X_m)\, y_m^{t-1} (y_n^{t-1})^\top (I \otimes X_n)^\top\big] - \mathbb{E}[\mathrm{vec}(g_m)]\,\mathbb{E}[\mathrm{vec}(g_n)]^\top.$$

Independence allows factorization, and Assumption **(A4)**–**(A5)** implies $\mathbb{E}[y_m^{t-1}(y_n^{t-1})^\top] = \mu_{y_m}\mu_{y_n}^\top$, which cancels with the product of means. Hence

$$\mathrm{Cov}\big(\mathrm{vec}(g_m), \mathrm{vec}(g_n)\big) = 0, \quad m \neq n.$$

$\square$

## D.2 PROPOSITION D.2

**Proposition D.2.** *Consider the client update rule*

$$\theta_m^{t+1} = \theta_m^t - \eta_1 g_{m,a}^t - \eta_2 g_{m,s}^t,$$

*where $g_{m,a}^t = \nabla_{\theta_m} L_m$ (local) and $g_{m,s}^t = \nabla_{\theta_m} L_s$ (global). Suppose Assumptions (A1)–(A5) hold. Then, for any two distinct clients $m \neq n$,*

$$\lim_{t \to \infty} \mathrm{Cov}(\theta_m^t, \theta_n^t) = 0.$$

*Proof.* Expanding the covariance for $m \neq n$ gives

$$\mathrm{Cov}(\theta_m^{t+1}, \theta_n^{t+1}) = \mathrm{Cov}(\theta_m^t, \theta_n^t) + \eta_2^2 \, \mathrm{Cov}(g_{m,s}^t, g_{n,s}^t) + \text{vanishing local terms}$$

where the omitted terms involve cross-products between local and global gradients.

By Lemma D.1, $\mathrm{Cov}(g_{m,s}^t, g_{n,s}^t) = 0$ for $m \neq n$. By **(A2)** and **(A3)**, $\theta_m^t \perp\!\!\!\perp \theta_n^t$ at initialization. By **(A4)**–**(A5)**, client data are uncorrelated across $m, n$, and all noise is additive. Hence all cross-terms vanish, and we obtain the recursion

$$\mathrm{Cov}(\theta_m^{t+1}, \theta_n^{t+1}) = \mathrm{Cov}(\theta_m^t, \theta_n^t).$$

Since the initial covariance is zero, induction gives $\mathrm{Cov}(\theta_m^t, \theta_n^t) = 0$ for all $t$, and thus

$$\lim_{t \to \infty} \mathrm{Cov}(\theta_m^t, \theta_n^t) = 0.$$

$\square$

## D.3 PROPOSITION D.3

As a direct consequence of Assumption **(A5)** we have the following proposition:

**Proposition D.3** (**Data Uncertainty with Zero Process Noise**)**.** *Consider the discrete LTI system*

$$x^{t+1} = A\,x^t + w^t, \ \ y^t = C\,x^t + v^t, \ \ w^t \sim \mathcal{N}(0, Q), \ and \ v^t \sim \mathcal{N}(0, R)$$

*and assume the state matrix is asymptotically stable, $\rho(A) < 1$. In the absence of process noise i.e., $Q = 0$, then the data variance converges to its noise covariance. $\lim_{t\to\infty} \mathrm{Var}(y^t) = R$.*

*Proof.* With $Q = 0$, the state update becomes deterministic:

$$x^{t+1} = Ax^t.$$

Thus, assuming some initial condition $x^0$, the state at time $t$ is given by,

$$x^t = A^t x^0.$$

Therefore, the data $y^t$ is given by,

$$y^t = Cx^t + v^t = CA^t x^0 + v^t.$$

To compute the variance of $y^t$, we consider the following equation,

$$\text{Var}(y^t) = \text{Var}(CA^t x^0 + v^t).$$

Since $v^t \sim \mathcal{N}(0, R)$ is independent of $x^0$, and $CA^t x^0$ is deterministic, we write $\text{Var}(y^t)$ as,

$$\text{Var}(y^t) = \text{Var}(CA^t x^0) + \text{Var}(v^t).$$

Now, since $CA^t x^0$ is deterministic, its variance is given by,

$$\text{Var}(CA^t x^0) = CA^t \text{Var}(x^0)(A^t)^\top C^\top.$$

Given that $\rho(A) < 1$, we have $A^t \to 0$ as $t \to \infty$ we have,

$$CA^t \text{Var}(x^0)(A^t)^\top C^\top \to 0.$$

Using the above result, and the fact that $\text{Var}(v^t) = R$ we can write $\text{Var}(y^t)$ as,

$$\lim_{t \to \infty} \text{Var}(y^t) = \lim_{t \to \infty} \left[ CA^t \text{Var}(x^0)(A^t)^\top C^\top + R \right] = R.$$

$\square$

### D.4 PRIVACY ANALYSIS

We formally prove that the cross-covariance terms between client quantities and server quantities can be made differentially private under the Gaussian mechanism.

**FedGC Latent Privacy.** In the original FedGC setup, each client $m$ communicates a compressed latent state $\hat{h}_{c,m}^t$ and an augmented latent state $\hat{h}_{a,m}^t$, both of which are computed from private data $y_m^t$ via local encoders (KF). As shown in Appendix F of FedGC, if the mapping $y_m^t \mapsto \hat{h}_{c,m}^t$ has bounded $\ell_2$-sensitivity $\Delta$, then adding Gaussian noise:

$$\tilde{h}_{c,m}^t = \hat{h}_{c,m}^t + \mathcal{N}(0, \sigma^2 I), \quad \sigma \geq \frac{\Delta \cdot \sqrt{2 \log(1.25/\delta)}}{\varepsilon}$$

ensures $(\varepsilon, \delta)$-differential privacy for each client's latent state. The same construction holds for $\hat{h}_{a,m}^t$ and is preserved in our framework.

**DP for Cross-Covariances.** Unlike FedGC, our method introduces server-side use of *cross-covariance* matrices between client representations, which may leak client-private correlations. The key objects are:

$$\Gamma_{mn}^t := \text{Cov}(\hat{h}_{c,m}^t, \hat{h}_{a,n}^t) = \mathbb{E}[\hat{h}_{c,m}^t (\hat{h}_{a,n}^t)^\top] - \bar{h}_{c,m}^t (\bar{h}_{a,n}^t)^\top$$

$$\Psi_{mn}^t := \text{Cov}(v_m^t, \hat{h}_{c,n}^t) = \mathbb{E}[v_m^t (\hat{h}_{c,n}^t)^\top] - \bar{v}_m^t (\bar{h}_{c,n}^t)^\top$$

where $v_m^t = \text{vec}(\theta_m^t)$ is the flattened client model parameter. We now show how to make these matrices differentially private via Gaussian noise.

Each client $m$ transmits perturbed values of $\hat{h}_{c,m}^t, \hat{h}_{a,m}^t$

$$\tilde{h}_{c,m}^t = \hat{h}_{c,m}^t + \xi_c^t, \quad \tilde{h}_{a,n}^t = \hat{h}_{a,n}^t + \xi_a^t$$

with $\xi_c^t, \xi_a^t \sim \mathcal{N}(0, \sigma_h^2 I)$. Furthermore, since cross-covariance $\Psi_{mn}$ requires $v_m$, we peturb it as:

$$\tilde{v}_m^t = v_m^t + \xi_v^t$$

with $\xi_v^t \sim \mathcal{N}(0, \sigma_v^2 I)$. The server then computes the cross-covariances:

$$\tilde{\Gamma}_{mn}^t := \text{Cov}(\tilde{h}_{c,m}^t, \tilde{h}_{a,n}^t), \quad \tilde{\Psi}_{mn}^t := \text{Cov}(\tilde{v}_m^t, \tilde{h}_{c,n}^t)$$

These estimators contain the desired cross-covariance along with stochastic masking from the perturbations.

Using classical state-space theory & Lipschitz assumptions, we have norm bounds: $\|\hat{h}_{c,m}^t\|_2 \leq B_c$, $\|\hat{h}_{a,n}^t\|_2 \leq B_a$, $\|v_m^t\|_2 \leq B_v$. Then, the $\ell_2$-sensitivities of $\Gamma_{mn}$, and $\Psi_{mn}$ are:

$$\Delta_2(\Gamma_{mn}^t) \leq 2B_cB_a, \quad \Delta_2(\Psi_{mn}^t) \leq 2B_vB_c$$

To achieve $(\varepsilon, \delta)$-DP via the Gaussian mechanism, it suffices to add i.i.d. noise to each element of the matrices:

$$\sigma_\Gamma \geq \frac{2B_cB_a\sqrt{2\log(1.25/\delta)}}{\varepsilon}, \quad \sigma_\Psi \geq \frac{2B_vB_c\sqrt{2\log(1.25/\delta)}}{\varepsilon}$$

This guarantees that the server-observed $\tilde{\Gamma}_{mn}^t$ and $\tilde{\Psi}_{mn}^t$ are differentially private with respect to any one client's data at time $t$.

**Implications on Uncertainty Propagation.** The injection of DP noise into $\Gamma_{mn}^t$ and $\Psi_{mn}^t$ affects the downstream uncertainty estimates in both client- and server-side propagation. Specifically, $\Gamma_{mn}^t$ appears exclusively in the server-side recursion (Theorem 6.6), while both $\Gamma_{mn}^t$ and $\Psi_{mn}^t$ are used in the client-side update (Theorem 6.8). Making these matrices differentially private means that the propagated covariances become slightly biased or inflated due to the added noise. This leads to an overestimation of uncertainty, which preserves the structure of the recursion but may reduce the accuracy of Granger causality. Nonetheless, the estimates remain valid under perturbed inputs, and the user can control the privacy-utility trade-off via the $(\varepsilon, \delta)$ parameters.

### D.5  COMPLEXITY ANALYSIS

Let each client $m$ have state dimension $p_m$ and data dimension $d_m$, with $M$ clients in total. We first provide the computation and communication complexity per client.

**(1) Computation.** In a naive implementation, updating the full parameter covariance matrix $\Sigma_\theta^t \in \mathbb{R}^{p_md_m \times p_md_m}$ would incur $\mathcal{O}(p_m^2d_m^2)$ computation per round due to matrix-matrix multiplications and Kronecker product evaluations. However, our implementation can avoid this cost by exploiting the structure of low-rank quantities such as $y^ty^{t^\top}$ (which is rank-1), when $d_m \gg p_m$. As a result, matrix-vector products involving these terms can be computed without materializing the full $p_md_m \times p_md_m$ matrices. If we adopt a block-diagonal or factored representation of $\Sigma_\theta^t$ across layers or time steps, the per-client computational cost is further reduced to $\mathcal{O}(p_md_m^2)$ per iteration.

**(2) Communication.** In terms of communication, our framework introduces no additional overhead compared to vanilla FedGC. Therefore, total communication cost per client remains $\mathcal{O}(p_m)$, matching that of FedGC. No additional communication is needed for tracking uncertainty.

**All $M$ clients.** Across all $M$ clients, the total additional computation scales as $\mathcal{O}\left(\sum_{m=1}^M p_md_m^2\right)$ per round, while communication remains unchanged at $\mathcal{O}\left(\sum_{m=1}^M p_m\right)$. These properties make the method scalable to realistic federated settings.

## E  EXTENSIONS TO THE FRAMEWORK

### E.1  NON-LINEAR MODELS

Our current theory is derived under the assumption of linear time-invariant dynamics and Gaussian noise, consistent with the original FedGC model. Nonetheless, we emphasize that the structure of our uncertainty propagation is not inherently tied to linearity; it generalizes to any dynamical model with well-defined latent state compression and state transition.

**Bayesian Models.** We consider an extension to *nonlinear but stationary* dynamical systems. Non-linear dynamics being highly model-specific, we first show that our theoretical structure carries over to two settings: (1) Extended Kalman Filters (EKF) - non-linear extension of the linear KF based formulation, and (2) Gaussian Processes (GPs) - for kernelized settings. Both EKF, and GP-based

frameworks support general nonlinear transition functions while admitting tractable recursive expressions analogous to Theorem 6.8.

**(1) Extended Kalman Filter (EKF).** EKF approximates nonlinear state transitions via first-order Taylor expansions. Consider the system:

$$h_c^t = f(h_c^{t-1}) + w^t, \quad y^t = g(h_c^t) + v^t, \quad w^t \sim \mathcal{N}(0, Q), \; v^t \sim \mathcal{N}(0, R),$$

where $f : \mathbb{R}^p \to \mathbb{R}^p$, $g : \mathbb{R}^p \to \mathbb{R}^d$, with $d \gg p$, are smooth nonlinear functions. The prediction step linearizes $f$ around the filtered mean:

$$\hat{h}_c^{t|t-1} = f(\hat{h}_c^{t-1}), \quad F^{t-1} := \left.\frac{\partial f}{\partial h}\right|_{h=\hat{h}_c^{t-1}}, \quad P^{t|t-1} = F^{t-1} P^{t-1} {F^{t-1}}^\top + Q.$$

For the observation model, define $J_g := \left.\frac{\partial g}{\partial h}\right|_{h=\hat{h}_c^{t|t-1}}$. The update step proceeds via:

$$K^t = P^{t|t-1} J_g^\top (J_g P^{t|t-1} J_g^\top + R)^{-1}, \quad \hat{h}_c^t = \hat{h}_c^{t|t-1} + K^t(y^t - g(\hat{h}_c^{t|t-1})), \quad P^t = (I - K^t J_g) P^{t|t-1}.$$

We define the augmented representation used in FedGC:

$$\hat{h}_a^t = \hat{h}_c^t + \theta^t y^t, \quad \theta^t \in \mathbb{R}^{p \times d}, \quad v^t := \mathrm{vec}(\theta^t).$$

Following our update rule, $v^{t+1} = v^t - \eta_1 \nabla_{v^t} \mathcal{L}_t^{\mathrm{local}} - \eta_2 \nabla_{v^t} \mathcal{L}_t^{\mathrm{server}}$, the parameter covariance update becomes:

$$\Sigma_\theta^{t+1} = H^t \Sigma_\theta^t {H^t}^\top + G^t P^t {G^t}^\top + H^t \Lambda^t {G^t}^\top + G^t {\Lambda^t}^\top H^t + P^t \Sigma_A^t {P^t}^\top,$$

where,

$$
\begin{aligned}
H^t &= I_{pd} - 2\eta_1 (y^t {y^t}^\top) \otimes (J_g^\top J_g) - 2\eta_2 (y^t {y^t}^\top) \otimes (A_{mm}^\top A_{mm}), \\
G^t &= 2\eta_1 (y^t \otimes J_g^\top), \\
P^t &= \mathrm{Cov}(\hat{h}_c^t), \\
\Lambda^t &= \mathrm{Cov}(v^t, \hat{h}_a^t).
\end{aligned}
$$

This recursion is structurally identical to Theorem 6.8, with $C$ and $P$ replaced by the Jacobian $J_g$ and EKF posterior $P^t$, respectively.

**(2) Gaussian Process (GP).** Consider a GP-based transition model, where $f \sim \mathcal{GP}(0, k(\cdot, \cdot))$ governs the latent dynamics:

$$h_c^t = f(h_c^{t-1}) + w^t.$$

Conditioning on past data $\{h_c^i, h_c^{i-1}\}_{i=1}^n$, the GP posterior yields:

$$h_c^t \sim \mathcal{N}(\mu_t^f, \Sigma_t^f + Q), \quad \mu_t^f = k^\top(h_c^{t-1}) K^{-1} \mathbf{h}, \quad \Sigma_t^f = k(h_c^{t-1}, h_c^{t-1}) - k^\top(h_c^{t-1}) K^{-1} k(h_c^{t-1}),$$

where $K$ is the kernel matrix over $\{h_c^{i-1}\}$. We again define:

$$\hat{h}_a^t = \mu_t^f + \theta^t y^t.$$

The parameter covariance propagates as:

$$\Sigma_\theta^{t+1} = H^t \Sigma_\theta^t {H^t}^\top + G^t \Sigma_t^f {G^t}^\top + H^t \Lambda^t {G^t}^\top + G^t {\Lambda^t}^\top H^t + P^t \Sigma_A^t {P^t}^\top,$$

with $J_g := \partial g / \partial h \big|_{h=\mu_t^f}$ and other terms as above. This recursion mirrors the EKF case, with GP posterior variance $\Sigma_t^f$ replacing $P^t$, and confirms that Theorem 6.8 applies structurally to kernelized approximations.

**Non-Bayesian Models.** While EKF and GP admit analytic propagation of posterior uncertainty, general deep models (e.g., RNNs, transformers) do not support closed-form covariance updates. Techniques such as Monte Carlo dropout, ensemble variance, or variational inference yield scalar uncertainties but do not enable recursive tracking. Thus, incorporating these models into our federated uncertainty framework would require entirely new derivations and likely sacrifice the tractability of the current analysis.

E.2   RELAXING STATIONARY ASSUMPTIONS

Our main theoretical results are derived under the classical assumption that each client observes weakly stationary data with time-invariant first- and second-order moments. This assumption enables closed-form uncertainty propagation through the FedGC recursions. Here we clarify how the framework behaves when stationarity is mildly violated and how it can be adapted to non-stationary settings.

**Exponentially weighted moments.**   The key observation is that all uncertainty recursions remain valid when the empirical moments are replaced by *exponentially weighted* (EWMA) moments,

$$\mu_t = (1 - \lambda)\mu_{t-1} + \lambda x_t, \qquad \Sigma_t = (1 - \lambda)\Sigma_{t-1} + \lambda x_t x_t^\top,$$

for a forgetting factor $0 < \lambda < 1$. Suppose the underlying data possess slowly drifting moments $(\mu_t^*, \Sigma_t^*)$ with bounded temporal variation $\|\mu_t^* - \mu_{t-1}^*\| \leq \delta$ and $\|\Sigma_t^* - \Sigma_{t-1}^*\| \leq \delta$. Standard results for stochastic approximation imply

$$\|\mu_t - \mu_t^*\| = O(\delta/\lambda), \qquad \|\Sigma_t - \Sigma_t^*\| = O(\delta/\lambda).$$

Thus, EWMA moments track the true time-varying moments whenever the drift is slower than the forgetting rate. Replacing stationary moments by EWMA moments preserves the algebraic form of our uncertainty recursions; the same propagation equations apply with $(\mu_t, \Sigma_t)$ in place of fixed moments.

Beyond EWMA, other forms of non-stationarity can be incorporated by modifying the moment-estimation step. Examples include sliding-window estimators, seasonally adjusted or periodic- window estimators, trend-filtered or total-variation–regularized moment updates, and online convex-combination estimators for abrupt regime changes. Neural-network–based models can also accommodate complex non-stationarity, but doing so would require deriving a new set of uncertainty–propagation equations beyond the linear FedGC framework. A systematic treatment of these extensions is an important direction for future work.

# F   PROOFS

## F.1   PROPOSITION 6.1

• **Proposition** (**Client Model-Client Data Dependence**) *Assume* $\mathrm{Var}(y_m^{t-1}) > 0$. *Then under the federated Granger-causality updates,* $\Omega_m^t := \mathrm{Cov}(v_m^t, y_m^t) \neq 0$.

*Proof.* From the gradient-descent update we have,

$$\theta_m^t = \theta_m^{t-1} + 2\eta_1 (C_{mm}A_{mm})^\top \left(y_m^t - C_{mm}A_{mm}[\hat{h}_{m,c}^{t-1} + \theta_m^{t-1}y_m^{t-1}]\right) y_m^{t-1\top}$$
$$- 2\eta_2 A_{mm}^\top \left(A_{mm}[\hat{h}_{m,a}^{t-1} - \hat{h}_{m,c}^{t-1}] - \sum_{n \neq m} \hat{A}_{mn}^{t-1} \hat{h}_{n,c}^{t-1}\right) y_m^{t-1\top}.$$

Rearrange to isolate dependence on $y_m^{t-1}$:

$$\theta_m^t = M + B\, y_m^{t-1\top},$$

where

$$M = \theta_m^{t-1} + 2\eta_1 (C_{mm}A_{mm})^\top \left(y_m^t - C_{mm}A_{mm}\, \hat{h}_{m,c}^{t-1}\right) y_m^{t-1\top} - 2\eta_2 A_{mm}^\top \left(A_{mm}[\hat{h}_{m,a}^{t-1} - \hat{h}_{m,c}^{t-1}]\right) y_m^{t-1\top},$$

$$B = -2\eta_1 (C_{mm}A_{mm})^\top C_{mm}A_{mm}\, \theta_m^{t-1} - 2\eta_2 A_{mm}^\top \sum_{n \neq m} \hat{A}_{mn}^{t-1} \hat{h}_{n,c}^{t-1}.$$

Thus $\theta_m^t = M + B\, y_m^{t-1\top}$ is an affine function of $y_m^{t-1}$.

Furthermore, the LTI measurement model gives

$$y_m^t = C_{mm}A_{mm}\, \hat{h}_{m,a}^{t-1} + \mathrm{Var}\,\epsilon_m^t = C_{mm}A_{mm}\left(\hat{h}_{m,c}^{t-1} + \theta_m^{t-1}y_m^{t-1}\right) + \mathrm{Var}\,\epsilon_m^t.$$

Rearranging we obtain,
$$y_m^t = N + D\, y_m^{t-1} + \operatorname{Var} \epsilon_m^t,$$
where,
$$N = C_{mm} A_{mm}\, \hat{h}_{m,c}^{t-1}, \quad D = C_{mm} A_{mm}\, \theta_m^{t-1}.$$
Thus $y_m^t$ is also an affine function of $y_m^{t-1}$.

Let $u = y_m^{t-1}$. Then
$$\theta_m^t = M + B\, u^\top, \quad y_m^t = N + D\, u + \operatorname{Var} \epsilon_m^t.$$
Since $\operatorname{Var} \epsilon_m^t$ is zero-mean and independent of $u$, we have
$$\operatorname{Cov}(\theta_m^t, y_m^t) = \operatorname{Cov}(M + B\, u^\top,\, N + D\, u) = B\, \operatorname{Cov}(u^\top, u)\, D^\top = B\, \operatorname{Var}(u)\, D^\top.$$
By assumption $\operatorname{Var}(u) = \operatorname{Var}(y_m^{t-1}) > 0$, and $B, D$ are nonzero (since the update and measurement matrices are full-rank). Therefore $\operatorname{Cov}(\theta_m^t, y_m^t) = B\, \operatorname{Var}(y_m^{t-1})\, D^\top \neq 0$.

Hence $\Omega_m^t \neq 0$, as claimed. $\qquad\square$

### F.2 PROPOSITION 6.2

• **Proposition** (**Client Model-Client State Dependence**) *Let* $\Lambda_m^t = \operatorname{Cov}(v_m^t,\, \hat{h}_{m,a}^t)$. *Then we have the following recursion within the client,* $\Lambda_m^t = \Sigma_{\theta_m}^t\, (I_{d_m} \otimes \mu_{y_m}^t)\, +\, \Omega_m^t\, (\mu_{v_m}^t \otimes I_{d_m})$,

*Proof.* We prove the recursion for $\Lambda_m^t = \operatorname{Cov}(v_m^t, \hat{h}_{m,a}^t)$ using the paper's definitions:

From Table 1 and Eq. (4), we have the following definitions,
$$\Sigma_{\theta_m}^t := \operatorname{Var}(v_m^t) = \mathbb{E}[v_m^t v_m^{t\top}] - \mu_{\theta_m}^t \mu_{\theta_m}^{t\top}$$
$$\Omega_m^t := \operatorname{Cov}(v_m^t, y_m^t) = \mathbb{E}[v_m^t y_m^{t\top}] - \mu_{\theta_m}^t \mu_{y_m}^{t\top}$$
$$\hat{h}_{m,a}^t := \hat{h}_{m,c}^t + \theta_m^t y_m^t = \hat{h}_{m,c}^t + (y_m^{t\top} \otimes I_{p_m})v_m^t$$

Expanding $\Lambda_m^t$ from first principles, and using the definition of $\hat{h}_{m,a}^t$ we have,
$$\Lambda_m^t = \mathbb{E}[v_m^t \hat{h}_{m,a}^{t\top}] - \mu_{\theta_m}^t \mu_{h_{m,a}}^{t\top}$$
$$= \mathbb{E}\left[v_m^t \left(\hat{h}_{m,c}^{t\top} + v_m^{t\top}(y_m^t \otimes I_{p_m})\right)\right] - \mu_{\theta_m}^t \left(\hat{h}_{m,c}^{t\top} + \mu_{\theta_m}^{t\top}(\mu_{y_m}^t \otimes I_{p_m})\right)$$
$$= \mathbb{E}[v_m^t v_m^{t\top}(y_m^t \otimes I_{p_m})] - \mu_{\theta_m}^t \mu_{\theta_m}^{t\top}(\mu_{y_m}^t \otimes I_{p_m})$$

Analyzing the key expectation terms and substituting the definitions of $\Sigma_{\theta_m}^t$, and $\Omega_m^t$ we have,
$$\mathbb{E}[v_m^t v_m^{t\top}(y_m^t \otimes I_{p_m})] = \mathbb{E}\left[\left(\Sigma_{\theta_m}^t + \mu_{\theta_m}^t \mu_{\theta_m}^{t\top}\right)(y_m^t \otimes I_{p_m})\right] \quad (\text{since } \mathbb{E}[v_m^t v_m^{t\top}] = \Sigma_{\theta_m}^t + \mu_{\theta_m}^t \mu_{\theta_m}^{t\top})$$
$$= \Sigma_{\theta_m}^t(\mu_{y_m}^t \otimes I_{p_m}) + \mu_{\theta_m}^t \mu_{\theta_m}^{t\top}(\mu_{y_m}^t \otimes I_{p_m})$$
$$+ \mathbb{E}[(v_m^t - \mu_{\theta_m}^t)(v_m^t - \mu_{\theta_m}^t)^\top(y_m^t - \mu_{y_m}^t \otimes I_{p_m})]$$
$$= \Sigma_{\theta_m}^t(\mu_{y_m}^t \otimes I_{p_m}) + \mu_{\theta_m}^t \mu_{\theta_m}^{t\top}(\mu_{y_m}^t \otimes I_{p_m})$$
$$+ \mu_{\theta_m}^t \mathbb{E}[(v_m^t - \mu_{\theta_m}^t)(y_m^t - \mu_{y_m}^t)^\top] \otimes I_{p_m}$$
$$= \Sigma_{\theta_m}^t(\mu_{y_m}^t \otimes I_{p_m}) + \mu_{\theta_m}^t \mu_{\theta_m}^{t\top}(\mu_{y_m}^t \otimes I_{p_m}) + \mu_{\theta_m}^t \Omega_m^{t\top} \otimes I_{p_m}$$

Substituting the expression for $\mathbb{E}[v_m^t v_m^{t\top}(y_m^t \otimes I_{p_m})]$ (obtained above), back into $\Lambda_m^t$ we have,
$$\Lambda_m^t = \left(\Sigma_{\theta_m}^t(\mu_{y_m}^t \otimes I_{p_m}) + \mu_{\theta_m}^t \mu_{\theta_m}^{t\top}(\mu_{y_m}^t \otimes I_{p_m}) + \mu_{\theta_m}^t \Omega_m^{t\top} \otimes I_{p_m}\right)$$
$$- \mu_{\theta_m}^t \mu_{\theta_m}^{t\top}(\mu_{y_m}^t \otimes I_{p_m})$$
$$= \Sigma_{\theta_m}^t(\mu_{y_m}^t \otimes I_{p_m}) + \mu_{\theta_m}^t \Omega_m^{t\top} \otimes I_{p_m}$$

Recognizing that $\Omega_m^{t\top} \otimes I_{p_m} = (\Omega_m^t \otimes I_{p_m})^\top$, and simplifying the second term we obtain,
$$\mu_{\theta_m}^t \Omega_m^{t\top} \otimes I_{p_m} = (\Omega_m^t \otimes I_{p_m})\mu_{\theta_m}^t = \Omega_m^t(\mu_{\theta_m}^t \otimes I_{d_m})$$

Substituting this into the expression for $\Lambda_m^t$ we have,

$$\Lambda_m^t = \Sigma_{\theta_m}^t (I_{d_m} \otimes \mu_{y_m}^t) + \Omega_m^t (\mu_{\theta_m}^t \otimes I_{d_m})$$

$\square$

### F.3 LEMMA 6.3

**Assumption (A6):** The client's augmented hidden state changes only by a small amount between two consecutive time-steps $\Delta h_m^t := \hat{h}_{m,a}^{t+1} - \hat{h}_{m,a}^t$ satisfying $\left\| \Delta h_m^t \right\|_2 \leq \varepsilon$ with $\varepsilon$ is small.

• **Lemma (Client State-Sever Model Dependence)** *The cross-covariance term* $\Gamma_{mn}^t := \text{Cov}(a_{mn}^t, \hat{h}_{m,a}^t)$ *follows the recursive equation:* $\Gamma_{mn}^{t+1} = D_n^t \Gamma_{mn}^t + 2\gamma B_{mn}^t \Sigma_{h_m}^t$ *where,* $D_n^t = (I - 2\gamma \hat{h}_{n,c}^t \hat{h}_{n,c}^{t\top}) \otimes I, B_{mn}^t = \hat{h}_{n,c}^t \otimes A_{mm}$, *and* $\Sigma_{h_m}^t = \text{Var}(\hat{h}_{m,a}^t)$.

*Proof.* Gradient descent on the quadratic loss $L_s$ with step size $\gamma$ gives,

$$a_{mn}^{t+1} = D_n^t a_{mn}^t + 2\gamma B_{mn}^t \hat{h}_{m,a}^t,$$

where $D_n^t = (I - 2\gamma \hat{h}_{n,c}^t \hat{h}_{n,c}^{t\top}) \otimes I$ and $B_{mn}^t = \hat{h}_{n,c}^t \otimes A_{mm}$.

Taking the column covariance with $\hat{h}_{m,a}^t$ yields the shifted covariance term

$$\tilde{\Gamma}_{mn}^{t+1} := \text{Cov}(a_{mn}^{t+1}, \hat{h}_{m,a}^t) = D_n^t \Gamma_{mn}^t + 2\gamma B_{mn}^t \Sigma_{h_m}^t.$$

By Assumption **(A6)**,

$$\hat{h}_{m,a}^{t+1} = \hat{h}_{m,a}^t + \Delta h_m^t, \qquad \|\Delta h_m^t\|_2 \leq \varepsilon.$$

Defining, $\Gamma_{mn}^{t+1} := \text{Cov}(a_{mn}^{t+1}, \hat{h}_{m,a}^{t+1})$ and expanding, we get,

$$\Gamma_{mn}^{t+1} = \text{Cov}(a_{mn}^{t+1}, \hat{h}_{m,a}^t + \Delta h_m^t)$$

Let us define $E_{mn}^t := \text{Cov}(a_{mn}^{t+1}, \hat{h}_{m,a}^t + \Delta h_m^t)$. Thne the matrix Cauchy–Schwarz inequality gives,

$$\|E_{mn}^t\|_2 \leq \sqrt{\text{tr}(\Sigma_{A_{mn}}^{t+1})} \, \varepsilon = O(\varepsilon).$$

Using the above expression we obtain,

$$\Gamma_{mn}^{t+1} = D_n^t \Gamma_{mn}^t + 2\gamma B_{mn}^t \Sigma_{h_m}^t + O(\varepsilon),$$

If $\epsilon$ is small (Assumption **(A6)**), we get,

$$\Gamma_{mn}^{t+1} = D_n^t \Gamma_{mn}^t + 2\gamma B_{mn}^t \Sigma_{h_m}^t$$

$\square$

### F.4 LEMMA 6.4

• **Lemma (Client Model-Server Model Dependence)** *The term* $\Psi_{mn}^t := \text{Cov}(a_{mn}^t, v_m^t)$ *evolves as,* $\Psi_{mn}^{t+1} = D_n^t \Psi_{mn}^t H_m^{t\top} + D_n^t \Gamma_{mn}^t G_m^{t\top} - D_n^t \Sigma_{A_{mn}}^t P_m^{t\top} + 2\gamma B_{mn}^t \Lambda_m^t H_m^{t\top} + 2\gamma B_{mn}^t \Sigma_{h_m}^t G_m^{t\top} - 2\gamma B_{mn}^t \Gamma_{mn}^{t\top} P_m^{t\top}$, *with the following gain matrices,* $B_{mn}^t = \hat{h}_{n,c}^t \otimes A_{mm}$,

$$D_n^t = (I - 2\gamma \hat{h}_{n,c}^t \hat{h}_{n,c}^{t\top}) \otimes I, G_m^t = 2\eta_1 (y_m^t \otimes (C_{mm} A_{mm})^\top), P_m^t = -2\eta_2 (y_m^t \otimes A_{mm}^\top)$$

*and* $H_m^t = I_{p_m d_m} - 2\eta_1 (y_m^t y_m^{t\top}) \otimes ((C_{mm} A_{mm})^\top C_{mm} A_{mm}) - 2\eta_2 (y_m^t y_m^{t\top}) \otimes (A_{mm}^\top A_{mm})$

*Proof.* From the loss $L_s$ one gradient–descent step with stepsize $\gamma$ gives,

$$a_{mn}^{t+1} = D_n^t a_{mn}^t + 2\gamma B_{mn}^t \hat{h}_{m,a}^t.$$

The update $\theta_m^{t+1} = \theta_m^t - \eta_1 \nabla_{\theta_m}(L_m)_a - \eta_2 \nabla_{\theta_m} L_s$ is linear in $(\theta_m^t, \hat{h}_{m,a}^t, a_{mn}^t)$; in vectorized form,

$$v_m^{t+1} = H_m^t v_m^t + G_m^t \hat{h}_{m,a}^t - P_m^t a_{mn}^t.$$

Compute $\Psi_{mn}^{t+1} = \mathrm{Cov}(a_{mn}^{t+1}, v_m^{t+1})$ using the above two equations,

$$
\begin{aligned}
\Psi_{mn}^{t+1} &= \mathrm{Cov}\big(D_n^t a_{mn}^t + 2\gamma B_{mn}^t \hat{h}_{m,a}, \ H_m^t v_m^t + G_m^t \hat{h}_{m,a}^t - P_m^t a_{mn}^t\big) \\
&= D_n^t \, \mathrm{Cov}(a_{mn}^t, v_m^t) H_m^{t\top} + D_n^t \, \mathrm{Cov}(a_{mn}^t, \hat{h}_{m,a}^t) G_m^{t\top} - D_n^t \, \mathrm{Cov}(a_{mn}^t, a_{mn}^t) P_m^{t\top} \\
&\quad + 2\gamma B_{mn}^t \, \mathrm{Cov}(\hat{h}_{m,a}^t, v_m^t) H_m^{t\top} + 2\gamma B_{mn}^t \, \mathrm{Cov}(\hat{h}_{m,a}^t, \hat{h}_{m,a}^t) G_m^{t\top} - 2\gamma B_{mn}^t \, \mathrm{Cov}(\hat{h}_{m,a}^t, a_{mn}^t) P_m^{t\top} \\
&= D_n^t \Psi_{mn}^t H_m^{t\top} + D_n^t \Gamma_{mn}^t G_m^{t\top} - D_n^t \Sigma_{A_{mn}}^t P_m^{t\top} + 2\gamma B_{mn}^t \Lambda_m^t H_m^{t\top} + 2\gamma B_{mn}^t \Sigma_{h_m}^t G_m^{t\top} - 2\gamma B_{mn}^t \Gamma_{mn}^{t\top} P_m^{t\top}.
\end{aligned}
$$

Grouping the six contributions yields the given recursion of $\Psi_{mn}^{t+1}$. $\qquad\square$

## F.5 LEMMA 6.5

• **Lemma (Uncertainty in Client to Server Communication)** *Let* $\kappa_m = \mathrm{tr}\big(\Sigma_{y_m}^t\big) + \|\mu_{y_m}^t\|^2$. *Then the variance in the* $\hat{h}_{m,a}^t$ *is given by,* $\Sigma_{h_m}^t = \kappa_m \Sigma_{\theta_m}^t + \Omega_m^t (\mu_{y_m}^t \otimes I_{p_m})^\top + (\mu_{y_m}^t \otimes I_{p_m}) \Omega_m^{t\top}.$

*Proof.* From the definition of augmented client states we have, $q_m^t := \hat{h}_{m,a}^t - \hat{h}_{m,c}^t = \theta_m^t \, y_m^t$, which in vectorized form is $q_m^t := Y_m^t v_m^t$, with $Y_m^t := \big(y_m^{t\top} \otimes I_{p_m}\big)$. Therefore, by definition we have,

$$\mathrm{Var}(q_m^t) = \mathbb{E}[Y_m^t v_m^t v_m^{t\top} Y_m^{t\top}] - \mathbb{E}[Y_m^t v_m^t]\mathbb{E}[Y_m^t v_m^t]^\top.$$

Computing the second moment we have,

$$
\begin{aligned}
\mathbb{E}[Y_m^t v_m^t v_m^{t\top} Y_m^{t\top}] &= \mathbb{E}\big[(y_m^t \otimes I_{p_m}) v_m^t v_m^{t\top} (y_m^{t\top} \otimes I_{p_m})\big] \\
&= \mathbb{E}\big[(y_m^t y_m^{t\top}) \otimes (v_m^t v_m^{t\top})\big] \\
&= \mathbb{E}[y_m^t y_m^{t\top}] \otimes \mathbb{E}[v_m^t v_m^{t\top}] + \mathrm{Cov}(y_m^t \otimes v_m^t) \\
&= (\Sigma_{y_m}^t + \mu_{y_m}^t \mu_{y_m}^{t\top}) \otimes (\Sigma_{\theta_m}^t + \mu_{\theta_m}^t \mu_{\theta_m}^{t\top}) \\
&\quad + \Omega_m^t \otimes (\mu_{y_m}^t \otimes I_{p_m})^\top + (\mu_{y_m}^t \otimes I_{p_m}) \otimes \Omega_m^{t\top}.
\end{aligned}
$$

The first moment is given by,

$$\mathbb{E}[Y_m^t v_m^t] = (\mu_{y_m}^t \otimes I_{p_m})\mu_{\theta_m}^t + \Omega_m^t$$

Computing the outer product of the first moments we have,

$$
\begin{aligned}
\mathbb{E}[Y_m^t v_m^t]\mathbb{E}[Y_m^t v_m^t]^\top &= (\mu_{y_m}^t \mu_{y_m}^{t\top}) \otimes (\mu_{\theta_m}^t \mu_{\theta_m}^{t\top}) \\
&\quad + \Omega_m^t (\mu_{y_m}^t \otimes I_{p_m})^\top + (\mu_{y_m}^t \otimes I_{p_m})\Omega_m^{t\top}.
\end{aligned}
$$

Subtracting outer product of first moment from second moment we obtain,

$$
\begin{aligned}
\mathrm{Var}(q_m^t) &= \big[(\Sigma_{y_m}^t + \mu_{y_m}^t \mu_{y_m}^{t\top}) \otimes \Sigma_{\theta_m}^t\big] + \big[(\Sigma_{y_m}^t + \mu_{y_m}^t \mu_{y_m}^{t\top}) \otimes \mu_{\theta_m}^t \mu_{\theta_m}^{t\top}\big] \\
&\quad - (\mu_{y_m}^t \mu_{y_m}^{t\top}) \otimes (\mu_{\theta_m}^t \mu_{\theta_m}^{t\top}) \\
&\quad + \Omega_m^t (\mu_{y_m}^t \otimes I_{p_m})^\top + (\mu_{y_m}^t \otimes I_{p_m})\Omega_m^{t\top}.
\end{aligned}
$$

The term $(\Sigma_{y_m}^t + \mu\mu^\top) \otimes \mu_\theta \mu_\theta^\top - \mu\mu^\top \otimes \mu_\theta \mu_\theta^\top$ simplifies to $\Sigma_{y_m}^t \otimes \mu_\theta \mu_\theta^\top$. By design, $\Sigma_{y_m}^t \otimes \mu_\theta \mu_\theta^\top$ is absorbed into $\kappa_m \Sigma_{\theta_m}^t$ via trace normalization, leaving,

$$\mathrm{Var}(q_m^t) = \kappa_m \Sigma_{\theta_m}^t + \Omega_m^t (\mu_{y_m}^t \otimes I_{p_m})^\top + (\mu_{y_m}^t \otimes I_{p_m})\Omega_m^{t\top}.$$

Using the definition of $q_m^t$ we have,

$$\mathrm{Var}(\hat{h}_{m,a}^t - \hat{h}_{m,c}^t) = \kappa_m \Sigma_{\theta_m}^t + \Omega_m^t (\mu_{y_m}^t \otimes I_{p_m})^\top + (\mu_{y_m}^t \otimes I_{p_m})\Omega_m^{t\top}.$$

Since $\hat{h}_{m,c}^t$ is deterministic, we have,

$$\Sigma_{h_m}^t := \mathrm{Var}(\hat{h}_{m,a}^t) = \kappa_m \Sigma_{\theta_m}^t + \Omega_m^t (\mu_{y_m}^t \otimes I_{p_m})^\top + (\mu_{y_m}^t \otimes I_{p_m})\Omega_m^{t\top}.$$

$\qquad\square$

## F.6   LEMMA 6.6

• **Lemma** (**Uncertainty in Server to Client Communication**) *With notation as above, the uncertainty in the gradient communicated by the server is given by,* $\mathrm{Var}\big(g_{m,s}^{t+1}\big) = A_{mm}^{\top} U^t A_{mm}$, *where* $U^t = A_{mm} \Sigma_{h_m}^t A_{mm}^{\top} + \sum_{n\neq m}(h_{n,c}^t h_{n,c}^{t\top}) \Sigma_{A_{mn}}^t - 2 \sum_{n\neq m} A_{mm} \Gamma_{mn}^t h_{n,c}^{t\top}$.

*Proof.* Let $r^t := A_{mm}(\hat{h}_{m,a}^t - \hat{h}_{m,c}^t) - \sum_{n\neq m} \hat{A}_{mn}^t \hat{h}_{n,c}^t$. We know that, $g_{m,s}^{t+1} = A_{mm}^{\top} r^t$. Then

$$\mathrm{Var}(g_{m,s}^{t+1}) = A_{mm}^{\top} \, \mathrm{Var}(r^t) \, A_{mm}.$$

We compute

$$\mathrm{Var}(r^t) := \mathrm{Var}\big(A_{mm} \, \hat{h}_{m,a}^t\big) + \sum_{n\neq m} \mathrm{Var}\big(\hat{A}_{mn}^t \, \hat{h}_{n,c}^t\big) - 2 \sum_{n\neq m} \mathrm{Cov}\big(A_{mm} \, \hat{h}_{m,a}^t, \, \hat{A}_{mn}^t \, \hat{h}_{n,c}^t\big),$$

Since $\hat{h}_{m,c}^t$ is deterministic. We have,

$$\mathrm{Var}(A_{mm} \, \hat{h}_{m,a}^t) = A_{mm} \Sigma_{h_m}^t A_{mm}^{\top},$$

$\mathrm{Var}(\hat{A}_{mn}^t \, \hat{h}_{n,c}^t) = (h_{n,c}^t h_{n,c}^{t\top}) \Sigma_{A_{mn}}^t$; $\mathrm{Cov}\big(A_{mm} \, \hat{h}_{m,a}^t, \, \hat{A}_{mn}^t \, \hat{h}_{n,c}^t\big) = A_{mm} \Gamma_{mn}^t h_{n,c}^{t\top}$.

Putting these into $\mathrm{Var}(r^t)$ gives exactly

$$U^t = A_{mm} \Sigma_{h_m}^t A_{mm}^{\top} + \sum_{n\neq m} (h_{n,c}^t h_{n,c}^{t\top}) \Sigma_{A_{mn}}^t - 2 \sum_{n\neq m} A_{mm} \Gamma_{mn}^t h_{n,c}^{t\top}.$$

Hence $\mathrm{Var}(g_{m,s}^{t+1}) = A_{mm}^{\top} U^t A_{mm}$, as claimed. $\qquad\square$

## F.7   THEOREM 6.7

• **Theorem** (**Uncertainty Propagation within the Server**) *The server model parameter $a_{mn}^t$'s covariance $\Sigma_{A_{mn}}^t$ evolves as,* $\Sigma_{A_{mn}}^{t+1} = D_n^t \Sigma_{A_{mn}}^t D_n^{t\top} + 4\gamma^2 \big(\hat{h}_{n,c}^t \otimes A_{mm}\big) \Sigma_{h_m}^t \big(\hat{h}_{n,c}^t \otimes A_{mm}\big)^{\top} + 2\gamma\Big(D_n^t \Gamma_{mn}^t B_{mn}^{t\top} + B_{mn}^t \Gamma_{mn}^{t\top} D_n^{t\top}\Big)$ *with* $D_n^t = \big(I - 2\gamma \, \hat{h}_{n,c}^t \, \hat{h}_{n,c}^{t\top}\big) \otimes I$, *and* $B_{mn}^t = \hat{h}_{n,c}^t \otimes A_{mm}$

*Proof.* At round $t$, the server gradient update as follows:

$$\hat{A}_{mn}^{t+1} = \hat{A}_{mn}^t \big(I - 2\gamma \, \hat{h}_{n,c}^t \, \hat{h}_{n,c}^{t\top}\big) + 2\gamma A_{mm}\big[\hat{h}_{m,a}^t - \hat{h}_{m,c}^t\big] \hat{h}_{n,c}^{t\top} - 2\gamma \sum_{p\neq m,n} \hat{A}_{mp}^t \hat{h}_{p,c}^t \hat{h}_{n,c}^{t\top}.$$

Under Assumption (**A1**) (off-diagonal blocks independent), the last summation term contributes no covariance with $\hat{A}_{mn}^t$ and can be omitted when computing $\mathrm{Var}(\hat{A}_{mn}^{t+1})$.

Apply $\mathrm{Vec}(\cdot)$ and use the property that: $\mathrm{Vec}(XB) = (B^{\top} \otimes I) \, \mathrm{Vec}(X)$ and $\mathrm{Vec}(AX) = (I \otimes A) \, \mathrm{Vec}(X)$ for any three matrices $A, B, X$.

We obtain the following after vectorization,

$$\mathrm{Vec}\big(\hat{A}_{mn}^{t+1}\big) = \Big(\big(I - 2\gamma \, \hat{h}_{n,c}^t \, \hat{h}_{n,c}^{t\top}\big) \otimes I\Big) \mathrm{Vec}\big(\hat{A}_{mn}^t\big) + 2\gamma \big(\hat{h}_{n,c}^t \otimes A_{mm}\big) \hat{h}_{m,a}^t.$$

We then define,

$$D_n^t = \big(I - 2\gamma \, \hat{h}_{n,c}^t \, \hat{h}_{n,c}^{t\top}\big) \otimes I, \quad B_{mn}^t = \hat{h}_{n,c}^t \otimes A_{mm}, \quad a_{mn}^t = \mathrm{Vec}(\hat{A}_{mn}^t).$$

Then the vectorized update is given by,

$$a_{mn}^{t+1} = D_n^t a_{mn}^t + 2\gamma B_{mn}^t \hat{h}_{m,a}^t.$$

We wish to compute $\Sigma_{A_{mn}}^{t+1} = \mathrm{Var}(a_{mn}^{t+1})$.

Using the property $\mathrm{Var}[X + Z] = \mathrm{Var}[X] + \mathrm{Var}[Z] + \mathrm{Cov}(X, Z) + \mathrm{Cov}(Z, X)$ for any two vectors X, and Z.

We set, $X = D_n^t \, a_{mn}^t, \quad Z = 2\gamma \, B_{mn}^t \, \hat{h}_{m,a}^t$. Then,
$\Sigma_{A_{mn}}^{t+1} = \mathrm{Var}[X] + \mathrm{Var}[Z] + \mathrm{Cov}(X, Z) + \mathrm{Cov}(Z, X)$.

**(1)** *Variance of X:* $D_n^t$ is deterministic, so

$$\mathrm{Var}[X] = D_n^t \, \mathrm{Var}\big(a_{mn}^t\big) \, D_n^{t\top} = D_n^t \, \Sigma_{A_{mn}}^t \, D_n^{t\top}.$$

**(2)** *Variance of Z:* $\hat{h}_{n,c}^t$ and $A_{mm}$ are fixed at round $t$, hence

$$\mathrm{Var}[Z] = 4\gamma^2 \, B_{mn}^t \, \mathrm{Var}\big(\hat{h}_{m,a}^t\big) \, B_{mn}^{t\top} = 4\gamma^2 \, (\hat{h}_{n,c}^t \otimes A_{mm}) \, \Sigma_{h_m}^t \, (\hat{h}_{n,c}^t \otimes A_{mm})^\top.$$

**(3)** *Cross-covariance terms:* Since $D_n^t$ and $B_{mn}^t$ are deterministic,

$$\mathrm{Cov}(X, Z) = 2\gamma \, D_n^t \, \mathrm{Cov}\big(a_{mn}^t, \, \hat{h}_{m,a}^t\big) \, B_{mn}^{t\top} = 2\gamma \, D_n^t \, \Gamma_{mn}^t \, B_{mn}^{t\top},$$

$$\mathrm{Cov}(Z, X) = 2\gamma \, B_{mn}^t \, \mathrm{Cov}\big(\hat{h}_{m,a}^t, \, a_{mn}^t\big) \, D_n^{t\top} = 2\gamma \, B_{mn}^t \, \Gamma_{mn}^{t\top} \, D_n^{t\top}.$$

Adding the four contributions we obtain,

$$\Sigma_{A_{mn}}^{t+1} = D_n^t \, \Sigma_{A_{mn}}^t \, D_n^{t\top} \; + \; 4\gamma^2 \, (\hat{h}_{n,c}^t \otimes A_{mm}) \, \Sigma_{h_m}^t \, (\hat{h}_{n,c}^t \otimes A_{mm})^\top$$
$$+ \; 2\gamma\Big(D_n^t \, \Gamma_{mn}^t \, B_{mn}^{t\top} \; + \; B_{mn}^t \, \Gamma_{mn}^{t\top} \, D_n^{t\top}\Big)$$

$\square$

## F.8 THEOREM 6.8

• **Theorem (Uncertainty Propagation within the Client)** *The client-parameter covariance $\Sigma_{\theta_m}^t$ obeys the following recursion:* $\Sigma_{\theta_m}^t = H_m^{t-1} \Sigma_{\theta_m}^{t-1} H_m^{t-1\top} + G_m^{t-1} \Sigma_{h_m}^{t-1} G_m^{t-1\top} + (X_m + X_m^\top)$
$- \sum_{n\neq m}(Y_{mn} + Y_{mn}^\top) - \sum_{n\neq m}(Z_{mn} + Z_{mn}^\top) + \sum_{n\neq m} P_m^{t-1} \Sigma_{A_{mn}}^{t-1} P_m^{t-1\top},$

*where,* $\quad X_m = H_m^{t-1} \Lambda_m^{t-1} G_m^{t-1\top}, \quad Y_{mn} = H_m^{t-1} \Psi_{mn}^{t-1} P_m^{t-1\top}, Z_{mn} = G_m^{t-1} \Gamma_{mn}^{t-1} P_m^{t-1\top},$
*and,* $\quad G_m^{t-1} := 2\eta_1\big(y_m^{t-1} \otimes (C_{mm}A_{mm})^\top\big), P_m^{t-1} := -2\eta_2\big(y_m^{t-1} \otimes A_{mm}^\top\big), H_m^{t-1} :=$
$I_{p_m d_m} - 2\eta_1 (y_m^{t-1} y_m^{t-1\top}) \otimes \big((C_{mm}A_{mm})^\top C_{mm}A_{mm}\big) - 2\eta_2 (y_m^{t-1} y_m^{t-1\top}) \otimes \big(A_{mm}^\top A_{mm}\big)$

*Proof.* All random variables, distributional assumptions (A1–A6) and second–moment symbols $\Sigma_{\theta_m}^t, \Sigma_{h_m}^t, \Sigma_{A_{mn}}^t, \Lambda_m^t, \Psi_{mn}^t, \Gamma_{mn}^t, \Omega_m^t$ are defined in Section 5.

Computing the analytical values of $\nabla_{\theta_m^t}(L_m)_a$, and $\nabla_{\theta_m^t} L_s$, and substituting them in Eq (5) of Section 3.1, we have the following client model update (for the FedGC framework):

$$\theta_m^t = \theta_m^{t-1} + 2\eta_1(C_{mm}A_{mm})^\top \Big(y_m^{t-1} - C_{mm}A_{mm}\hat{h}_{m,c}^{t-1}\Big)y_m^{t-1\top} - 2\eta_2 A_{mm}^\top \Big(\sum_{n\neq m} A_{mn}^{t-1}\hat{h}_{n,c}^{t-1}\Big)y_m^{t-1\top}.$$

Let $v_m^t := \mathrm{Vec}(\theta_m^t)$. Using the property $\mathrm{Vec}(AXB) = (B^\top \otimes A)\,\mathrm{Vec}(X)$ and $\mathrm{Vec}(A\,\hat{h}) = (\hat{h}^\top \otimes I)\,\mathrm{Vec}(A)$ for any matrices $A, B, X$ and vector $\hat{h}$, we obtain the following:

$$v_m^t = H_m^{t-1} v_m^{t-1} + G_m^{t-1} \hat{h}_{m,a}^{t-1} + P_m^{t-1} \sum_{n\neq m} a_{mn}^{t-1} + u_m^{t-1},$$

where these matrices $(H_m^t, G_m^t, P_m^t)$ are exactly those stated in the theorem. We define $u_m^{t-1} := 2\eta_1\big(y_m^{t-1} \otimes (C_{mm}A_{mm})^\top\big)\big(y_m^t - C_{mm}A_{mm}\hat{h}_{m,c}^{t-1}\big)$. By assumption **(A4)** $u_m^{t-1}$ has mean 0 and vanishing covariance: $\mathrm{E}[u_m^{t-1}] = 0, \; \mathrm{Var}(u_m^{t-1}) = 0$.

First, we have the following:

$$\mathrm{Var}(H_m^{t-1} v_m^{t-1}) = H_m^{t-1} \Sigma_{\theta_m}^{t-1} H_m^{t-1\top}, \quad \mathrm{Var}(G_m^{t-1} \hat{h}_{m,a}^{t-1}) = G_m^{t-1} \Sigma_{h_m}^{t-1} G_m^{t-1\top}.$$

We also denote $S_m^{t-1} := \sum_{n \neq m} a_{mn}^{t-1}$. Independence of different off-diagonal blocks $a_{mn}^{t-1}$ in assumption (A2) yields $\text{Var}(S_m^{t-1}) = \sum_{n \neq m} \Sigma_{A_{mn}}^{t-1}$; hence

$$\text{Var}(P_m^{t-1} S_m^{t-1}) = \sum_{n \neq m} P_m^{t-1} \Sigma_{A_{mn}}^{t-1} P_m^{t-1\top}.$$

Independence assumptions (A2) imply that cross terms with different client indices cancel. The only non-zero covariances are

$$\text{Cov}(Hv,\, G\hat{h}) = H_m^{t-1} \Lambda_m^{t-1} G_m^{t-1\top} = X_m,$$
$$\text{Cov}(Hv,\, PS) = \sum_{n \neq m} H_m^{t-1} \Psi_{mn}^{t-1} P_m^{t-1\top} = \sum_{n \neq m} Y_{mn},$$
$$\text{Cov}(G\hat{h},\, PS) = \sum_{n \neq m} G_m^{t-1} \Gamma_{mn}^{t-1} P_m^{t-1\top} = \sum_{n \neq m} Z_{mn}.$$

Each term $X_m, Y_{mn}, Z_{mn}$ appears together with its transpose in the variance expansion. Applying $\text{Var}(\cdot)$ to $\text{Vec}(\theta_m)$, and using $\text{Var}(u_m^{t-1}) = 0$, we obtain:

$$\Sigma_{\theta_m}^t = H_m^{t-1} \Sigma_{\theta_m}^{t-1} H_m^{t-1\top} + G_m^{t-1} \Sigma_{h_m}^{t-1} G_m^{t-1\top} + (X_m + X_m^\top)$$
$$- \sum_{n \neq m} (Y_{mn} + Y_{mn}^\top) - \sum_{n \neq m} (Z_{mn} + Z_{mn}^\top) + \sum_{n \neq m} P_m^{t-1} \Sigma_{A_{mn}}^{t-1} P_m^{t-1\top}.$$

$\square$

### F.9 PROPOSITION 7.1

• **Proposition** (**Gain Matrices Convergence**) *Under the above assumptions, the gain matrices used in Section 6 converges as,* $\lim_{t \to \infty} (D_n^t, H_m^t, G_m^t, P_m^t) = (D_n, H_m, G_m, P_m)$ *where,* $D_n = (I - 2\gamma \hat{h}_{n,c} \hat{h}_{n,c}^\top) \otimes I$, $G_m = 2\eta_1(\mu_{y_m} \otimes (C_{mm} A_{mm})^\top)$, $P_m = -2\eta_2(\mu_{y_m} \otimes A_{mm}^\top)$ *and* $H_m = I_{p_m d_m} - 2\eta_1(\mu_{y_m} \mu_{y_m}^\top) \otimes ((C_{mm} A_{mm})^\top C_{mm} A_{mm}) - 2\eta_2(\mu_{y_m} \mu_{y_m}^\top) \otimes (A_{mm}^\top A_{mm})$.

*Proof.* We prove the convergence of the gain matrices under the assumptions (provided in Section 7):

(I) $\lim_{t \to \infty} y_m^t = \mu_{y_m}$ (client data converges)

(II) $\lim_{t \to \infty} \hat{h}_{m,c}^t = \hat{h}_{m,c}$ (client state estimates converge)

First, for $D_n^t = (I - 2\gamma \hat{h}_{n,c}^t \hat{h}_{n,c}^{t\top}) \otimes I$ we have,

$$\lim_{t \to \infty} D_n^t = \left( I - 2\gamma \left( \lim_{t \to \infty} \hat{h}_{n,c}^t \right) \left( \lim_{t \to \infty} \hat{h}_{n,c}^t \right)^\top \right) \otimes I$$
$$= (I - 2\gamma \hat{h}_{n,c} \hat{h}_{n,c}^\top) \otimes I =: D_n$$

Next for $G_m^t = 2\eta_1(y_m^t \otimes (C_{mm} A_{mm})^\top)$ we have,

$$\lim_{t \to \infty} G_m^t = 2\eta_1 \left( \left( \lim_{t \to \infty} y_m^t \right) \otimes (C_{mm} A_{mm})^\top \right)$$
$$= 2\eta_1(\mu_{y_m} \otimes (C_{mm} A_{mm})^\top) =: G_m$$

Similarly for $P_m^t = -2\eta_2(y_m^t \otimes A_{mm}^\top)$ we have,

$$\lim_{t \to \infty} P_m^t = -2\eta_2 \left( \left( \lim_{t \to \infty} y_m^t \right) \otimes A_{mm}^\top \right)$$
$$= -2\eta_2(\mu_{y_m} \otimes A_{mm}^\top) =: P_m$$

Finally for $H_m^t$ we have,

$$\lim_{t\to\infty} H_m^t = I_{p_m d_m} - 2\eta_1 \left( \left( \lim_{t\to\infty} y_m^t y_m^{t\top} \right) \otimes \left( (C_{mm}A_{mm})^\top C_{mm}A_{mm} \right) \right)$$
$$- 2\eta_2 \left( \left( \lim_{t\to\infty} y_m^t y_m^{t\top} \right) \otimes (A_{mm}^\top A_{mm}) \right)$$

Using the fact that $\lim_{t\to\infty} y_m^t y_m^{t\top} = \mu_{y_m} \mu_{y_m}^\top + \Sigma_{y_m}$ (from the stationary distribution), but under Assumption **(A4)** that $\Sigma_{y_m}$ is constant, we get,

$$H_m = I_{p_m d_m} - 2\eta_1 (\mu_{y_m} \mu_{y_m}^\top \otimes (C_{mm}A_{mm})^\top C_{mm}A_{mm})$$
$$- 2\eta_2 (\mu_{y_m} \mu_{y_m}^\top \otimes A_{mm}^\top A_{mm})$$

$\square$

### F.10 PROPOSITION 7.2

• **Proposition** *If $\rho(D_n) < 1$ and $\rho(H_m) < 1$ then we have, $\lim_{t\to\infty}\left(\Gamma_{mn}^t, \Psi_{mn}^t\right) = \left(\Gamma_{mn}^\infty, \Psi_{mn}^\infty\right)$ with $\Gamma_{mn}^\infty = (I - D_n)^{-1} 2\gamma B_{mn}\Sigma_{h_m}^\infty$, & $\Psi_{mn}^\infty = (I - H_m \otimes D_n)^{-1} \mathrm{Vec}\left(D_n\Gamma_{mn}^\infty G_m^\top - D_n\Sigma_{A_{mn}}^\infty P_m^\top\right)$.*

*Proof.* Convergence of $\Gamma_{mn}^t$: From Lemma 6.3, we have the recursion as follows,

$$\Gamma_{mn}^{t+1} = D_n^t \Gamma_{mn}^t + 2\gamma B_{mn}^t \Sigma_{h_m}^t$$

Taking limits $t \to \infty$ and using Proposition 7.1 we obtain,

$$\Gamma_{mn}^\infty = D_n\Gamma_{mn}^\infty + 2\gamma B_{mn}\Sigma_{h_m}^\infty$$
$$(I - D_n)\Gamma_{mn}^\infty = 2\gamma B_{mn}\Sigma_{h_m}^\infty$$

Since $\rho(D_n) < 1$, the matrix $(I - D_n)$ is invertible, giving:

$$\Gamma_{mn}^\infty = (I - D_n)^{-1} 2\gamma B_{mn}\Sigma_{h_m}^\infty$$

Convergence of $\Psi_{mn}^t$: From Lemma 6.4, the recursion is given by,

$$\Psi_{mn}^{t+1} = D_n^t \Psi_{mn}^t H_m^{t\top} + D_n^t \Gamma_{mn}^t G_m^{t\top} - D_n^t \Sigma_{A_{mn}}^t P_m^{t\top}$$
$$+ 2\gamma B_{mn}^t \Lambda_m^t H_m^{t\top} + 2\gamma B_{mn}^t \Sigma_{h_m}^t G_m^{t\top} - 2\gamma B_{mn}^t \Gamma_{mn}^{t\top} P_m^{t\top}$$

At steady-state, using Proposition 7.1 we obtain,

$$\Psi_{mn}^\infty = D_n\Psi_{mn}^\infty H_m^\top + D_n\Gamma_{mn}^\infty G_m^\top - D_n\Sigma_{A_{mn}}^\infty P_m^\top$$
$$+ 2\gamma B_{mn}\Lambda_m^\infty H_m^\top + 2\gamma B_{mn}\Sigma_{h_m}^\infty G_m^\top - 2\gamma B_{mn}\Gamma_{mn}^{\infty\top} P_m^\top$$

This can be rewritten as a vectorized equation using $\mathrm{Vec}(\cdot)$,

$$\mathrm{Vec}(\Psi_{mn}^\infty) = (H_m \otimes D_n)\mathrm{Vec}(\Psi_{mn}^\infty) + \mathrm{Vec}(X)$$

where $X$ collects all remaining terms.

Since $\rho(H_m \otimes D_n) = \rho(H_m)\rho(D_n) < 1$ by assumption, we have,

$$\mathrm{Vec}(\Psi_{mn}^\infty) = (I - H_m \otimes D_n)^{-1}\mathrm{Vec}(X)$$

Substituting back $X$ we obtain,

$$\Psi_{mn}^\infty = (I - H_m \otimes D_n)^{-1}\mathrm{Vec}\left(D_n\Gamma_{mn}^\infty G_m^\top - D_n\Sigma_{A_{mn}}^\infty P_m^\top\right)$$

$\square$

### F.11  COROLLARY 7.3

• **Corollary** *The above assumptions lead to convergence of the uncertainty of the client states $\Sigma_{h_m}^t$ as follows:* $\lim_{t\to\infty} \Sigma_{h_m}^t := \Sigma_{h_m}^\infty = \kappa_m \Sigma_{\theta_m}^\infty + \Omega_m^\infty (\mu_{y_m} \otimes I)^\top + (\mu_{y_m} \otimes I)\Omega_m^{\infty\top}$, *where* $\kappa_m = \mathrm{tr}(\Sigma_{y_m}) + \|\mu_{y_m}\|^2$ *and* $\Omega_m^\infty = \Sigma_{\theta_m}^\infty \mu_{y_m}$.

*Proof.* From Lemma 6.5, the client state variance evolves as follows,

$$\Sigma_{h_m}^t = \kappa_m^t \Sigma_{\theta_m}^t + \Omega_m^t (\mu_{y_m}^t \otimes I_{p_m})^\top + (\mu_{y_m}^t \otimes I_{p_m})\Omega_m^{t\top}$$

where $\kappa_m^t = \mathrm{tr}(\Sigma_{y_m}^t) + \|\mu_{y_m}^t\|^2$.

Under the stationarity Assumption **(A4)** and Proposition 7.1 we define the following,

$$\Sigma_{\theta_m}^\infty := \lim_{t\to\infty} \Sigma_{\theta_m}^t$$
$$\Omega_m^\infty := \lim_{t\to\infty} \Omega_m^t$$
$$\kappa_m := \lim_{t\to\infty} \kappa_m^t = \mathrm{tr}(\Sigma_{y_m}) + \|\mu_{y_m}\|^2$$

From Proposition 6.1 and the steady-state analysis we have,

$$\Omega_m^\infty = \mathrm{Cov}(v_m^\infty, y_m) = \Sigma_{\theta_m}^\infty \mu_{y_m}$$

since at steady-state, the parameter covariance dominates the data-model correlation.

Substituting these limits into the variance expression, we obtain,

$$\begin{aligned}
\Sigma_{h_m}^\infty &= \kappa_m \Sigma_{\theta_m}^\infty + \Sigma_{\theta_m}^\infty \mu_{y_m} (\mu_{y_m} \otimes I_{p_m})^\top + (\mu_{y_m} \otimes I_{p_m}) \mu_{y_m}^\top \Sigma_{\theta_m}^\infty \\
&= \kappa_m \Sigma_{\theta_m}^\infty + \Sigma_{\theta_m}^\infty (\mu_{y_m} \mu_{y_m}^\top \otimes I_{p_m}) + (\mu_{y_m} \mu_{y_m}^\top \otimes I_{p_m})\Sigma_{\theta_m}^\infty
\end{aligned}$$

Simplifying using the Kronecker product properties, we have,

$$\lim_{t\to\infty} \Sigma_{h_m}^t = \kappa_m \Sigma_{\theta_m}^\infty + \Omega_m^\infty (\mu_{y_m} \otimes I_{p_m})^\top + (\mu_{y_m} \otimes I_{p_m})\Omega_m^{\infty\top}$$

with $\Omega_m^\infty = \Sigma_{\theta_m}^\infty \mu_{y_m}$. $\qquad\qquad\square$

### F.12  THEOREM 7.4

• **Theorem** (**Convergence of Server Model's Uncertainty**) Let $\rho(D_n) < 1$. Define the linear map $\mathcal{L}_n(X) = D_n X D_n^\top$ and the injection $Q_{mn}(\Sigma) = 4\gamma^2 B_{mn}(\kappa_m \Sigma + \Sigma M_m M_m^\top + M_m M_m^\top \Sigma)B_{mn}^\top$. Then, $\Sigma_{A_{mn}}^\infty := \lim_{t\to\infty} \Sigma_{A_{mn}}^t$ exists, is unique, and is given by, $\Sigma_{A_{mn}}^\infty = \sum_{k=0}^\infty \mathcal{L}_n^k(Q_{mn}(\Sigma_{\theta_m}^\infty))$

*Proof.* From Theorem 6.7, the server parameter covariance evolves as follows,

$$\begin{aligned}
\Sigma_{A_{mn}}^{t+1} &= D_n^t \Sigma_{A_{mn}}^t D_n^{t\top} + 4\gamma^2 B_{mn}^t \Sigma_{h_m}^t B_{mn}^{t\top} \\
&\quad + 2\gamma\left(D_n^t \Gamma_{mn}^t B_{mn}^{t\top} + B_{mn}^t \Gamma_{mn}^{t\top} D_n^{t\top}\right)
\end{aligned}$$

Taking limits $t \to \infty$ and using Proposition 7.1 we have,

$$\begin{aligned}
\Sigma_{A_{mn}}^\infty &= D_n \Sigma_{A_{mn}}^\infty D_n^\top + 4\gamma^2 B_{mn} \Sigma_{h_m}^\infty B_{mn}^\top \\
&\quad + 2\gamma\left(D_n \Gamma_{mn}^\infty B_{mn}^\top + B_{mn} \Gamma_{mn}^{\infty\top} D_n^\top\right)
\end{aligned}$$

From Corollary 7.3, we substitute $\Sigma_{h_m}^\infty$ as follows,

$$\Sigma_{h_m}^\infty = \kappa_m \Sigma_{\theta_m}^\infty + \Sigma_{\theta_m}^\infty M_m M_m^\top + M_m M_m^\top \Sigma_{\theta_m}^\infty$$

where $M_m = \mu_{y_m} \otimes I_{p_m}$ and $\kappa_m = \mathrm{tr}(\Sigma_{y_m}) + \|\mu_{y_m}\|^2$.

Defining the linear operator $\mathcal{L}_n(X) = D_n X D_n^\top$ and the quadratic form we have,

$$Q_{mn}(\Sigma) = 4\gamma^2 B_{mn} \left(\kappa_m \Sigma + \Sigma M_m M_m^\top + M_m M_m^\top \Sigma\right) B_{mn}^\top$$

The steady-state equation thus becomes,

$$\Sigma_{A_{mn}}^\infty = \mathcal{L}_n(\Sigma_{A_{mn}}^\infty) + Q_{mn}(\Sigma_{\theta_m}^\infty)$$

Since $\rho(D_n) < 1$ (given as a condition), the operator $\mathcal{L}_n(.)$ is a contraction, and the solution is given by the Neumann series:

$$\Sigma_{A_{mn}}^\infty = \sum_{k=0}^\infty \mathcal{L}_n^k \left(Q_{mn}(\Sigma_{\theta_m}^\infty)\right)$$

With $I$ being the identity operator, the above expression can be re-written as,

$$(I - \mathcal{L}_n)(\Sigma_{A_{mn}}^\infty) = Q_{mn}$$

This has a formal solution with operator inversion such that,

$$\Sigma_{A_{mn}}^\infty = (I - \mathcal{L}_n)^{-1} Q_{mn}$$

Since $\rho(D_n) < 1$, the Neumann series expansion is valid thus we can use,

$$(I - \mathcal{L}_n)^{-1} = \sum_{k=0}^\infty \mathcal{L}_n^k$$

Substituting this into the equation for $\Sigma_{A_{mn}}^\infty$ we obtain,

$$\Sigma_{A_{mn}}^\infty = \sum_{k=0}^\infty \mathcal{L}_n^k(Q_{mn})$$

The series converges because $\|\mathcal{L}_n^k(Q_{mn})\| \leq \rho(D_n)^{2k}\|Q_{mn}\| \to 0$ as $k \to \infty$ Uniqueness follows from the Banach fixed-point theorem, as $\mathcal{L}_n$ is a contraction mapping on the space of positive semidefinite matrices with a matrix norm.

$\square$

### F.13  THEOREM 7.5

• **Theorem** (**Convergence of Client Model's Uncertainty**) Let $\rho(H_m) < 1$. Write $\mathcal{M}_m(\Sigma) = H_m \Sigma H_m^\top$ and $R_m(\Sigma) = G_m\left(\kappa_m \Sigma + \Sigma M_m M_m^\top + M_m M_m^\top \Sigma\right)G_m^\top$ Then the steady-state $\Sigma_{\theta_m}^\infty := \lim_{t\to\infty} \Sigma_{\theta_m}^t$ is the unique solution to $\Sigma_{\theta_m}^\infty = \mathcal{M}_m(\Sigma_{\theta_m}^\infty) + R_m(\Sigma_{\theta_m}^\infty) + P_m \Sigma_{A_{mn}}^\infty P_m^\top$.

*Proof.* From Theorem 6.8, the client parameter covariance evolves as follows,

$$\Sigma_{\theta_m}^{t+1} = H_m^t \Sigma_{\theta_m}^t H_m^{t\top} + R_m^t(\Sigma_{\theta_m}^t) + P_m^t \Sigma_{A_{mn}}^t P_m^{t\top} + \text{cross terms}$$

where $R_m^t$ collects terms quadratic in $\Sigma_{\theta_m}^t$.

Under Proposition 7.1's convergence and Theorem 7.4's steady-state for $\Sigma_{A_{mn}}^\infty$, we take limits as,

$$\Sigma_{\theta_m}^\infty = H_m \Sigma_{\theta_m}^\infty H_m^\top + R_m(\Sigma_{\theta_m}^\infty) + P_m \Sigma_{A_{mn}}^\infty P_m^\top$$

The quadratic term $R_m$ derives from Corollary 7.3's expression as,

$$R_m(\Sigma) = G_m\left(\kappa_m \Sigma + \Sigma M_m M_m^\top + M_m M_m^\top \Sigma\right) G_m^\top$$

with $G_m = 2\eta_1(\mu_{y_m} \otimes (C_{mm} A_{mm})^\top)$ and $M_m = \mu_{y_m} \otimes I_{p_m}$.

Rewriting the fixed-point equation using the linear operator $\mathcal{M}_m(X) = H_m X H_m^\top$ we obtain,

$$\Sigma_{\theta_m}^\infty = \mathcal{M}_m(\Sigma_{\theta_m}^\infty) + R_m(\Sigma_{\theta_m}^\infty) + P_m \Sigma_{A_{mn}}^\infty P_m^\top$$

Since $\rho(H_m) < 1$ (given), $\mathcal{M}_m$ is a contraction, guaranteeing a unique solution. $\square$

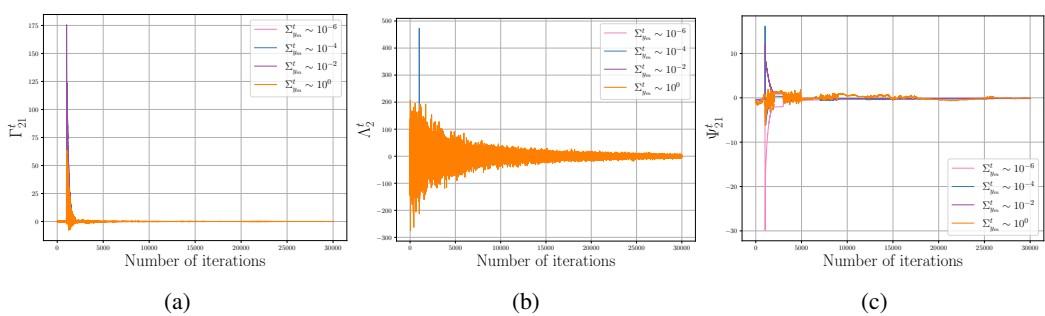

(a)                                           (b)                                           (c)

Figure 5: Uncertainty propagation in the cross-covariance terms during FedGC learning for different regimes of $\Sigma_{y_m}^t$ (a) $\Gamma_{21}^t$ vs iterations, (b) $\Lambda_2^t$ vs iterations, (c) $\Psi_{21}^t$ vs iterations.

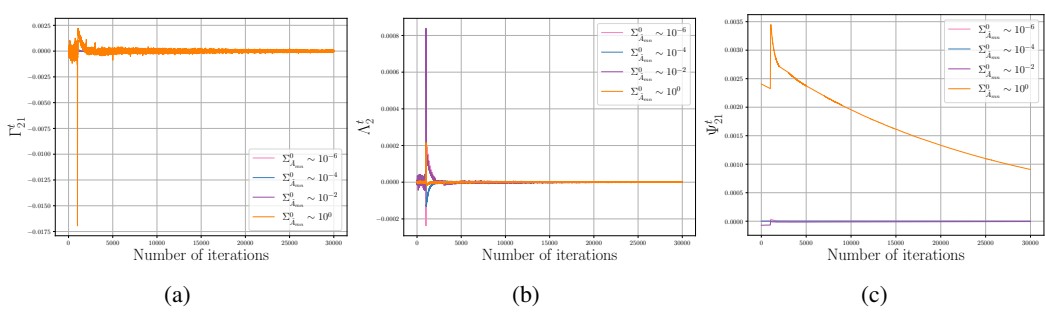

(a)                                           (b)                                           (c)

Figure 6: Uncertainty propagation in the cross-covariance terms during FedGC learning for different regimes of $\Sigma_{\hat{A}_{mn}}^0$ (a) $\Gamma_{21}^t$ vs iterations, (b) $\Lambda_2^t$ vs iterations, (c) $\Psi_{21}^t$ vs iterations.

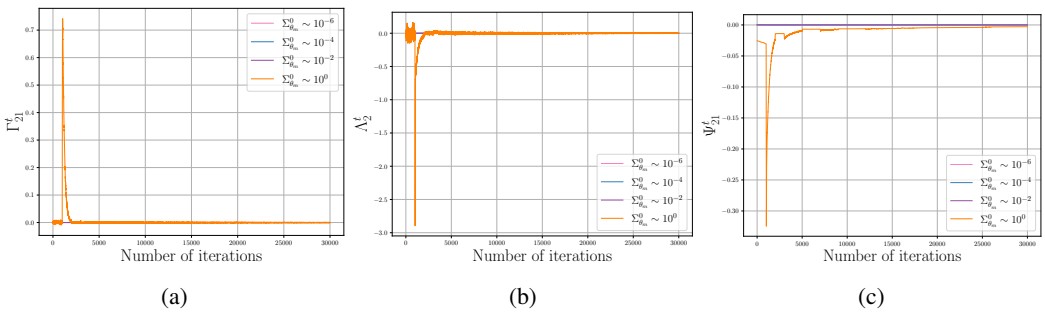

(a)                                           (b)                                           (c)

Figure 7: Uncertainty propagation in the cross-covariance terms during FedGC learning for different regimes of $\Sigma_{\theta_m^t}$ (a) $\Gamma_{21}^t$ vs iterations, (b) $\Lambda_2^t$ vs iterations, (c) $\Psi_{21}^t$ vs iterations.

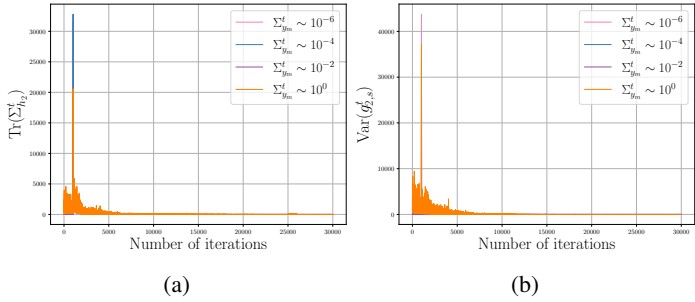

(a)                                           (b)

Figure 8: Uncertainty propagation in the communicated terms during FedGC learning for different regimes of $\Sigma_{y_m}^t$ (a) $\mathrm{Var}\left(g_{m,s}^t\right)$ vs iterations, (b) $\Sigma_{h_m}^t$ vs iterations,

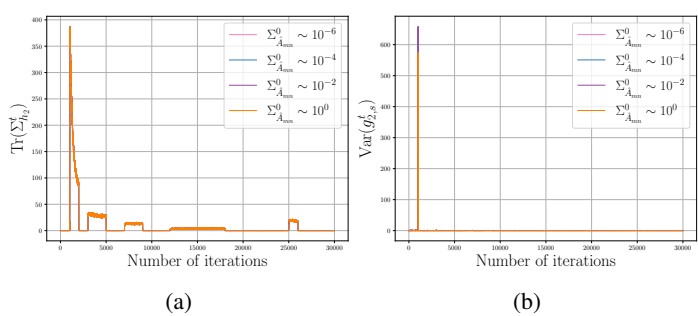

(a)  (b)

Figure 9: Uncertainty propagation in the communicated terms during FedGC learning for different regimes of $\Sigma_{\hat{A}_{mn}^0}$ (a) $\mathrm{Var}\big(g_{m,s}^t\big)$ vs iterations, (b) $\Sigma_{h_m}^t$ vs iterations,

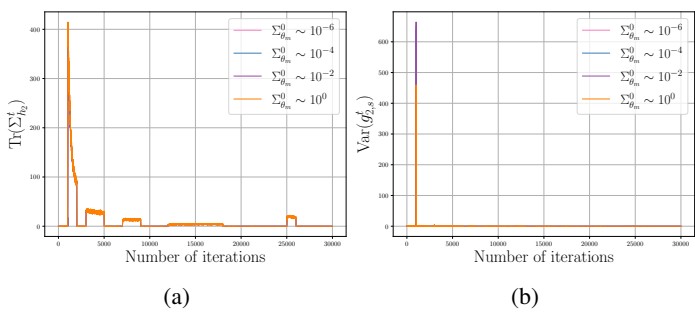

(a)  (b)

Figure 10: Uncertainty propagation in the communicated terms during FedGC learning for different regimes of $\Sigma_{\theta_m^t}$ (a) $\mathrm{Var}\big(g_{m,s}^t\big)$ vs iterations, (b) $\Sigma_{h_m}^t$ vs iterations,

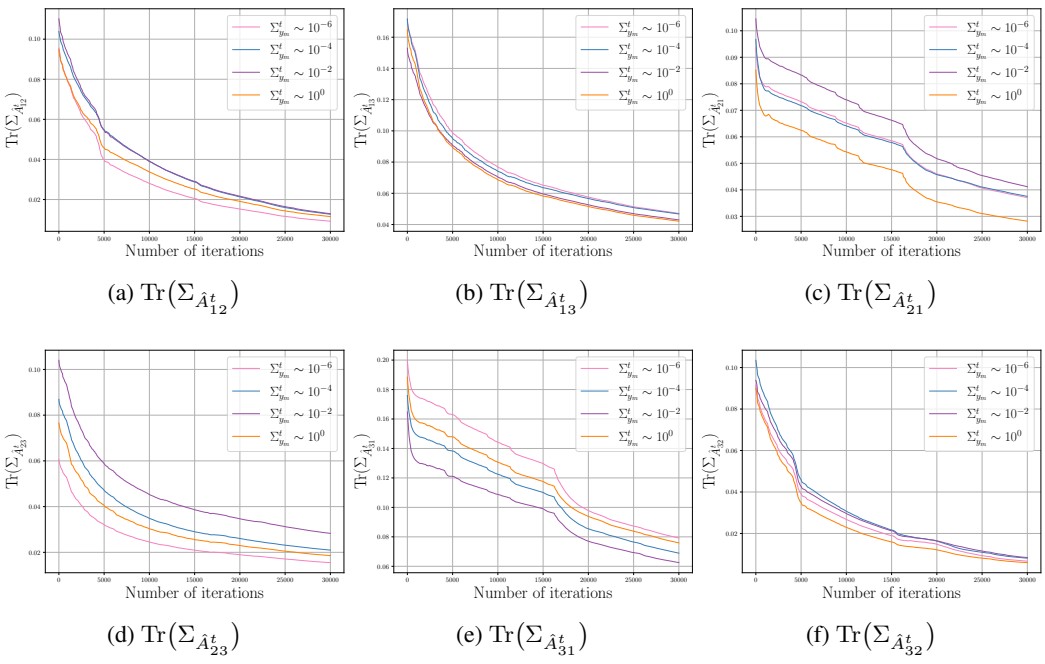

(a) $\mathrm{Tr}\big(\Sigma_{\hat{A}_{12}^t}\big)$  (b) $\mathrm{Tr}\big(\Sigma_{\hat{A}_{13}^t}\big)$  (c) $\mathrm{Tr}\big(\Sigma_{\hat{A}_{21}^t}\big)$

(d) $\mathrm{Tr}\big(\Sigma_{\hat{A}_{23}^t}\big)$  (e) $\mathrm{Tr}\big(\Sigma_{\hat{A}_{31}^t}\big)$  (f) $\mathrm{Tr}\big(\Sigma_{\hat{A}_{32}^t}\big)$

Figure 11: Trace of the covariance for each off-diagonal block of the $A$ matrix during FedGC learning for different regimes of $\Sigma_{y_m}^t$ for HAI dataset: (a) $\mathrm{Tr}(\Sigma_{\hat{A}_{12}^t})$, (b) $\mathrm{Tr}(\Sigma_{\hat{A}_{13}^t})$, (c) $\mathrm{Tr}(\Sigma_{\hat{A}_{21}^t})$, (d) $\mathrm{Tr}(\Sigma_{\hat{A}_{23}^t})$, (e) $\mathrm{Tr}(\Sigma_{\hat{A}_{31}^t})$, (f) $\mathrm{Tr}(\Sigma_{\hat{A}_{32}^t})$ vs. iteration $t$.

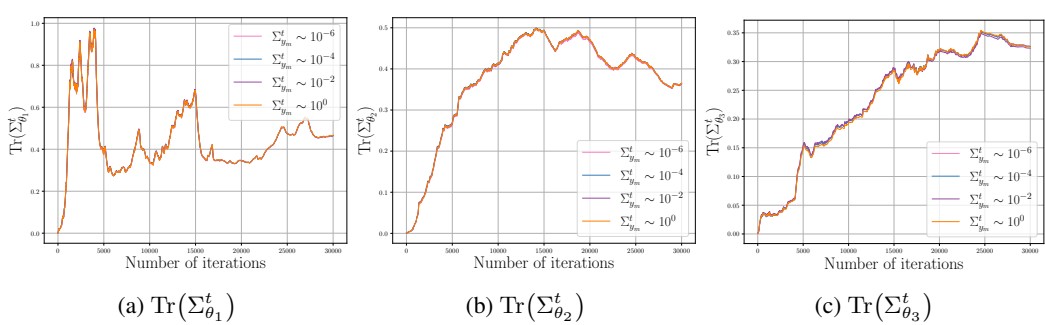

Figure 12: Trace of the covariance of the local model parameters $\theta$ for each of the three components during FedGC learning for different regimes of $\Sigma^t_{y_m}$ for HAI dataset: (a) $\text{Tr}(\Sigma^t_{\theta_1})$, (b) $\text{Tr}(\Sigma^t_{\theta_2})$, (c) $\text{Tr}(\Sigma^t_{\theta_3})$ vs. iteration $t$.

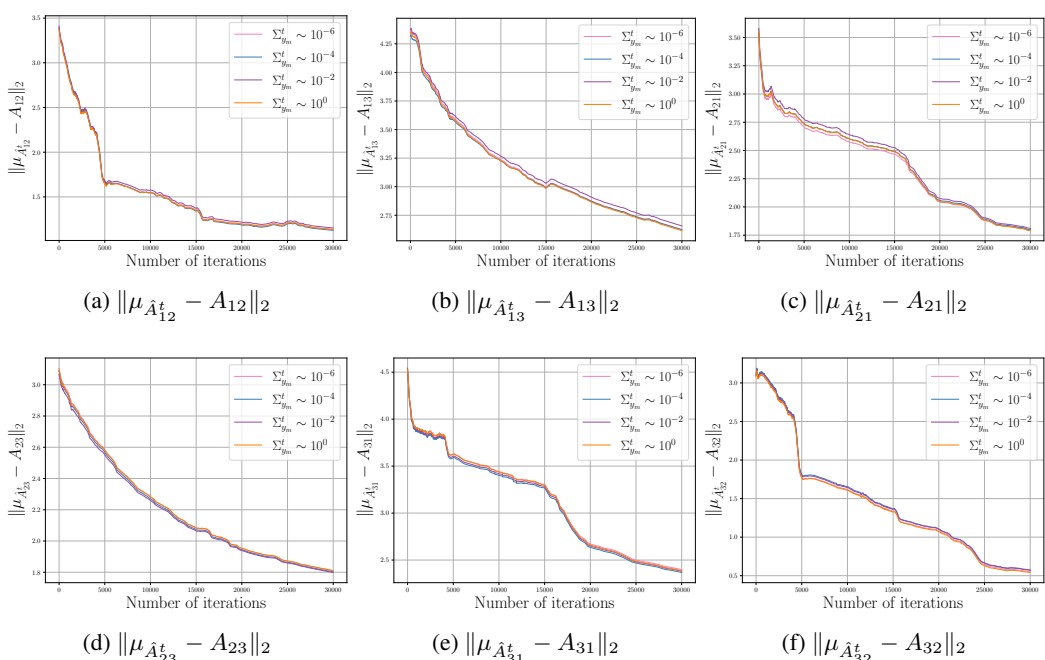

Figure 13: Average $L_2$ norm error of each off-diagonal block of the matrix $A$ during FedGC learning for different regimes of $\Sigma^t_{y_m}$ for HAI dataset: (a) $\|\mu_{\hat{A}^t_{12}} - A_{12}\|_2$, (b) $\|\mu_{\hat{A}^t_{13}} - A_{13}\|_2$, (c) $\|\mu_{\hat{A}^t_{21}} - A_{21}\|_2$, (d) $\|\mu_{\hat{A}^t_{23}} - A_{23}\|_2$, (e) $\|\mu_{\hat{A}^t_{31}} - A_{31}\|_2$, (f) $\|\mu_{\hat{A}^t_{32}} - A_{32}\|_2$ vs. iteration $t$.

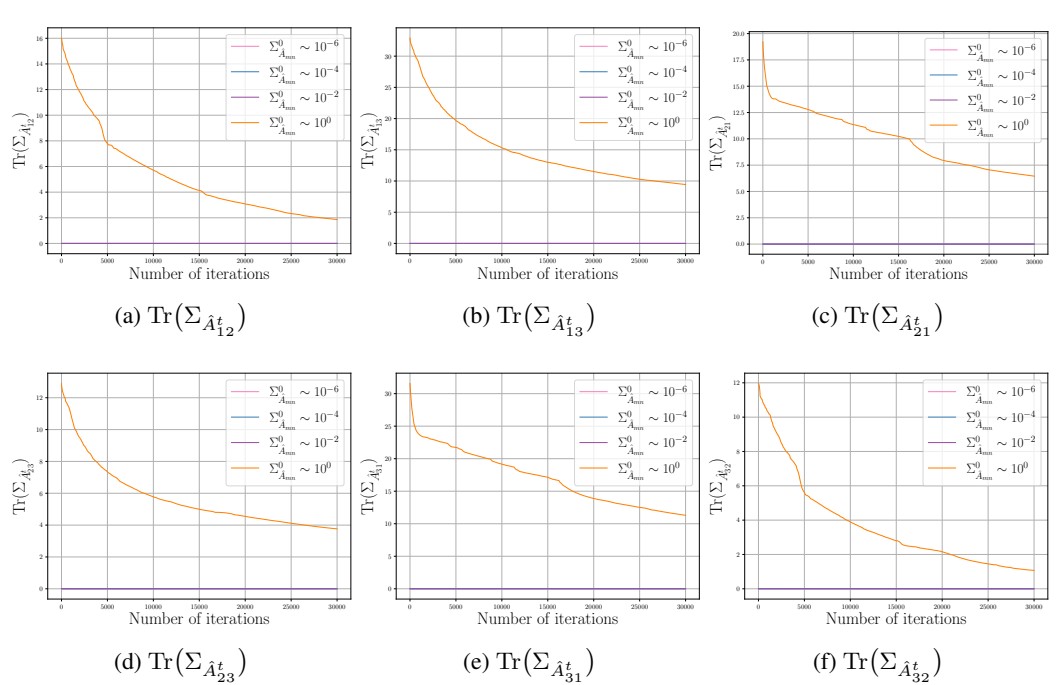

(a) $\mathrm{Tr}\left(\Sigma_{\hat{A}_{12}^t}\right)$      (b) $\mathrm{Tr}\left(\Sigma_{\hat{A}_{13}^t}\right)$      (c) $\mathrm{Tr}\left(\Sigma_{\hat{A}_{21}^t}\right)$

(d) $\mathrm{Tr}\left(\Sigma_{\hat{A}_{23}^t}\right)$      (e) $\mathrm{Tr}\left(\Sigma_{\hat{A}_{31}^t}\right)$      (f) $\mathrm{Tr}\left(\Sigma_{\hat{A}_{32}^t}\right)$

Figure 14: Trace of the covariance for each off-diagonal block of the $A$ matrix during FedGC learning for different regimes of $\Sigma^0_{\hat{A}_{mn}}$ for HAI dataset: (a) $\mathrm{Tr}(\Sigma_{\hat{A}_{12}^t})$, (b) $\mathrm{Tr}(\Sigma_{\hat{A}_{13}^t})$, (c) $\mathrm{Tr}(\Sigma_{\hat{A}_{21}^t})$, (d) $\mathrm{Tr}(\Sigma_{\hat{A}_{23}^t})$, (e) $\mathrm{Tr}(\Sigma_{\hat{A}_{31}^t})$, (f) $\mathrm{Tr}(\Sigma_{\hat{A}_{32}^t})$ vs. iteration $t$.

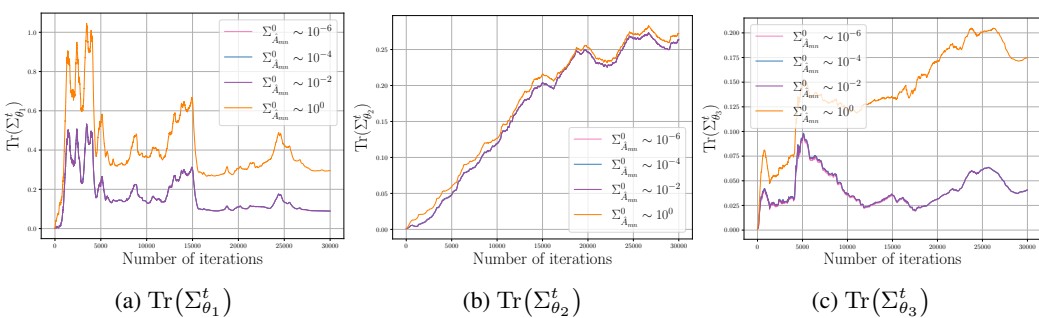

(a) $\mathrm{Tr}\left(\Sigma_{\theta_1}^t\right)$      (b) $\mathrm{Tr}\left(\Sigma_{\theta_2}^t\right)$      (c) $\mathrm{Tr}\left(\Sigma_{\theta_3}^t\right)$

Figure 15: Trace of the covariance of the local model parameters $\theta$ for each of the three components during FedGC learning for different regimes of $\Sigma^0_{\hat{A}_{mn}}$ for HAI dataset: (a) $\mathrm{Tr}(\Sigma_{\theta_1}^t)$, (b) $\mathrm{Tr}(\Sigma_{\theta_2}^t)$, (c) $\mathrm{Tr}(\Sigma_{\theta_3}^t)$ vs. iteration $t$.

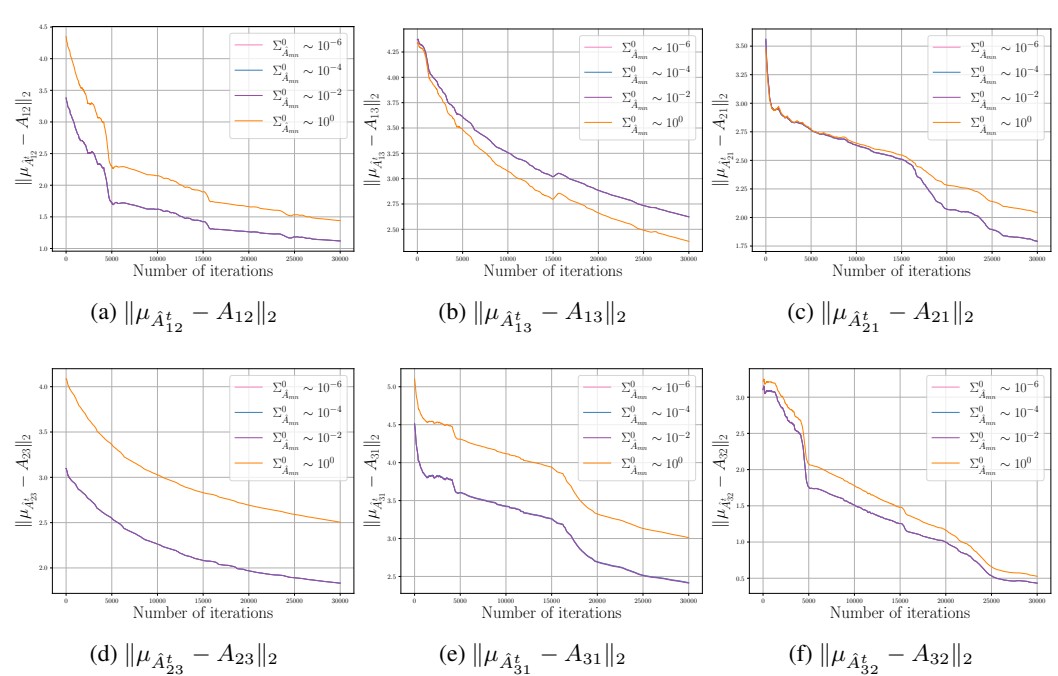

Figure 16: Average $L_2$ norm error of each off-diagonal block of the matrix $A$ during FedGC learning for different regimes of $\Sigma^0_{\hat{A}_{mn}}$ for HAI dataset: (a) $\|\mu_{\hat{A}^t_{12}} - A_{12}\|_2$, (b) $\|\mu_{\hat{A}^t_{13}} - A_{13}\|_2$, (c) $\|\mu_{\hat{A}^t_{21}} - A_{21}\|_2$, (d) $\|\mu_{\hat{A}^t_{23}} - A_{23}\|_2$, (e) $\|\mu_{\hat{A}^t_{31}} - A_{31}\|_2$, (f) $\|\mu_{\hat{A}^t_{32}} - A_{32}\|_2$ vs. iteration $t$.

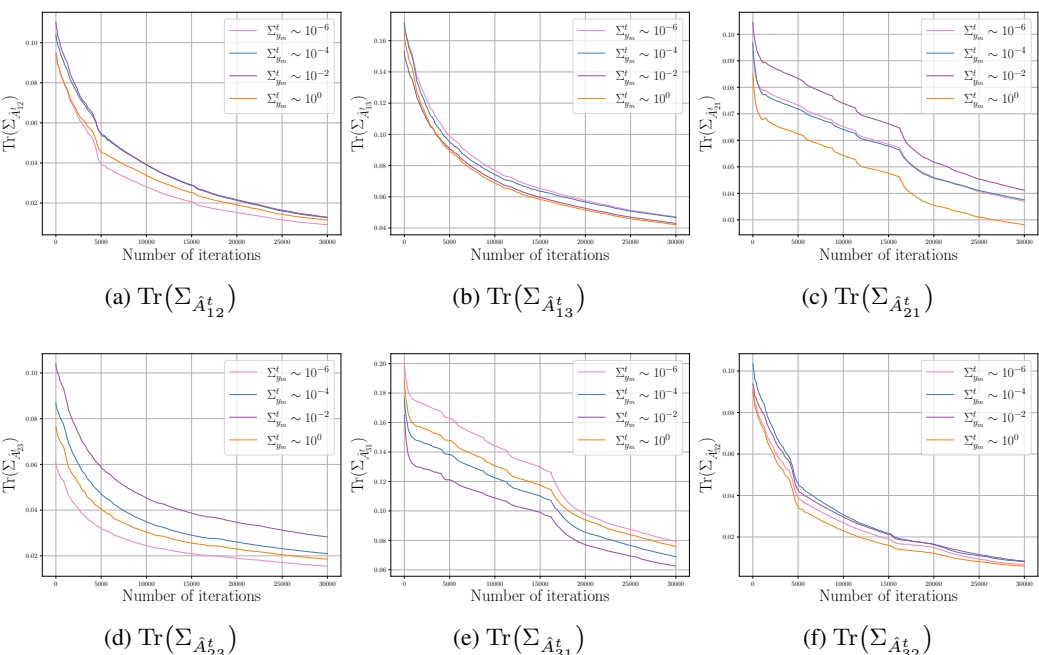

Figure 17: Trace of the covariance for each off-diagonal block of the $A$ matrix during FedGC learning for different regimes of $\Sigma^0_{\theta_m}$ for HAI dataset: (a) $\mathrm{Tr}(\Sigma_{\hat{A}^t_{12}})$, (b) $\mathrm{Tr}(\Sigma_{\hat{A}^t_{13}})$, (c) $\mathrm{Tr}(\Sigma_{\hat{A}^t_{21}})$, (d) $\mathrm{Tr}(\Sigma_{\hat{A}^t_{23}})$, (e) $\mathrm{Tr}(\Sigma_{\hat{A}^t_{31}})$, (f) $\mathrm{Tr}(\Sigma_{\hat{A}^t_{32}})$ vs. iteration $t$.

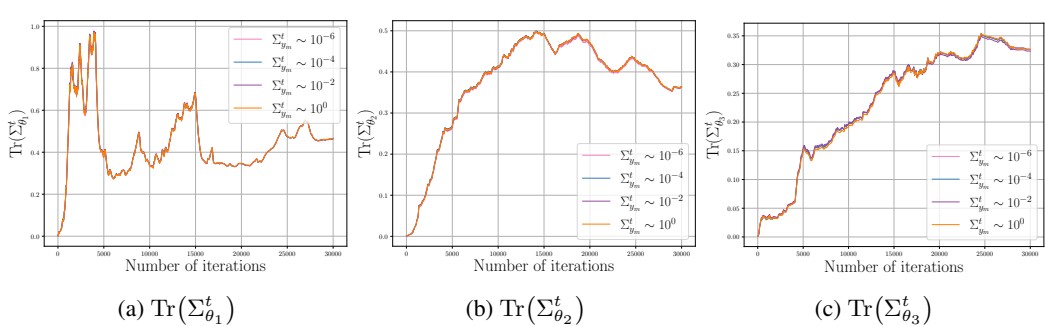

Figure 18: Trace of the covariance of the local model parameters $\theta$ for each of the three components during FedGC learning for different regimes of $\Sigma_{\theta_m}^0$ for HAI dataset: (a) $\mathrm{Tr}(\Sigma_{\theta_1}^t)$, (b) $\mathrm{Tr}(\Sigma_{\theta_2}^t)$, (c) $\mathrm{Tr}(\Sigma_{\theta_3}^t)$ vs. iteration $t$.

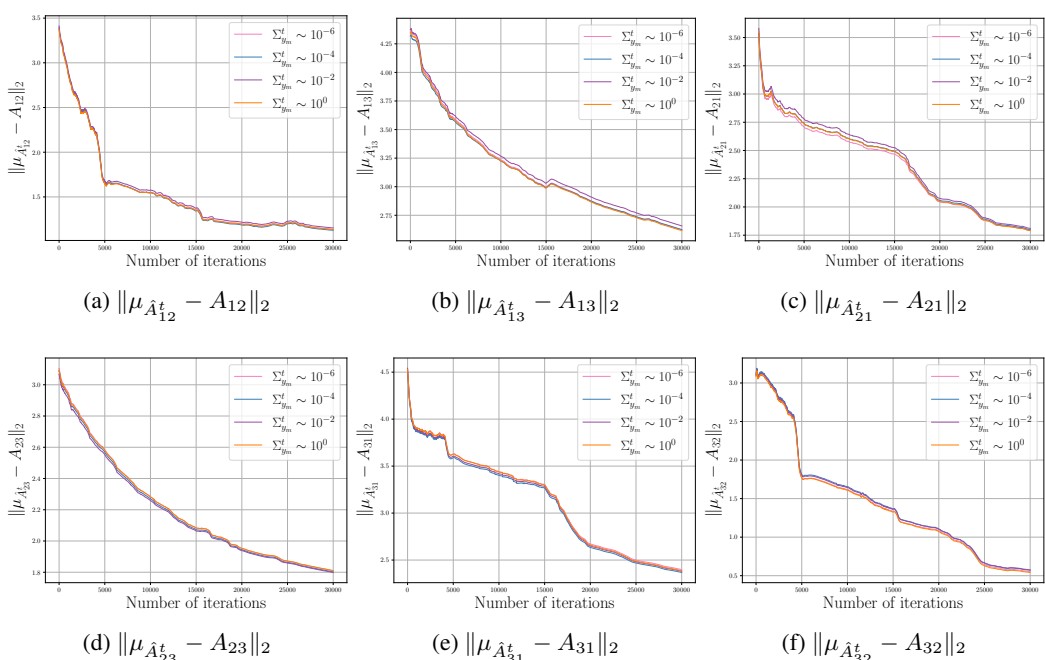

Figure 19: Average $L_2$ norm error of each off-diagonal block of the matrix $A$ during FedGC learning for different regimes of $\Sigma_{\theta_m}^0$ for HAI dataset: (a) $\|\mu_{\hat{A}_{12}^t} - A_{12}\|_2$, (b) $\|\mu_{\hat{A}_{13}^t} - A_{13}\|_2$, (c) $\|\mu_{\hat{A}_{21}^t} - A_{21}\|_2$, (d) $\|\mu_{\hat{A}_{23}^t} - A_{23}\|_2$, (e) $\|\mu_{\hat{A}_{31}^t} - A_{31}\|_2$, (f) $\|\mu_{\hat{A}_{32}^t} - A_{32}\|_2$ vs. iteration $t$.

Figure 20: Trace of the covariance for each off-diagonal block of the $A$ matrix during FedGC learning for different regimes of $\Sigma_{y_m}^t$ on SWaT dataset, plotted vs. iteration $t$.

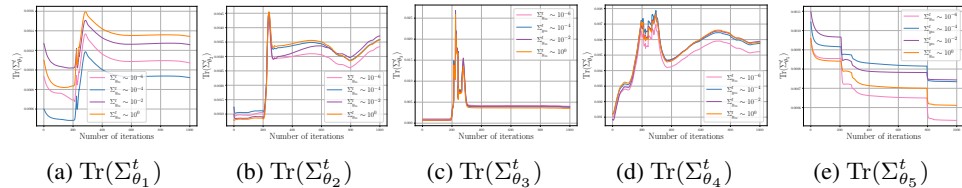

(a) $\mathrm{Tr}(\Sigma_{\theta_1}^t)$    (b) $\mathrm{Tr}(\Sigma_{\theta_2}^t)$    (c) $\mathrm{Tr}(\Sigma_{\theta_3}^t)$    (d) $\mathrm{Tr}(\Sigma_{\theta_4}^t)$    (e) $\mathrm{Tr}(\Sigma_{\theta_5}^t)$

Figure 21: Trace of the covariance of the local model parameters $\theta_m$ for each of the five components during FedGC learning for different regimes of $\Sigma_{y_m}^t$ on SWaT dataset, plotted vs. iteration $t$.

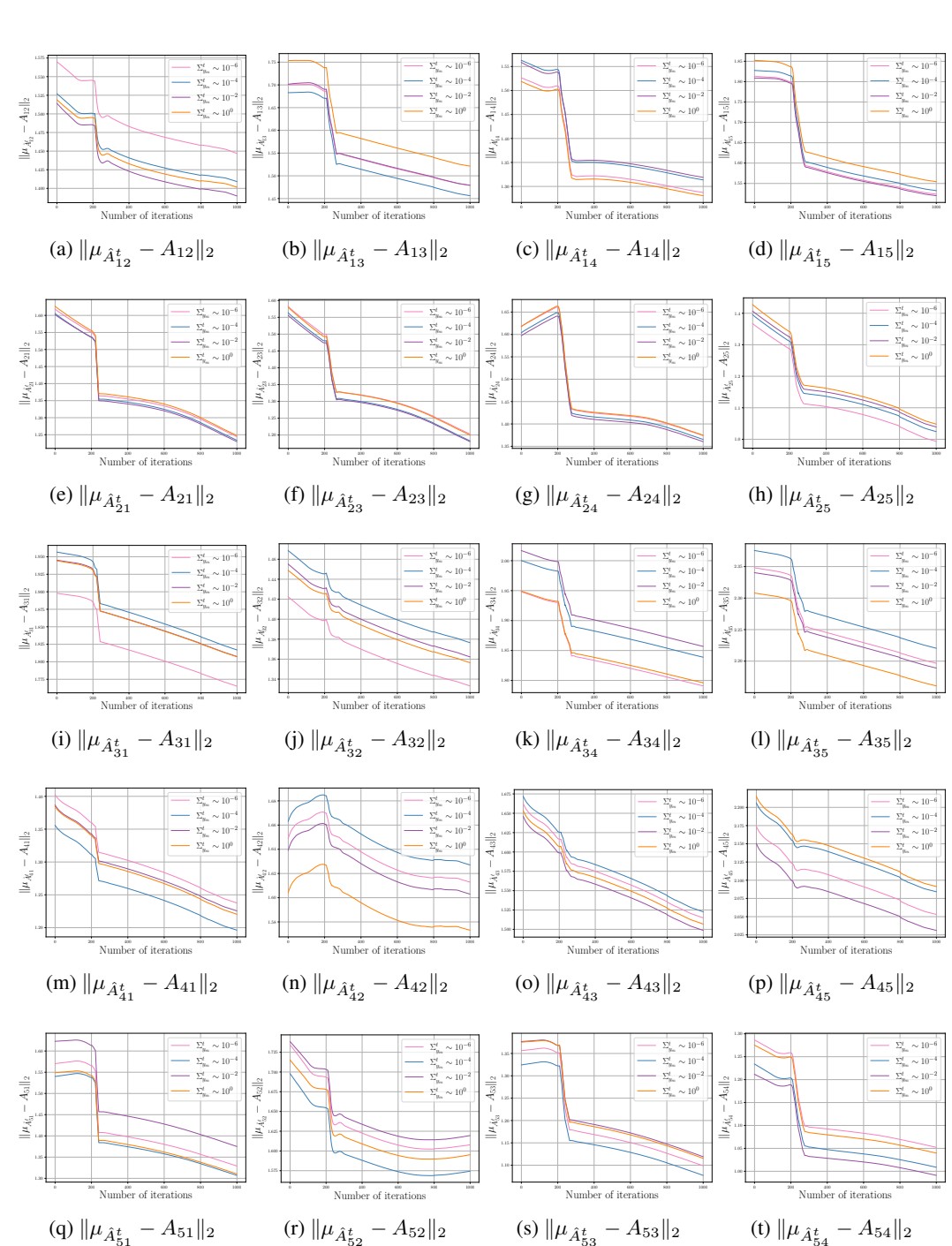

Figure 22: Average $L_2$ norm error of each off-diagonal block of the $5 \times 5$ matrix $A$ during FedGC learning for different regimes of $\Sigma_{y_m}^t$ on SWaT dataset, plotted vs. iteration $t$.

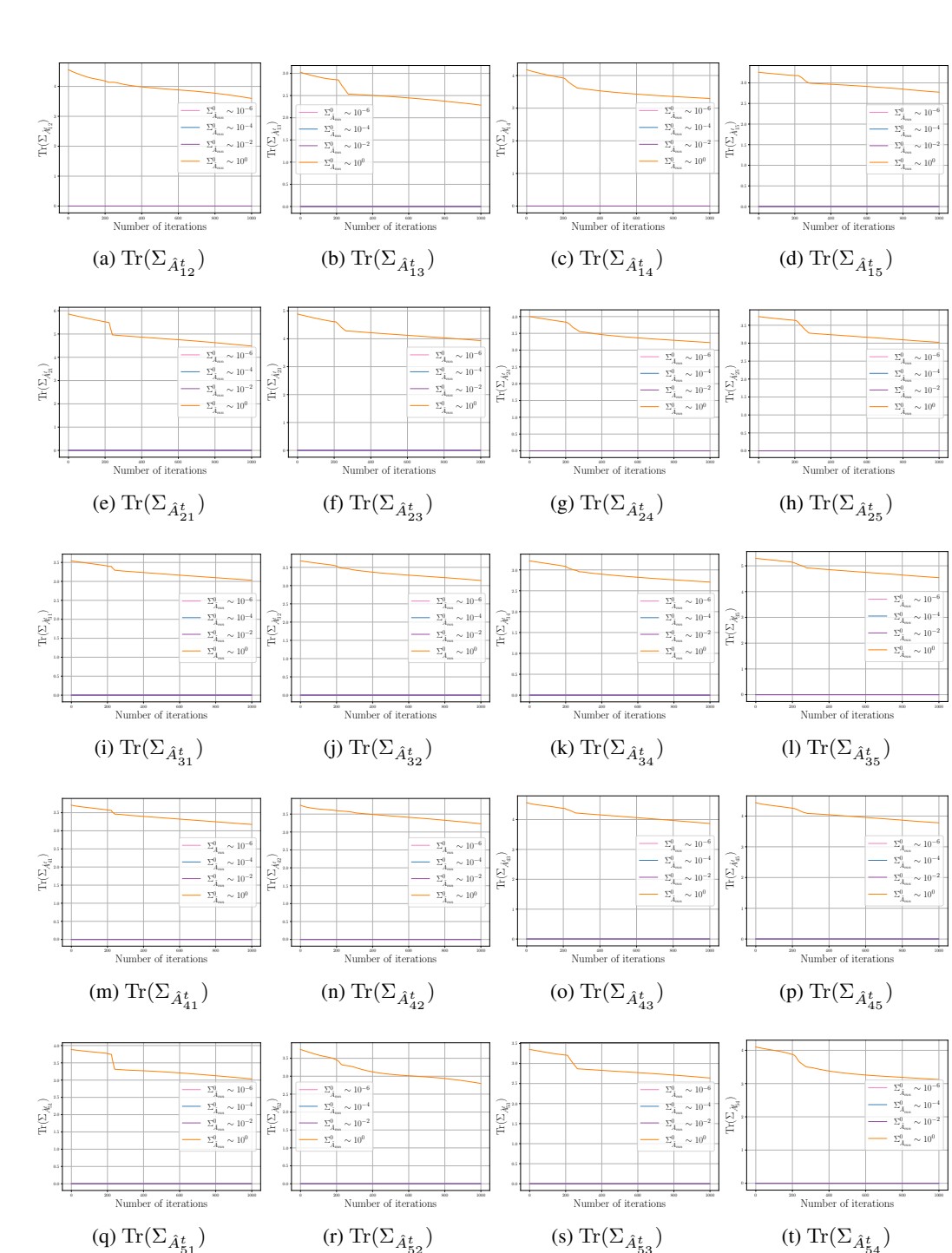

Figure 23: Trace of the covariance for each off-diagonal block of the $A$ matrix during FedGC learning for different regimes of $\Sigma^0_{\hat{A}_{mn}}$ on SWaT dataset, plotted vs. iteration $t$.

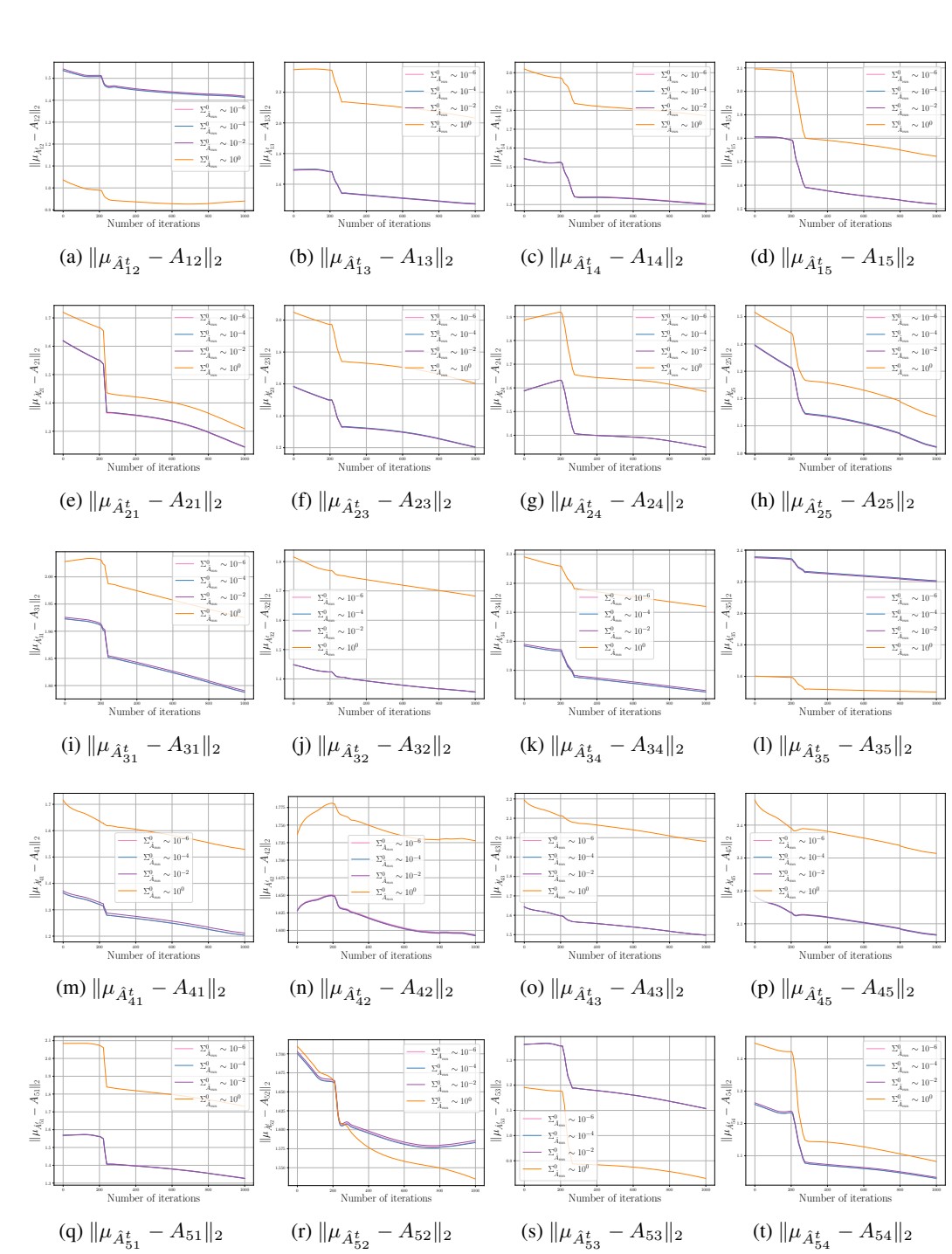

Figure 24: Average $L_2$ norm error of each off-diagonal block of the $A$ matrix during FedGC learning for different regimes of $\Sigma^0_{\hat{A}_{mn}}$ on SWaT dataset, plotted vs. iteration $t$.

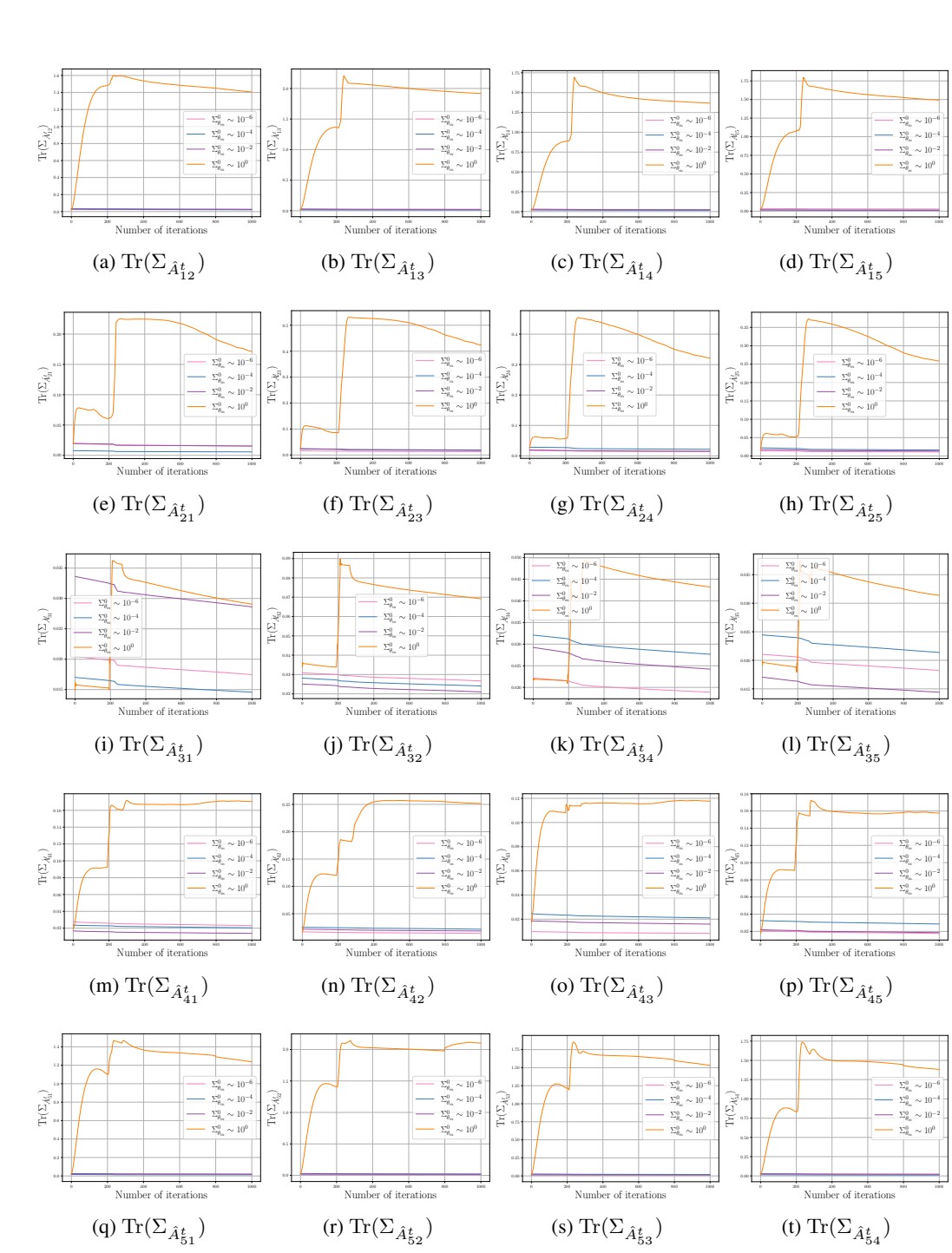

Figure 25: Trace of the covariance for each off-diagonal block of the $A$ matrix during FedGC learning for different regimes of $\Sigma_{\theta_m}^0$ on SWaT dataset, plotted vs. iteration $t$.

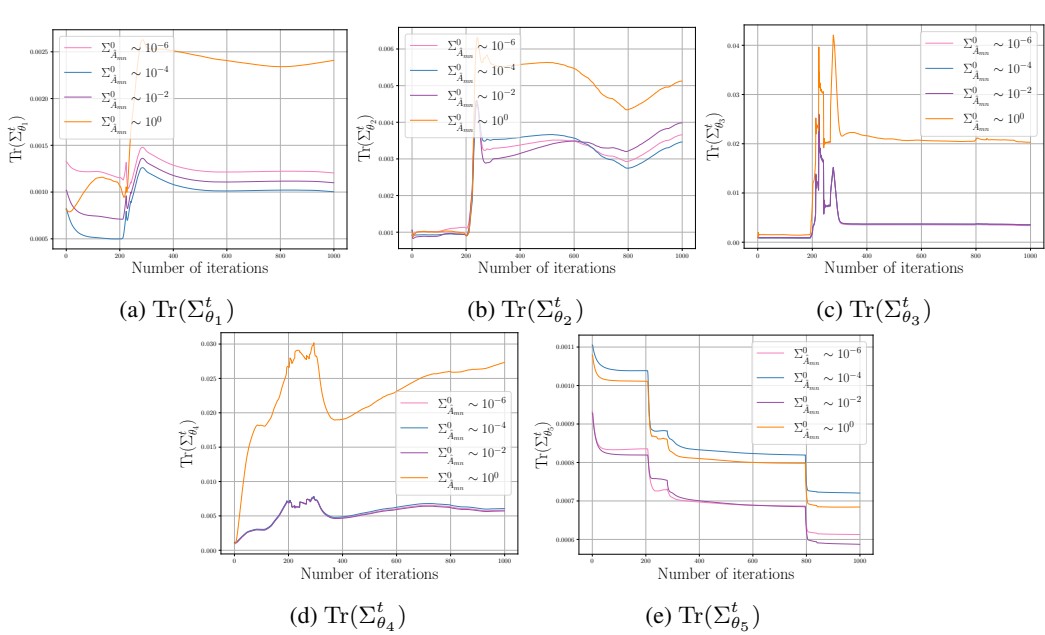

Figure 26: Trace of the covariance of the local model parameters $\theta$ for each of the five components during FedGC learning for different regimes of $\Sigma^0_{\hat{A}_{mn}}$ on SWaT dataset, plotted vs. iteration $t$.

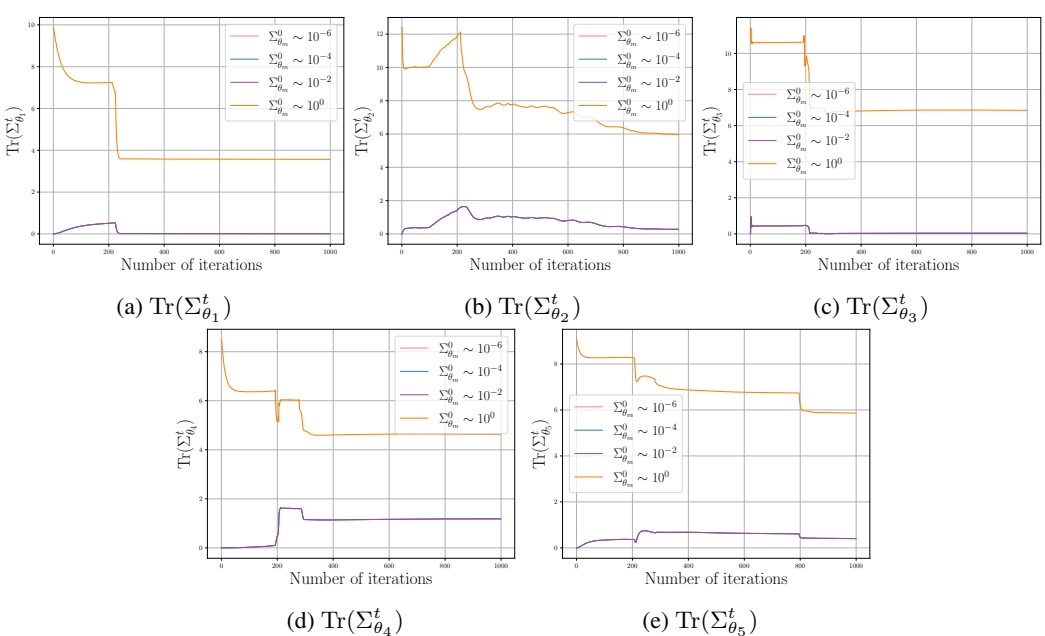

Figure 27: Trace of the covariance of the local model parameters $\theta$ for each of the five components during FedGC learning for different regimes of $\Sigma^0_{\theta_m}$ on SWaT dataset, plotted vs. iteration $t$.

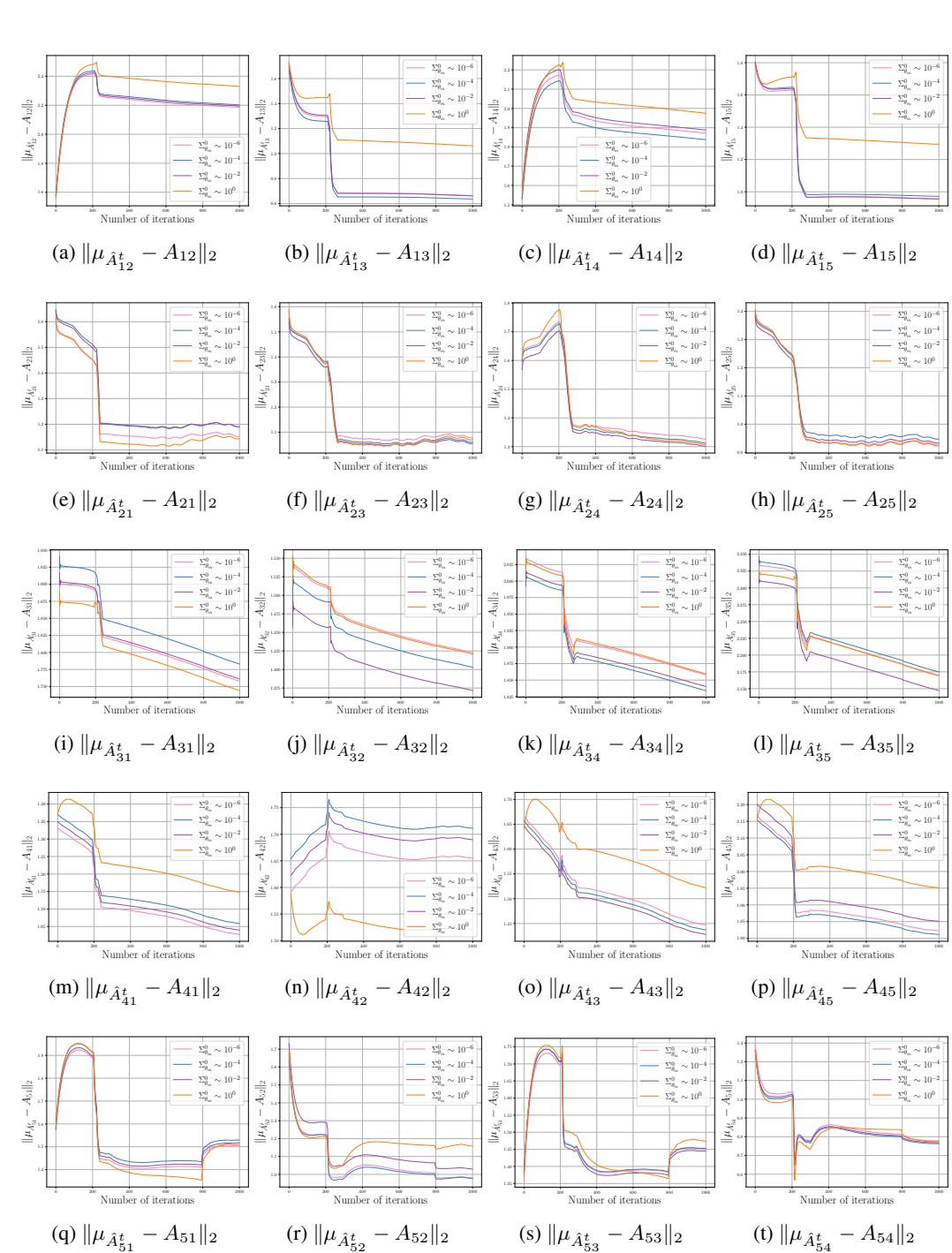

Figure 28: Average $L_2$ norm error of each off-diagonal block of the $A$ matrix during FedGC learning for different regimes of $\Sigma_{\theta_m}^0$ on SWaT dataset, plotted vs. iteration $t$.