# OpenReview forum: "Uncertainty in Federated Granger Causality: From Origins to Systemic Consequences"
_ICLR.cc/2026/Conference — Submitted to ICLR 2026_

### Official Review · Reviewer_4bfG · 2025-10-27

**Soundness:** 2
**Presentation:** 2
**Contribution:** 3
**Rating:** 2
**Confidence:** 3

**Summary:**

1.The manuscript puts forward a systematic classification scheme for uncertainty sources in federated causal learning, which effectively distinguishes between the impacts of data noise and model variability—a foundational step for clarifying uncertainty components in this paradigm.

2.It successfully derives closed-form propagation expressions that trace how uncertainties propagate through client-server, server-client communication channels, and internal processing paths of both entities. Notably, these expressions uncover four previously unrecognized cross-covariance terms, which reveal the coupling relationships between data and parameters within the FedGC framework.

3.The authors establish spectral-radius-based convergence conditions that ensure the convergence of all covariance recursions. This not only provides theoretical guarantees for uncertainty evolution stability but also enables the derivation of explicit solutions for the steady-state variances of server-side global models and client-side local models.

4.Theoretical analyses on convergence yield a key finding: the steady-state variance of the FedGC framework relies exclusively on the raw data statistics of clients, thereby eliminating the influence of initial epistemic uncertainty priors—a result that simplifies practical uncertainty quantification.

**Strengths:**

1.The paper identifies a meaningful gap in existing FedGC research and links it to real-world consequences, highlighting the problem’s practical relevance.

2.It provides a systematic classification of uncertainty sources in FedGC and derives closed-form recursions for uncertainty propagation, which could serve as a baseline for future work on linear FedGC systems.

3.The use of both synthetic and real-world industrial datasets is a strength, as it grounds theoretical claims in practical scenarios relevant to the paper’s target domain.

**Weaknesses:**

1.The paper’s core convergence results depend on strict stationarity of client data and zero process noise. In practice, industrial time-series data are often non-stationary and contain process noise. The paper’s brief discussion of relaxing stationarity is insufficient—no formal bounds on convergence or experimental validation of non-stationary data are provided, limiting the theory’s real-world applicability.

2.Without comparing to centralized GC, vanilla FedGC, or Bayesian FedGC methods, the paper cannot prove that its uncertainty modeling improves performance. For example, it is unclear whether the observed “faster variance decay with higher aleatoric noise” is unique to the proposed method or a general property of FedGC.

3.Real-world experiments focus on parameter variance and L2 error but omit metrics critical for causal inference. This makes it impossible to assess if uncertainty quantification actually enhances “reliability and interpretability” in practice.

4.Synthetic experiments use only 2 clients with fixed pm=2 and dm=8 . Scalability studies only vary dm, failing to validate the method for large federated systems, a key use case for FedGC.

5.The paper’s “novel cross-covariance components” are a natural extension of covariance propagation in federated systems. There is no new algorithmic innovation that distinguishes it from prior work.

**Questions:**

1.Your convergence results depend on strict data stationarity. For non-stationary industrial data, how would your uncertainty recursions behave? Do you have formal bounds on convergence or experimental results for non-stationary data to support the claim that steady-state variances remain independent of initial priors?

2.Why did you not compare your method to baseline approaches? Without these comparisons, it is impossible to determine if your uncertainty modeling provides unique value. For example, could vanilla FedGC with simple variance estimation achieve similar or better performance?

3.Appendix D.4 proposes a Gaussian mechanism for DP, but no experimental results are provided. What ε,δ values did you use, and how did DP noise affect: (a) Steady-state variances of server/client parameters; (b) Accuracy of causal link detection? Do you have evidence that your method balances privacy and utility better than existing DP-FedGC approaches?

4.Your synthetic experiments use 2 clients, and scalability studies only vary dm​. For federated systems with 50+ clients, how does the computational complexity of your covariance propagation scale? Do you have plans to optimize this?

5.You claim your method enables “robust root-cause analysis”. Can you provide experimental results for a simulated fault scenario where your uncertainty-aware FedGC outperforms vanilla FedGC in: (a) Detecting the fault’s root cause; (b) Providing early warnings via confidence intervals?

---

> ### Author Response · Authors · 2025-11-27
> **Weakness 1 (Stationarity and Zero Process Noise)**
>
> # Stationarity
> In the revised version of the manuscript, **Section E.2 (Appendix)** now provides an extended discussion of how our uncertainty–propagation framework can be adapted to non-stationary settings.
>
> Our analysis uses the classical assumption of weak stationarity because it enables closed-form propagation of uncertainty through the FedGC updates. However, the same recursions remain structurally valid when the empirical moments are replaced by \emph{exponentially weighted} (EWMA) moments. In this setting, each client updates
>
> $\mu_t = (1-\lambda)\mu_{t-1} + \lambda x_t,
> \qquad
> \Sigma_t = (1-\lambda)\Sigma_{t-1} + \lambda x_t x_t^\top,
> $
>
> with forgetting factor $0<\lambda<1$. When the underlying data exhibit slowly drifting moments $(\mu_t^*,\Sigma_t^*)$, standard results imply the following,
>
> $
> \|\mu_t - \mu_t^* \| = O(\delta/\lambda), \qquad
> \|\Sigma_t - \Sigma_t^* \| = O(\delta/\lambda),
> $
>
> where $\delta$ bounds the temporal drift, and $(\mu_t^*,\Sigma_t^*)$ are the true non-stationary moments of the underlying data. The above bounds show that EWMA can track the true moment as long as the true moment is not drifting faster than EWMA forgets older data. One main advantage of using this technique is that EWMA moments track non-stationarity while preserving the form of our theoretical results in uncertainty propagations. Inspired by several works in the machine-learning literature that handle non-stationarity through exponential weighting or forgetting factors [1-3], we adopt the same principle here.
>
> Beyond EWMA, other forms of non-stationarity can be incorporated by modifying the moment-estimation step. Examples include sliding-window estimators, seasonally adjusted or periodic-window estimators, trend-filtered or total-variation–regularized moment updates, and online convex-combination estimators for abrupt regime changes. Neural-network–based models can also accommodate complex non-stationarity, but doing so would require deriving a new set of uncertainty–propagation equations beyond the linear FedGC framework. A systematic treatment of these extensions is an important direction for future work.
>
> **References:**
>
> **[1]** Bozkurt, B., Pehlevan, C. and Erdogan, A., 2022. Biologically-plausible determinant maximization neural networks for blind separation of correlated sources. Advances in Neural Information Processing Systems, 35, pp.13704-13717.
>
> **[2]** Kurle, R., Cseke, B., Klushyn, A., Van Der Smagt, P. and Günnemann, S., 2019. Continual learning with bayesian neural networks for non-stationary data. In International Conference on Learning Representations.
>
> **[3]** Wang, T., Zhang, W., Ye, C., Wei, J., Zhong, H. and Huang, T., 2015. FD4C: Automatic fault diagnosis framework for web applications in cloud computing. IEEE Transactions on Systems, Man, and Cybernetics: Systems, 46(1), pp.61-75.
>
> # Zero Process Noise
> We agree that assuming process noise $Q=0$ is a simplifying choice. In our formulation, this assumption allows us to interpret aleatoric uncertainty purely as measurement (sensor) noise and to isolate the contribution of data variability in a transparent manner. If the underlying state evolution is itself noisy, i.e., $Q \neq 0$ then the aleatoric component would naturally include \emph{both} measurement noise and state-process noise. The propagation framework we derive remains valid in this case; the only additional engineering challenge lies in estimating $Q$ in a decentralized setting.
>
> Addressing the non-zero process noise case (i.e., $Q \neq 0$) would require extending FedGC, which currently estimates only the state-transition matrix $A$, to also estimate the state-process covariance $Q$ across clients. This is technically feasible but adds another layer of decentralized system identification. We view this as an important direction for future work.

---

> ### Author Response · Authors · 2025-11-27
> **Weakness 2**
>
> We thank the reviewer for this question. In **Section B.3 of the Appendix**, we now include quantitative comparisons to several natural baselines.
>
> For comparison, we include the following baselines:
>
>  **(1) Centralized GC:** The high-dimensional measurement data $y^t$ from all clients are pooled and $A_{21}$ is estimated centrally.
>
> **(2) Our:** Our method corresponds to ensemble FedGC (multi-start), where each run begins from an independent prior sample and the empirical covariance provides the uncertainty estimate.
>
> **(3) Independent Clients:** Obtained by setting $\eta_2 = 0$ in Eq.~(5), so each client ignores cross-dependencies and updates its block independently.
>
> Regarding the suggestion to compare with *Bayesian FedGC*, we respectfully disagree with the premise.
>
> 1. To the best of our knowledge, *no Bayesian formulation of FedGC exists in the literature*. There is no established architecture or algorithm for Bayesian FedGC against which our method could be compared.
>
> 2. Existing works on Bayesian learning in federated settings all focus on horizontal FL, where the likelihood factorizes across clients. These approaches cannot be adapted to the vertical FedGC regime without redesigning the model and the inference procedure entirely, since FedGC couples client and server parameters through a shared dynamical system.
>
> 3. One of the attempt at Bayesian inference for vertical FL is the recent, non-peer-reviewed preprint [1]. This preprint study generalized regression models in vertical FL and performed approximate Bayesian inference. The other is a published study [2] which we cited in our **Section 2**, that does vertical FL on static data without any temporal dynamics. In contrast, FedGC is a state-space Granger-causal model with a well-defined temporal dynamics that enables tightly coupled client--server parameter blocks. These methods have no notion of causal structure, temporal evolution, or state-transition matrices, and do not propagate uncertainty through iterative alternating minimization as FedGC does. Therefore, adapting their approach to FedGC would require building an entirely new Bayesian state-space inference framework that handles these differences. This would amount to creating a completely new algorithm unrelated to FedGC.
>
> **References:**
>
> **[1]** Hassan, Conor, Matthew Sutton, Antonietta Mira, and Kerrie Mengersen. "Scalable vertical federated learning via data augmentation and amortized inference." arXiv preprint arXiv:2405.04043 (2024).
>
> **[2]** Van Daalen, Florian, et al. "VertiBayes: learning Bayesian network parameters from vertically partitioned data with missing values." Complex & Intelligent Systems 10.4 (2024): 5317-5329.

---

> ### Author Response · Authors · 2025-11-27
> **Weaknesses 3 & 4**
>
> # Weakness 3
> Our paper is a theoretical analysis of how uncertainty would propagate in FedGC if the exchanged quantities were treated as stochastic. Vanilla FedGC is a state-space Granger-causal method whose goal is to estimate the state-transition matrix $A$ (or, equivalently, the GC coefficients). In the Granger-causality literature, these parameters fully determine the causal relations: $i$ Granger-causes $j$ if and only if the corresponding entries of $A$ are nonzero. Thus, understanding the reliability of Granger causality directly reduces to understanding the uncertainty of these parameters. Accordingly, our contribution derives closed-form uncertainty–propagation recursions, showing how uncertainty in the estimated $A$ evolves in time across rounds and in location across client and server blocks, and we characterize its sources and steady-state limits. This analysis assesses the reliability of Granger-causal parameters themselves. Similarly, $L_2$ scores analyze the effect of learning accurate Granger causal parameters.
>
> We believe this comment stems from the confusion the reviewer might have with the structural-causal literature, which often uses metrics such as SHD or ATE. These do not apply here, because Granger causality is a predictive notion defined entirely by the dynamics $x_{t+1} = A x_t$ rather than by interventional or counterfactual semantics.
>
> # Weakness 4
> As detailed in **Section B.2**, our scalability study varies both the feature dimension $d_m$ and the number of clients $M$. In particular, \textbf{Table~3} reports results for $M \in \{2,4,8,16\}$, covering four powers of two. During experimentation, configurations with $M \ge 2^{5} = 32$ clients exceeded the available memory and resulted in system crashes at the time of submission. Due to this computational constraint, we limited our experiments to $M \le 16$. Nevertheless, the trend across these experiments is clear and consistent:
>
> 1. For low prior variance regimes, increasing the number of clients has minimal effect on $\mathrm{tr}(\Sigma_{A})$, and the propagated uncertainty converges to similar values for all tested $M$.
>
> 2. For high prior variance regimes, the uncertainty growth diverges similarly across all client counts, with $\mathrm{tr}(\Sigma_{A})$ increasing at comparable rates.
>
> 3. Across all experiments, scaling $M$ does not introduce qualitatively new behaviors; the uncertainty curves follow stable and predictable patterns.
>
> We respectfully direct the reviewer to **Table 3**, which illustrates these scaling trends clearly. Although we could not run experiments with $M \ge 32$ due to hardware limitations, the results for $M \in \{2,4,8,16\}$ already show how covariance propagation behaves as the number of clients increases.

---

> ### Author Response · Authors · 2025-11-27
> **Weakness 5**
>
> We respectfully disagree with the reviewer’s assertion that our cross-covariance components are a “natural extension” of prior work on covariance propagation in federated systems. To the best of our knowledge, there is no prior literature either in horizontal or vertical federated learning that develops any form of covariance propagation through the federated optimization dynamics. In particular, no existing work analyzes how uncertainty evolves over time (across rounds), across locations (client $\leftrightarrow$ server), or across model components that become statistically dependent due to the iterative FedGC updates. Our contributions are fundamentally new in several respects:
>
> 1. We are not aware of any prior work that derives closed-form recursions for the propagation of parameter covariances in a federated optimization algorithm, nor any work that characterizes the induced cross-covariances between client and server parameters.
>
> 2. Traditional FL focuses on horizontal data partitioning; FedGC is vertically federated, where the server and clients jointly estimate different components of a coupled linear model. The resulting interdependencies produce cross-covariance structures that **do not arise** in standard FL, and we derive them for the first time.
>
> 3. Our theoretical results explicitly characterize how covariances evolve:
> **(a)** temporally, across federated rounds, and **(b)** spatially, between the client and server parameter blocks,
> capturing novel statistical dependencies introduced by the FedGC update rule.
>
> 4. Prior work on Bayesian FL provides uncertainty at the model level, not at the level of inter-block cross-covariance recursions. None of these works derive the implicit coupling structure that arises from alternating minimization across distributed parameter blocks, primarily due to its development for horizontal federated learning systems.
>
> While our derivations may appear as a natural extension once presented in closed form, the absence of any prior covariance-propagation framework for federated learning (and especially for vertically federated causal models) highlights the innovation of this work.

---

> ### Author Response · Authors · 2025-11-27
> **Answers to the Questions**
>
> # Question 1
> Please refer to our response to **Weakness 1** above
>
> # Question 2
> Please refer to our response to **Weakness 2** above
>
> # Question 3
> In this version of the paper (as well as the previous version), the **Appendix D.4** describes how one may calibrate the Gaussian mechanism to a target
> $(\epsilon,\delta)$ via the standard relation
>
> $
> \sigma \;\ge\; \frac{\Delta}{\epsilon}\sqrt{2\log(1.25/\delta)},
> $
>
> where $\Delta$ is the $\ell_2$-sensitivity of the communicated quantity.
>
> During the course of this rebuttal, we perfomred additional experiments to report DP privacy-utility results in **Appendix B.4**. However, we do not fix a particular
> $(\epsilon,\delta)$ pair. Instead, we sweep the injected noise level directly,
>
> $
> \sigma \in \{10^{-6},10^{-5},\dots,10^{-1}\},
> $
>
> for both client$\to$server and server$\to$client messages.
> This choice allows us to study robustness to perturbations independently of any
> deployment-specific sensitivity bound.
> Given any desired $(\epsilon,\delta)$ and sensitivity $\Delta$, the corresponding
> $\sigma$ can be obtained from the formula above; conversely, each $\sigma$ in our
> sweep induces an implied~$\epsilon$.
>
> **(a) Steady-state variances.**
> Injecting Gaussian noise with variance $\sigma^2$ and re-running the multi-start
> experiments shows the behavior predicted by our theory: the uncertainty recursions
> remain stable for all tested $\sigma$, and the steady-state traces
> $\mathrm{tr}(\Sigma_A)$ and $\mathrm{tr}(\Sigma_\theta)$ increase smoothly as
> $\sigma$ grows.  For $\sigma \le 10^{-3}$, this inflation is modest; only the
> largest noise levels produce substantial variance growth.
>
> **(b) Causal link detection.**
> We report $\|\hat A - A\|_F$ and support recovery as functions of $\sigma$.
> Both metrics remain close to the non-DP baseline for small and moderate noise, with
> noticeable degradation only at the upper end of the sweep, mirroring the rise in
> parameter variance.
>
> **Privacy--utility comparison.**
> Our DP mechanism applies the same Gaussian noise used in DP-FedGC to the original
> FedGC representations and extends it to the additional cross-covariance quantities
> introduced by our uncertainty analysis.
> When matched at the same $\sigma$, the incremental utility loss from privatizing these additional statistics is negligible.
>
> # Question 4
>   Please refer to our response to **Weakness 4** above
>
> # Question 5
> A direct comparison to vanilla FedGC is inherently trivial. Vanilla FedGC provides only point estimates of the Granger coefficients and does not produce any measure of uncertainty, confidence intervals, or variance evolution. Our method is a stochastic formulation of FedGC together with closed-form recursions that characterize how uncertainty propagates in time and between the client and server parameter blocks. As a consequence, vanilla FedGC corresponds precisely to the expectation of our model, while our method additionally supplies the variance and cross-covariance structure that enables confidence intervals and early-warning detection.
>
> Therefore, in any simulated fault scenario, reporting both methods would simply show that vanilla FedGC yields the mean trajectory, whereas our uncertainty-aware formulation yields the mean and the confidence intervals whose widening reveals impending instability. Since vanilla FedGC is not capable of producing confidence bounds at all, it cannot participate meaningfully in tasks such as early-warning detection or robust root-cause analysis. Our method provides these capabilities by definition, through its stochastic characterization of FedGC’s parameter evolution.

---

### Official Review · Reviewer_3TSK · 2025-10-30

**Soundness:** 3
**Presentation:** 3
**Contribution:** 3
**Rating:** 6
**Confidence:** 4

**Summary:**

This paper presents a novel and rigorous theoretical framework for quantifying and propagating uncertainty within Federated Granger Causality (FedGC). The authors systematically classify uncertainty sources (aleatoric/epistemic) and derive closed-form recursions for their propagation through client-server interactions, identifying key cross-covariance terms. A significant theoretical contribution is proving that the steady-state uncertainties depend solely on client data statistics, not initial priors. The work is technically sound, addresses a clear gap in the literature, and is supported by synthetic and real-world experiments. However, it seems that some simplified assumptions are used in this paper. It is better the authors can explain the practical relevance of such assumptions or the reason why the analytical insights under such assumptions are helpful for advancing the development of your field.

**Strengths:**

- Novelty and Significance: This is the first work, to the best of this reviewer's knowledge, to rigorously formalize uncertainty propagation in a federated causal learning setting. The problem is well-motivated by real-world needs in safety-critical systems (e.g., the 2003 blackout example), and the solution represents a substantial contribution beyond deterministic FedGC methods.

- Theoretical Rigor: The paper is exceptionally strong on theory. The derivation of propagation recursions for variances and cross-covariances (Theorems 6.7, 6.8) is meticulous. The convergence analysis (Section 7) is a key highlight, demonstrating that the influence of initial epistemic priors vanishes at steady-state, a non-trivial and valuable result that enhances the method's robustness.

- Comprehensive Formulation: The systematic breakdown of uncertainty sources and their propagation paths (client-to-server, server-to-client, within-client, within-server) is clear and insightful. The identification of the four cross-covariance terms (Ω, Λ, Γ, Ψ) elegantly captures the complex coupling between data and model uncertainties in a federated system.

- Empirical Validation: The experiments on both synthetic and real-world (HAI, SWaT) datasets validate the theoretical claims. The results convincingly show how uncertainty evolves and converges, and how it is affected by aleatoric and epistemic noise, aligning with the theoretical predictions.

**Weaknesses:**

- It is better if the authors can explain the following assumptions:

   - Linearity and LTI: The core model is a Linear Time-Invariant (LTI) state-space system. While Appendix E briefly discusses extensions to EKF and GPs, these remain conceptual and lack empirical validation. Many real-world dynamical systems exhibit strong non-linearities.

   - Stationarity (A4): The assumption of weakly stationary client data is often violated in practice (e.g., due to system faults, operational changes, or trends). The paper acknowledges this limitation and shows in Fig. 1(e-f) that mean shifts cause performance degradation, but the proposed framework does not inherently handle non-stationarity.

   - No Process Noise (A5): Assuming Q=0 (no process noise) is a significant simplification. In real systems, the underlying state evolution is often noisy, and this omission may lead to an underestimation of total uncertainty.

- Scalability and Computational Complexity: The analysis in Appendix D.5 acknowledges that a naive implementation has a computational cost of O(p_m² d_m²) per client per round. While the proposed structural optimizations are noted, their efficacy is not demonstrated empirically. For high-dimensional client data (d_m large), this could be a severe bottleneck, potentially rendering the method impractical without further approximations.

- Limited Discussion on Practical Utility: The paper excellently shows that uncertainty can be quantified, but could do more to discuss how this uncertainty should be used in practice by a system operator.

   - How should one set thresholds on the predictive confidence intervals for actionable alerts?

   - Could the uncertainty estimates be used for active learning or to guide resource allocation among clients? A more concrete discussion on the operational decision-making pipeline enhanced by these uncertainty estimates would strengthen the impact.

**Questions:**

- How robust is the convergence and performance of the method when the stationarity assumption (A4) is mildly violated? Could the framework be combined with online change-point detection to reset the uncertainty estimates?
- It is better if the authors can also introduce some other related works:

   - FedCSL: a scalable and accurate approach to federated causal structure learning
   - Enhancing causal discovery in federated settings with limited local samples

---

> ### Author Response · Authors · 2025-11-27
> **Weakness 1 (Linearity and Stationarity Assumptions)**
>
> # Linearity and LTI
> We respectfully disagree on this comment. We have shown the structural illustration for nonlinear models in **Section E.1 (Appendix)**. The exact convergence guarantees rely only on differentiable state-update rules and standard covariance-composition identities, not on linearity itself. This means that whenever the underlying dynamical model admits a differentiable transition map, the propagation equations take the same algebraic form with Jacobians replacing the constant transition matrix. Establishing rigorous convergence guarantees for fully general nonlinear dynamical systems would require strong, model-specific conditions on Jacobian stability, smoothness, and curvature. Such assumptions are not available even for centralized nonlinear Granger-causal inference. Developing these guarantees lies beyond the scope of the current paper
> Then, the goal of this paper is to characterize uncertainty propagation in the mathematically tractable LTI setting in which FedGC itself is defined.
>
> We also want to highlight the importance of performing analysis on EKF and GP for the structural illustration of nonlinear models. EKF and GP are chosen intentionally as generic nonlinear alternatives. EKF already subsumes any differentiable nonlinear transition model, including neural-network–based Granger-causal operators, since it only requires first-order local linearizations of $f(\cdot)$. Thus, EKF and its standard extensions (such as UKF, higher-order filters) constitute the canonical machinery for propagating uncertainty in arbitrary neural or nonparametric dynamical systems. Similarly, GP state-space models provide a fully nonparametric representation of nonlinear transitions with closed-form posterior means and variances, offering an even broader modeling class. These two families are therefore not restrictive. Instead, we model general-purpose nonlinear dynamical systems, while also aligning with the structure of our covariance recursions.
>
> # Stationarity (A4)
> In the revised version of the manuscript, **Section E.2 (Appendix)** now provides an extended discussion of how our uncertainty–propagation framework can be adapted to non-stationary settings.
>
> Our analysis uses the classical assumption of weak stationarity because it enables closed-form propagation of uncertainty through the FedGC updates. However, the same recursions remain structurally valid when the empirical moments are replaced by \emph{exponentially weighted} (EWMA) moments. In this setting, each client updates
>
> $\mu_t = (1-\lambda)\mu_{t-1} + \lambda x_t,
> \qquad
> \Sigma_t = (1-\lambda)\Sigma_{t-1} + \lambda x_t x_t^\top,
> $
>
> with forgetting factor $0<\lambda<1$. When the underlying data exhibit slowly drifting moments $(\mu_t^*,\Sigma_t^*)$, standard results imply the following,
>
> $
> \|\mu_t - \mu_t^* \| = O(\delta/\lambda), \qquad
> \|\Sigma_t - \Sigma_t^* \| = O(\delta/\lambda),
> $
>
> where $\delta$ bounds the temporal drift, and $(\mu_t^*,\Sigma_t^*)$ are the true non-stationary moments of the underlying data. The above bounds show that EWMA can track the true moment as long as the true moment is not drifting faster than EWMA forgets older data. One main advantage of using this technique is that EWMA moments track non-stationarity while preserving the form of our theoretical results in uncertainty propagations. Inspired by several works in the machine-learning literature that handle non-stationarity through exponential weighting or forgetting factors [1-3], we adopt the same principle here.
>
> Beyond EWMA, other forms of non-stationarity can be incorporated by modifying the moment-estimation step. Examples include sliding-window estimators, seasonally adjusted or periodic-window estimators, trend-filtered or total-variation–regularized moment updates, and online convex-combination estimators for abrupt regime changes. Neural-network–based models can also accommodate complex non-stationarity, but doing so would require deriving a new set of uncertainty–propagation equations beyond the linear FedGC framework. A systematic treatment of these extensions is an important direction for future work.
>
> **References:**
>
> **[1]** Bozkurt, B., Pehlevan, C. and Erdogan, A., 2022. Biologically-plausible determinant maximization neural networks for blind separation of correlated sources. Advances in Neural Information Processing Systems, 35, pp.13704-13717.
>
> **[2]** Kurle, R., Cseke, B., Klushyn, A., Van Der Smagt, P. and Günnemann, S., 2019. Continual learning with bayesian neural networks for non-stationary data. In International Conference on Learning Representations.
>
> **[3]** Wang, T., Zhang, W., Ye, C., Wei, J., Zhong, H. and Huang, T., 2015. FD4C: Automatic fault diagnosis framework for web applications in cloud computing. IEEE Transactions on Systems, Man, and Cybernetics: Systems, 46(1), pp.61-75.

---

> ### Author Response · Authors · 2025-11-27
> **Weakness 1 (No Process Noise Assumption)**
>
> # No Process Noise (A5)
> We agree that assuming $Q=0$ is a simplifying choice. In our formulation, this assumption allows us to interpret aleatoric uncertainty purely as measurement (sensor) noise and to isolate the contribution of data variability in a transparent manner. If the underlying state evolution is itself noisy, i.e., $Q \neq 0$ then the aleatoric component would naturally include \emph{both} measurement noise and state-process noise. The propagation framework we derive remains valid in this case; the only additional engineering challenge lies in estimating $Q$ in a decentralized setting.
>
> Addressing $Q \neq 0$ would require extending FedGC, which currently estimates only the state-transition matrix $A$, to also estimate the state-process covariance $Q$ across clients. This is technically feasible but adds another layer of decentralized system identification. We view this as an important direction for future work.

---

> ### Author Response · Authors · 2025-11-27
> **Weakness 2 (Scalability and Computational Complexity)**
>
> # Scalability and Computational Complexity
> The purpose of this paper is to theoretically analyze how uncertainty propagates and converges in vanilla FedGC when the exchanged quantities are treated as random variables rather than deterministic updates. Our contribution is the closed-form characterization of these uncertainty recursions and their steady-state limits; a full engineering implementation of covariance propagation inside the FedGC training loop is outside the scope of this theoretical analysis. For this reason, we do not investigate implementation-level optimizations or runtime trade-offs for the naive $O(p_m^2 d_m^2)$ covariance update.
>
> Instead, our empirical evaluation focuses on empirical uncertainty estimation using multi-start FedGC (see response **Weakness 4 of Reviewer qSyr**) and shows that these empirical covariances follow the same qualitative trends predicted by our closed-form recursions. Practical scalability with respect to feature dimension $d_m$ is demonstrated directly in **Table 2**, where varying $d_m$ shows stable and consistent behavior of the uncertainty measures. This confirms scalability in the regime relevant to FedGC, even though the paper does not implement the naive worst-case covariance propagator.

---

> ### Author Response · Authors · 2025-11-27
> **Weakness 3 (Limited Discussion on Practical Utility)**
>
> # Limited Discussion on Practical Utility
> We emphasize that triggering alarms by thresholding one-step-ahead predictive confidence intervals is an ad hoc decision rule and is not the operational mechanism our uncertainty framework is designed to support. In FedGC, the quantities of interest are the server-side Granger-causal parameters $A$ (and client parameters $\theta$), and our uncertainty propagation characterizes the covariance of these causal parameters themselves. Because the underlying dynamics follow the linear state-space recursion
>
> $
> x_{t+1} = A x_t + w_t, \qquad w_t \sim \mathcal{N}(0,Q),
> $
>
> uncertainty in $A$ induces corresponding uncertainty in the latent state trajectory $x_t$. This viewpoint aligns naturally with reliability engineering: widening parameter covariance leads to widening state uncertainty, reducing operator confidence in the system’s health estimate.
>
> Accordingly, our uncertainty estimates can be used to guide maintenance timing and intervention decisions such as preventive shutdowns and repairs. This state uncertainty is a quantifiable probability and has typically been used in a plethora of reliability engineering and condition monitoring literature. A classic example is the age-replacement model, where the probability are weighted using the cost of failure vs correct of preventive intervention (preventive maintenance).
>
> $
> \min_{\tau} (1 - F(\tau)) \times C_p + F(\tau) \times C_f.
> $
>
> where $C_p$ denotes the cost of early maintenance, $C_f$ the cost of failure, and $F$ is the probability of failure before time $\tau$. The failure probability $F$ is captured by the uncertainty of the latent state evaluated using our uncertainty propagation framework. As $\tau$ varies, this objective yields a Pareto front between intervening too early (high $C_{p}$) and intervening too late (high expected $C_{f}$). Our uncertainty propagation provides the key input to this trade-off by quantifying how parameter uncertainty maps into state-risk evolution. The costs ($C_p,$ $C_f)$ drive the decision making depending on the criticality of the application. For example, higher critical applications will have higher $C_p$ to $C_f$ ratios.
>
> Vanilla FedGC **cannot** support such analyses: it produces only point estimates and therefore cannot compute parameter or state variances, failure probabilities, or maintenance trade-offs induced by $C_p, C_f$ above. While our method supplies the full uncertainty structure necessary for such operational models, extending this uncertainty analysis to an end-to-end decision-making pipeline is an important direction for future work.

---

> ### Author Response · Authors · 2025-11-27
> **Answers to the Questions**
>
> # Question 1
> Please refer to our response on "Stationarity (A4)" in **Weakness 1** above.
>
> # Question 2
> We thank the reviewer for pointing out these relevant works. We have now added both *FedCSL: A Scalable and Accurate Approach to Federated Causal Structure Learning* and *Enhancing Causal Discovery in Federated Settings with Limited Local Samples* to **Section 2** of the revised manuscript.
>
> Conceptually, these methods address horizontally federated causal discovery in settings where clients hold i.i.d. samples of the same variables and aim to recover a global causal graph via constraint-based or score-based
> structure learning. In contrast, our work studies vertically partitioned dynamical systems, where each client observes different subsets of the state vector and Granger-causal relations arise through the temporal recursion
> $x_{t+1} = A x_t$. Thus, the statistical model, the communication pattern, and the causal object of interest (state-transition matrix $A$ rather than a directed acyclic graph) differ
> fundamentally.
>
> Nevertheless, including these works strengthens our positioning within the broader federated causal discovery literature, and we now explicitly cite them in **Section 2** when discussing related approaches.

---

### Official Review · Reviewer_pVDz · 2025-10-30

**Soundness:** 3
**Presentation:** 3
**Contribution:** 3
**Rating:** 6
**Confidence:** 2

**Summary:**

The paper adds uncertainty awareness to federated Granger causality. It classifies uncertainties (the primary contributors), derives how they propagate through the federated loop (client→server messages, server→client gradients, and local updates), identifies four key cross‑covariances that couple them, and proves that with stable updates, uncertainties converge to values determined by data noise rather than initial beliefs. Empirically, the method’s uncertainty measures track noise levels and provide confidence intervals around causal links, which is crucial for risk‑aware operations in distributed, privacy‑constrained systems.

**Strengths:**

1. The first framework rigorously quantifies uncertainty and its propagation within federated Granger causality frameworks.
2. In federated platforms, the paper explores and characterizes the source of uncertainty, how propagation recursions occur, and estimates the impact of uncertainty.
3. Theoretically complete and comprehensive experience.

**Weaknesses:**

1. The assumption is a linear time‑invariant state‑space model, Gaussian noise, and (for tractability) in places where all randomness enters via measurement noise (no process noise). Real systems can be nonlinear and non‑Gaussian.
2.  While formulas are closed‑form, implementing full covariance and cross‑covariance tracking adds engineering complexity compared to point‑estimate FedGC.
3. For the section 5 source of uncertainty, Aleatoric and Epistemic uncertainty is the intrinsic one, right? There is no specific algorithm to identify the main contributors, right?
4. Double comma in Line 263
5. What is the causal structure before estimating the cumulative effect of uncertainties?

**Questions:**

Same as weaknesses.

---

> ### Author Response · Authors · 2025-11-27
> **Weaknesses 1-5**
>
> # Weakness 1
> We agree with the reviewer that real-world systems can be non-linear. In fact we have shown the structural illustration for nonlinear models in **Section E.1 (Appendix)**. The exact convergence guarantees rely only on differentiable state-update rules and standard covariance-composition identities, not on linearity itself. This means that whenever the underlying dynamical model admits a differentiable transition map, the propagation equations take the same algebraic form with Jacobians replacing the constant transition matrix. Establishing rigorous convergence guarantees for fully general nonlinear dynamical systems would require strong, model-specific conditions on Jacobian stability, smoothness, and curvature. Such assumptions are not available even for centralized nonlinear Granger-causal inference. Developing these guarantees lies beyond the scope of the current paper
> Then, the goal of this paper is to characterize uncertainty propagation in the mathematically tractable LTI setting in which FedGC itself is defined
>
> We also want to highlight the importance of performing analysis on EKF and GP for the structural illustration of nonlinear models. EKF and GP are chosen intentionally as generic nonlinear alternatives. EKF already subsumes any differentiable nonlinear transition model, including neural-network–based Granger-causal operators, since it only requires first-order local linearizations of $f(\cdot)$. Thus, EKF and its standard extensions (such as UKF, higher-order filters) constitute the canonical machinery for propagating uncertainty in arbitrary neural or nonparametric dynamical systems. Similarly, GP state-space models provide a fully nonparametric representation of nonlinear transitions with closed-form posterior means and variances, offering an even broader modeling class. These two families are therefore not restrictive. Instead, we model general-purpose nonlinear dynamical systems, while also aligning with the structure of our covariance recursions.
>
> # Weakness 2
> We appreciate the reviewer’s careful observation on this point. It is true that full covariance and cross-covariance tracking introduce additional engineering effort compared to point-estimate FedGC. This is precisely why one of the goals of our paper is to show that empirical covariance and cross-covariance computed from the actual federated optimization dynamics follow the same trends predicted by our closed-form derivations.
>
> # Weakness 3
> The reviewer is correct that Section~5 describes the sources of uncertainty (aleatoric vs. epistemic) rather than an algorithmic procedure for attributing them. This decomposition reflects the structure of FedGC: aleatoric uncertainty arises from measurement noise and client-specific data variability, whereas epistemic uncertainty originates from limited information and the prior on the parameters.
>
> Our steady-state analysis further clarifies their relative contributions. Under the FedGC convergence conditions, all terms involving the initial prior are multiplied by powers of a contraction operator and therefore decay geometrically as $t \to \infty$. Consequently, the effect of epistemic uncertainty (prior) vanishes asymptotically, and the limiting parameter covariance depends only on the data noise statistics (aleatoric component).
>
> Thus, while we do not propose a separate algorithm for identifying “main contributors,” the theory itself shows that epistemic uncertainty is transient, whereas aleatoric uncertainty is the sole driver of steady-state parameter uncertainty.
>
> # Weakness 4
> Apologies for the typo. It is now corrected in the revised pdf.
>
> # Weakness 5
> In our setting, the “causal structure” is the linear Granger-causal graph encoded by the state-transition matrix $A$ (and its client-specific blocks), exactly as in the vanilla FedGC formulation. We do not assume an additional, fixed causal structure beyond this; instead, FedGC first estimates $A$ from the data as a point estimate, and our uncertainty analysis is then performed around this learned structure.
>
> Concretely, once FedGC has produced an estimate $\hat A$, our framework characterizes how uncertainty in the entries of $\hat A$ (and in the client parameters) propagates through the state-space recursion and across the edges of the learned Granger graph, yielding the cumulative effect of uncertainties over time and across client/server locations.

---

### Official Review · Reviewer_qSyr · 2025-11-02

**Soundness:** 3
**Presentation:** 2
**Contribution:** 3
**Rating:** 6
**Confidence:** 3

**Summary:**

This paper presents the first theoretical framework for uncertainty quantification in Federated Granger Causality (FedGC) learning. The authors systematically classify uncertainty sources into aleatoric (data noise) and epistemic (model variability) components, derive closed-form recursion expressions for uncertainty propagation through client-server interactions, and identify four novel cross-covariance terms coupling data and model parameters. Key theoretical contributions include spectral-radius-based convergence conditions and proofs that steady-state variances depend exclusively on client data statistics, eliminating dependence on initial epistemic priors.

**Strengths:**

1. This is the first work to quantify uncertainty propagation in federated Granger causality systematically.
2. The proof that steady-state variances depend only on client data statistics (independent of initial epistemic priors) is both theoretically elegant and practically valuable. This robustness property ensures that poor initialization doesn't permanently affect uncertainty estimates, a crucial property for deployment in safety-critical systems.
3. The paper validates theoretical predictions through well-designed experiments on synthetic data with varying noise regimes, demonstrating that uncertainty evolution matches theoretical predictions.

**Weaknesses:**

1. The entire theoretical framework is limited to linear time-invariant (LTI) systems. While Appendix E discusses extensions to EKF and GP, these remain sketchy and lack rigorous convergence guarantees. Many real-world causal systems exhibit significant nonlinearity, severely limiting applicability.
2. The assumption of weakly stationary data with time-invariant moments is quite restrictive. Real-world industrial systems often exhibit distribution shifts, seasonal patterns, and non-stationary dynamics. Figure 1(e-f) shows performance degradation during mean shifts, but the paper provides insufficient guidance on handling non-stationarity beyond acknowledging it as a limitation.
3. What is the wall-clock time overhead of tracking uncertainties compared to vanilla FedGC? Table 2-3 show trace values but not runtime comparisons.
4. How does your method compare quantitatively to simpler approaches like: (a) ensemble-based uncertainty estimation, (b) Monte Carlo dropout in a federated setting, or (c) treating each client's parameter variance independently without cross-covariance terms?
5. In the 2003 blackout example (Introduction), you mention monitoring variance widening. What thresholds or decision rules would you recommend for triggering alarms based on uncertainty estimates?

**Questions:**

Please see Weaknesses.

---

> ### Author Response · Authors · 2025-11-27
> **Weaknesses 1 & 2**
>
> # Weakness 1
>
> We thank the reviewer, but we respectfully disagree on this comment. We have shown the structural illustration for nonlinear models in **Section E.1 (Appendix)**. The exact convergence guarantees rely only on differentiable state-update rules and standard covariance-composition identities, not on linearity itself. This means that whenever the underlying dynamical model admits a differentiable transition map, the propagation equations take the same algebraic form with Jacobians replacing the constant transition matrix. Establishing rigorous convergence guarantees for fully general nonlinear dynamical systems would require strong, model-specific conditions on Jacobian stability, smoothness, and curvature. Such assumptions are not available even for centralized nonlinear Granger-causal inference. Developing these guarantees lies beyond the scope of the current paper
> Then, the goal of this paper is to characterize uncertainty propagation in the mathematically tractable LTI setting in which FedGC itself is defined
>
> We also want to highlight the importance of performing analysis on EKF and GP for the structural illustration of nonlinear models. EKF and GP are chosen intentionally as generic nonlinear alternatives. EKF already subsumes any differentiable nonlinear transition model, including neural-network–based Granger-causal operators, since it only requires first-order local linearizations of $f(\cdot)$. Thus, EKF and its standard extensions (such as UKF, higher-order filters) constitute the canonical machinery for propagating uncertainty in arbitrary neural or nonparametric dynamical systems. Similarly, GP state-space models provide a fully nonparametric representation of nonlinear transitions with closed-form posterior means and variances, offering an even broader modeling class. These two families are therefore not restrictive. Instead, we model general-purpose nonlinear dynamical systems, while also aligning with the structure of our covariance recursions.
>
> # Weakness 2
>
> We thank the reviewer for raising this point. In the revised version of the manuscript, **Section E.2 (Appendix)** now provides an extended discussion of how our uncertainty–propagation framework can be adapted to non-stationary settings.
>
> Our analysis uses the classical assumption of weak stationarity because it enables closed-form propagation of uncertainty through the FedGC updates. However, the same recursions remain structurally valid when the empirical moments are replaced by \emph{exponentially weighted} (EWMA) moments. In this setting, each client updates
>
> $\mu_t = (1-\lambda)\mu_{t-1} + \lambda x_t,
> \qquad
> \Sigma_t = (1-\lambda)\Sigma_{t-1} + \lambda x_t x_t^\top,
> $
>
> with forgetting factor $0<\lambda<1$. When the underlying data exhibit slowly drifting moments $(\mu_t^*,\Sigma_t^*)$, standard results imply the following,
>
> $
> \|\mu_t - \mu_t^* \| = O(\delta/\lambda), \qquad
> \|\Sigma_t - \Sigma_t^* \| = O(\delta/\lambda),
> $
>
> where $\delta$ bounds the temporal drift, and $(\mu_t^*, \Sigma_t^*)$ are the true non-stationary moments of the underlying data. The above bounds show that EWMA can track the true moment as long as the true moment is not drifting faster than EWMA forgets older data. One main advantage of using this technique is that EWMA moments track non-stationarity while preserving the form of our theoretical results in uncertainty propagations. Inspired by several works in the machine-learning literature that handle non-stationarity through exponential weighting or forgetting factors [1-3], we adopt the same principle here.
>
> Beyond EWMA, other forms of non-stationarity can be incorporated by modifying the moment-estimation step. Examples include sliding-window estimators, seasonally adjusted or periodic-window estimators, trend-filtered or total-variation–regularized moment updates, and online convex-combination estimators for abrupt regime changes. Neural-network–based models can also accommodate complex non-stationarity, but doing so would require deriving a new set of uncertainty–propagation equations beyond the linear FedGC framework. A systematic treatment of these extensions is an important direction for future work.
>
> **References:**
>
> **[1]** Bozkurt, B., Pehlevan, C. and Erdogan, A., 2022. Biologically-plausible determinant maximization neural networks for blind separation of correlated sources. Advances in Neural Information Processing Systems, 35, pp.13704-13717.
>
> **[2]** Kurle, R., Cseke, B., Klushyn, A., Van Der Smagt, P. and Günnemann, S., 2019. Continual learning with bayesian neural networks for non-stationary data. In International Conference on Learning Representations.
>
> **[3]** Wang, T., Zhang, W., Ye, C., Wei, J., Zhong, H. and Huang, T., 2015. FD4C: Automatic fault diagnosis framework for web applications in cloud computing. IEEE Transactions on Systems, Man, and Cybernetics: Systems, 46(1), pp.61-75.

---

> ### Author Response · Authors · 2025-11-27
> **Weaknesses 3 & 4**
>
> # Weakness 3
>
> Our contribution is not a modification of the FedGC algorithm, but a principled uncertainty analysis of the random variables exchanged during its federated optimization. The theoretical recursions we derive describe how these uncertainties propagate over rounds and across client--server locations. Vanilla FedGC provides only point estimates of the Granger coefficients and therefore cannot quantify parameter or state uncertainty; this is the central limitation our work addresses. Since our method uses the same algorithmic pipeline as vanilla FedGC, the wall-clock training time is effectively identical. The critical conceptual difference is that we treat the exchanged quantities as random parameters rather than deterministic ones and explicitly track their uncertainty. The explicit characterization of uncertainty enables a rigorous analysis of the trade-off between competing operational objectives in high-consequence engineering applications, where risk and cost consequences depend directly on the level of uncertainty.
>
> The additional statistics reported in **Tables 2--3** (e.g., traces of propagated covariances) are obtained from lightweight closed-form evaluations on these random parameters. Their computational cost is negligible compared to running FedGC itself, so separate runtime measurements would not show any meaningful difference.
>
> # Weakness 4
> We thank the reviewer for this question. In **Section B.3 of the Appendix**, we now include quantitative comparisons to several natural baselines.
>
> For comparison, we include the following baselines:
>
>  **(1) Centralized GC:** The high-dimensional measurement data $y^t$ from all clients are pooled and $A_{21}$ is estimated centrally.
>
> **(2) Our:** Our method corresponds to ensemble FedGC (multi-start), where each run begins from an independent prior sample and the empirical covariance provides the uncertainty estimate.
>
> **(3) Independent Clients:** Obtained by setting $\eta_2 = 0$ in Eq.~(5), so each client ignores cross-dependencies and updates its block independently.
>
> **Monte Carlo Dropout.**
> MC-dropout is meaningful for neural networks, where dropping neurons approximates Bayesian model averaging. In linear state-space models such as FedGC, the analogue would require randomly dropping features (states), which directly breaks the physical transition $x_{t+1} = A x_t$ and leads to instability or divergence. For this reason, MC-dropout is not applicable in our case.

---

> ### Author Response · Authors · 2025-11-27
> **Weakness 5**
>
> # Weakness 5
> We emphasize that triggering alarms by thresholding one-step-ahead predictive confidence intervals is an ad hoc decision rule and is not the operational mechanism our uncertainty framework is designed to support. In FedGC, the quantities of interest are the server-side Granger-causal parameters $A$ (and client parameters $\theta$), and our uncertainty propagation characterizes the covariance of these causal parameters themselves. Because the underlying dynamics follow the linear state-space recursion
>
> $
> x_{t+1} = A x_t + w_t, \qquad w_t \sim \mathcal{N}(0,Q),
> $
>
> uncertainty in $A$ induces corresponding uncertainty in the latent state trajectory $x_t$. This viewpoint aligns naturally with reliability engineering: widening parameter covariance leads to widening state uncertainty, reducing operator confidence in the system’s health estimate.
>
> Accordingly, our uncertainty estimates can be used to guide maintenance timing and intervention decisions such as preventive shutdowns and repairs. This state uncertainty is a quantifiable probability and has typically been used in a plethora of reliability engineering and condition monitoring literature. A classic example is the age-replacement model, where the probability are weighted using the cost of failure vs correct of preventive intervention (preventive maintenance).
>
> $
> \min_{\tau} (1 - F(\tau)) \times C_p + F(\tau) \times C_f.
> $
>
> where $C_p$ denotes the cost of early maintenance, $C_f$ the cost of failure, and $F$ is the probability of failure before time $\tau$. The failure probability $F$ is captured by the uncertainty of the latent state evaluated using our uncertainty propagation framework. As $\tau$ varies, this objective yields a Pareto front between intervening too early (high $C_{p}$) and intervening too late (high expected $C_{f}$). Our uncertainty propagation provides the key input to this trade-off by quantifying how parameter uncertainty maps into state-risk evolution. The costs ($C_p,$ $C_f)$ drive the decision making depending on the criticality of the application. For example, higher critical applications will have higher $C_p$ to $C_f$ ratios.
>
> Vanilla FedGC **cannot** support such analyses: it produces only point estimates and therefore cannot compute parameter or state variances, failure probabilities, or maintenance trade-offs induced by $C_p, C_f$ above. While our method supplies the full uncertainty structure necessary for such operational models, extending this uncertainty analysis to an end-to-end decision-making pipeline is an important direction for future work.

---

### Author Response · Authors · 2025-12-04
**Comment to the Area Chair**

Dear Area Chair,

We prepared this brief note to concisely summarize the review process for our submission and to highlight the primary concerns raised by the reviewers along with the corresponding updates made in the revision. Based on all reviewer comments and discussions, we organized the feedback into four central themes and indicate below how we addressed each of them in the updated manuscript.

---

## Concern 1: Assumptions on linearity, stationarity, and absence of process noise

*How we addressed it:*
- We clarified that the uncertainty-propagation framework depends on differentiability and covariance-composition identities rather than strict linearity, and we illustrated this using EKF and GP state-space formulations (**Appendix E.1**).
- We expanded the discussion on handling non-stationary data through exponentially weighted moment tracking and other practical adaptation mechanisms (**Appendix E.2**).
- We explained that process noise can be treated as part of the aleatoric component, and we noted decentralized estimation as a direction for future work (also added to **Appendix E.2**).

---

## Concern 2: Practical interpretation and operational use of uncertainty

*How we addressed it:*
- We expanded our explanation of how parameter uncertainty translates to uncertainty in system-state evolution and supports reliability-driven decisions such as maintenance planning (**included directly in our rebuttal responses**).
- We clarified that thresholding predictive intervals is not the intended mechanism; rather, the framework supports principled risk evaluation and confidence assessment in the learned causal relations (**also conveyed in the rebuttal**).

---

## Concern 3: Computational cost, scalability, and runtime

*How we addressed it:*
- We clarified that our contribution is a theoretical analysis of uncertainty propagation and does not alter the FedGC optimization loop, so runtime remains unchanged (**addressed in our rebuttal text**).
- We highlighted structural simplifications already presented for efficient implementation of covariance-tracking if desired (**Appendix D.5**).
- We added scaling experiments that vary the number of clients and feature dimensions, showing consistent uncertainty behavior across configurations (**Appendix B.2, including Table 3**).

---

## Concern 4: Baseline comparisons, novelty of cross-covariance structure, and related-work positioning

*How we addressed it:*
- We added quantitative comparisons to centralized GC, multi-start FedGC, and independent-client updates (**Appendix B.3**).
- We explained why dropout-based uncertainty approximations are not applicable in state-space Granger models (**Appendix B.3**).
- We added citations requested by reviewers and clarified how horizontal causal-structure learning methods differ from our vertically partitioned Granger-causal setting (**Section 2**).
- We strengthened the novelty explanation that, to our knowledge, no prior federated-learning work derives closed-form parameter-covariance or cross-covariance propagation induced by the alternating client–server FedGC updates (**summarized explicitly in our rebuttal text**).

---

We are confident that our responses, together with the updates made to the manuscript, fully address the concerns raised by the reviewers and have materially improved the clarity and impact of the work. We greatly appreciate your careful consideration of these revisions in your final recommendation, and we would be glad to provide any additional information should anything remain unclear. Thank you very much for your time and engagement.

Best regards,
Authors

---

### Meta-Review · Area_Chair_P7pw · 2026-01-08

**Summary:**

The paper builds a theory for uncertainty quantification in Federated Granger Causality. It:
- derives closed-form recursive formulas for the covariance (including client–server cross-covariances),
- gives convergence conditions based on the spectral radius, and
- shows a nice steady-state property: uncertainty is determined by clients’ data statistics, not by how the model was initialized.

**Reviewer Concerns:**

3 reviewers see the work as novel and technically solid, with experiments that mostly match the theory.
1 reviewer stays strongly negative, arguing the assumptions are too restrictive (LTI/weak stationarity/no process noise), the paper gives limited practical guidance, and novelty/baselines aren’t convincing enough.

In the rebuttal/revision, the authors:
- clarify they are analyzing vanilla FedGC (not proposing a new algorithm),
- add stronger baselines (centralized GC, multi-start/ensemble FedGC, independent-client approximation),
- expand scalability tests (more clients and higher dimensions),
- discuss practical extensions (handling nonstationarity with EWMA moments, process-noise treatment, and using uncertainty for risk-aware decisions instead of ad hoc thresholding).

Remaining gaps:
- nonlinear/nonstationary extensions are mostly conceptual, and
- end-to-end utility plus runtime/engineering overhead are still not fully measured.

**Reviewer Scores:**

My guess on score changes:
- qSyr: likely 6 ->8 (their asks were mostly clarity/baselines/runtime, which the revision addresses).
- pVDz: likely 6->6  (already positive; mostly minor practicality/wording issues).
- 3TSK: likely 6->8 (assumptions/practical relevance and related work were strengthened).
- 4bfG: likely 2->2, more fundamental skepticism remains.

---

### Decision · Program_Chairs · 2026-01-26

Reject